# Feynman-Kac Correctors in Diffusion:
# Annealing, Guidance, and Product of Experts

**Marta Skreta** [* 1 2]  **Tara Akhound-Sadegh** [* 3 4]  **Viktor Ohanesian** [* 5]  **Roberto Bondesan** [5]  **Alán Aspuru-Guzik** [1 2]
**Arnaud Doucet** [6]  **Rob Brekelmans** [2]  **Alexander Tong** [† 7 4 8]  **Kirill Neklyudov** [† 7 4 9]

## Abstract

While score-based generative models are the model of choice across diverse domains, there are limited tools available for controlling inference-time behavior in a principled manner, e.g. for composing multiple pretrained models. Existing classifier-free guidance methods use a simple heuristic to mix conditional and unconditional scores to approximately sample from conditional distributions. However, such methods do not approximate the intermediate distributions, necessitating additional 'corrector' steps. In this work, we provide an efficient and principled method for sampling from a sequence of *annealed*, *geometric-averaged*, or *product* distributions derived from pretrained score-based models. We derive a weighted simulation scheme which we call FEYNMAN-KAC CORRECTORS (FKCs) based on the celebrated Feynman-Kac formula by carefully accounting for terms in the appropriate partial differential equations (PDEs). To simulate these PDEs, we propose Sequential Monte Carlo (SMC) resampling algorithms that leverage inference-time scaling to improve sampling quality. We empirically demonstrate the utility of our methods by proposing amortized sampling via inference-time temperature annealing, improving multi-objective molecule generation using pretrained models, and improving classifier-free guidance for text-to-image generation. Our code is available at https://github.com/martaskrt/fkc-diffusion.

---

[*]Equal contribution ,[†]Equal senior-authorship [1]University of Toronto [2]Vector Institute [3]McGill University [4]Mila - Quebec AI Institute [5]Imperial College London [6]Google DeepMind [7]Université de Montréal [8]Current AT affiliation: Duke University [9]Institut Courtois. Correspondence to: AT <ayt14@duke.edu>, KN <k.necludov@gmail.com>.

*Proceedings of the $42^{nd}$ International Conference on Machine Learning*, Vancouver, Canada. PMLR 267, 2025. Copyright 2025 by the author(s).

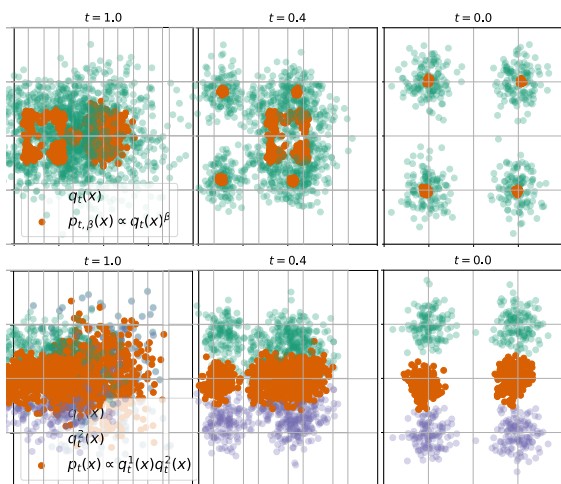

*Figure 1.* FEYNMAN-KAC CORRECTOR Inference for annealed $p_{t,\beta}(x) \propto q_t(x)^{\beta=10}$ and product $p_t(x) \propto q_t^1(x)q_t^2(x)$ densities.

## 1. Introduction

Score-based generative models, also known as diffusion models, have emerged as the model of choice across diverse generative tasks such as image generation, natural language, and protein simulation (Saharia et al., 2022; Sahoo et al., 2024; Abramson et al., 2024). These models leverage the ability to estimate scores of the sequence of noise-corrupted distributions and then use the learned scores to reverse the corruption process enabling high-quality generation. Thus, diffusion models aim to produce new samples from the same distribution as the training data.

However, the classical paradigm of generative modeling as the problem of reproducing the training data distribution becomes less relevant for many applications including drug discovery and text-to-image generation. In practice, generative models demonstrate the best performance when tailored to specific needs at inference time. For instance, linear combinations of scores allow for concept composition (Liu et al., 2022) or for increasing image-prompt consistency as in classifier-free guidance (CFG) (Ho & Salimans, 2021). However, by modifying the scores, one loses control over the marginal distributions of the generated samples. Various approaches from the Monte Carlo sampling literature have been adapted to 'correct' samples along a trajectory to more

closely match the prescribed intermediate distributions. Assuming access to an exact score, additional Langevin corrector steps with the desired invariant distribution can be applied with additional simulation steps as the only practical overhead (Song et al., 2021; Bradley & Nakkiran, 2024). However, these corrector schemes are only exact in the limit of infinite intermediate steps. Accept-reject or Sequential Monte Carlo techniques may be used when the score is parameterized through a scalar energy function (Du et al., 2023; Phillips et al., 2024), although these parameterizations require extra computation during training and may sacrifice expressivity in practice (Salimans & Ho, 2021; Thornton et al., 2025). While methods for sampling from mixtures or equiprobable regions of diffusion models have been proposed (Skreta et al., 2025), general solutions to accurately sample from combinations or temperings of flexibly-parameterized diffusion models with limited computational overhead remain elusive.

To address these challenges, we introduce FEYNMAN-KAC CORRECTOR (FKCs), which enable efficient and principled sampling from a sequence of *annealed*, *geometric-averaged*, or *product* distributions derived from pretrained diffusion models. To develop FEYNMAN-KAC CORRECTORS and test their efficacy, we make the following contributions:

- We propose a flexible recipe for constructing weighted stochastic differential equations (SDEs), which account for additional terms appearing when manipulating the distribution of generated samples.
- As our primary examples, we derive the correction terms for multiple heuristic schemes commonly used to approximate annealed, product, or geometric averaged distributions, including CFG (Sec. 3).
- To simulate these weighted SDEs, we propose a family of Sequential Monte Carlo (SMC) resampling schemes, which 'correct' a batch of simulated samples to closely approximate the intermediate target distributions (Sec. 4).
- For the problem of sampling from an unnormalized density, we demonstrate that FKC allows for sampling from a variety of temperatures without retraining (Sec. 5.2). Moreover, we demonstrate that a high-temperature learning, low-temperature inference scheme can be more efficient than the notoriously difficult task of directly training a sampler at a lower temperature.
- For pretrained diffusion models we demonstrate that adding FKC terms enhances compositional generation of molecules with multiple properties (Sec. 5.3) and classifier-free guidance for image generation (Sec. 5.1).

## 2. Background

### 2.1. Diffusion Models

Generative modeling via diffusion models can be formulated as the simulation of the Stochastic Differential Equation (SDE) corresponding to the reverse-time process.

In particular, during training, one gradually destroys samples from the data-distribution $p_{\text{data}}(x)$ by simulating the following noising SDE:

$$dx_\tau = f_\tau(x_\tau)d\tau + \sigma_\tau d\overline{W}_\tau\,, \quad x_{\tau=0} \sim p_{\text{data}}(x)\,, \quad (1)$$

where $f_\tau(x_\tau)$ is usually some linear drift function $f_\tau(x_\tau) = \alpha_\tau x_\tau$, $\sigma_\tau$ defines the scale of noise through time, and $d\overline{W}_\tau$ is the standard Wiener process. The drift $f_\tau$ and the diffusion coefficient $\sigma_\tau$ are chosen so the final density is close to the standard normal distribution $p_{\tau=1} \approx \mathcal{N}(0, I_d)$.

The generation process then can be defined as the family of denoising SDEs in the opposite time direction ($t = 1 - \tau$),

$$dx_t = \left(-f_t(x_t) + \sigma_t^2 \nabla \log p_t(x_t)\right)dt + \sigma_t dW_t\,, \quad (2)$$

where $p_t = p_{1-\tau}$ is the density of the marginals induced by the noising process in Eq. (1); hence, the process starts with $x_0 \sim \mathcal{N}(x \,|\, 0, I_d)$. By training a model of the score functions $\nabla \log p_t(\cdot)$, one can generate new samples from $p_{\text{data}}(x)$ using Eq. (2) (Song et al., 2021).

### 2.2. Feynman-Kac PDEs

While Eq. (2) describes a procedure for simulating individual particles, we can also derive Partial Differential Equations (PDEs) which describe the time-evolution of the density of samples $p_t(x)$ under this SDE. We begin by describing the relevant equations for the standard SDE case.

**(1) Continuity Equation**, which describes how the density changes when the samples move in space according to a flow or ODE with drift $v_t$

$$dx_t = v_t(x_t)dt \implies \frac{\partial p_t^{\text{ode}}(x)}{\partial t} = -\left\langle \nabla, p_t^{\text{ode}}(x)v_t(x) \right\rangle. \quad (3)$$

where $p_t^{\text{ode}}$ indicates the evolution only according to a flow.

**(2) Diffusion Equation**, which describes the change of the density for the pure Brownian motion with coefficient $\sigma_t$,

$$dx_t = \sigma_t dW_t \implies \frac{\partial p_t^{\text{diff}}(x)}{\partial t} = \frac{\sigma_t^2}{2}\Delta p_t^{\text{diff}}(x). \quad (4)$$

where $p_t^{\text{diff}}$ denotes evolution due to the diffusion term only.

The SDE in Eq. (2) can be viewed as the composition of a flow and diffusion terms, where the corresponding Fokker-Planck PDE describes the combined evolution

$$\frac{\partial p_t^{\text{sde}}(x)}{\partial t} = -\left\langle \nabla, p_t^{\text{sde}}(x)v_t(x) \right\rangle + \frac{\sigma_t^2}{2}\Delta p_t^{\text{sde}}(x). \quad (5)$$

However, our main focus in this work will be to study a third type of PDE, which will yield *weighted* SDEs that we eventually use to simulate a sequence of marginals other those the forward noising process $p_{1-\tau}$ (Sec. 3).

**(3) Reweighting Equation**, which describes the change of density when samples have time-dependent log-weights $w_t$

which are updated based on the positions of samples $x_t$,

$$dw_t = \bar{g}_t(x_t)dt \implies \frac{\partial p_t^w(x)}{\partial t} = \bar{g}_t(x)p_t^w(x),$$
$$\text{where} \quad \bar{g}_t(x) = g_t(x) - \int g_t(x)p_t^w(x)dx \quad (6)$$

where the last equation guarantees the conservation of the normalization constant, i.e. $\int dx\, \bar{g}_t(x)p_t^w(x) = 0$.

**Feynman-Kac Formula** We now focus on the combination of all three components to describe the *Feynman-Kac PDE*,

$$\frac{\partial p_t^{\text{FK}}(x)}{\partial t} = -\left\langle \nabla, p_t^{\text{FK}}(x)v_t(x) \right\rangle + \frac{\sigma_t^2}{2}\Delta p_t^{\text{FK}}(x)$$
$$+ \bar{g}_t(x)p_t^{\text{FK}}(x), \quad (7)$$

where to sample from $p_t^{\text{FK}}(x)$, one first has to sample $x_t$ via the following SDE

$$dx_t = v_t(x_t)dt + \sigma_t dW_t, \quad dw_t = \bar{g}_t(x_t)dt, \quad (8)$$

and then reweight the obtained samples using $w_t$. Thus, $p_t^{\text{FK}}(x)$ reflects the density of *weighted* samples, which differs from the density $p_t^{\text{sde}}(x)$ obtained via the Fokker-Planck PDE in Eq. (5) due to the addition of reweighting terms.

In practice, we can account for this difference by sampling

$$i \sim \text{Categorical}\left\{ \frac{\exp(w_T^k)}{\sum_{j=1}^K \exp(w_T^j)} \right\}_{k=1}^K, \quad (9)$$

and returning $x_T^{(i)}$ as an approximate sample from $p_T$. We discuss more refined resampling techniques in Sec. 4. For estimating the expectation of test functions $\phi$, we account for the weights by reweighting a collection of $K$ particles, i.e.,

$$\mathbb{E}_{p_T}[\phi(x)] \approx \sum_{k=1}^K \frac{\exp(w_T^k)}{\sum_j \exp(w_T^j)} \phi(x_T^k). \quad (10)$$

For justification of the validity of this weighting scheme for Feynman-Kac PDEs, see App. A. The expression in Eq. (10) corresponds to Self-Normalized Importance Sampling (SNIS) estimation, which converges to exact expectation estimators when $K \to \infty$ (e.g. Naesseth et al. (2019)).

### 2.3. Flexibility of Simulation for Given Marginals

Given a PDE describing the time-evolution of a particular density $p_t(x)$, there may exist multiple simulation methods. For instance, it is well-known that the diffusion equation (4) can be simulated using an ODE (Song et al., 2021).

**Diffusion → Continuity** Through simple manipulations, we can rewrite the diffusion equation using a continuity equation and change the simulation scheme accordingly

$$\frac{\partial p_t(x)}{\partial t} = \frac{\sigma_t^2}{2}\Delta p_t(x) = -\left\langle \nabla, p_t(x)\left(-\frac{\sigma_t^2}{2}\nabla \log p_t(x)\right) \right\rangle$$
$$\implies dx_t = -\frac{\sigma_t^2}{2}\nabla \log p_t(x_t)dt. \quad (11)$$

The reweighting equation adds an extra dimension to the interplay between different simulation schemes.

**Continuity → Reweighting** We first recast the continuity equation in terms of reweighting, in which case the simulation changes the density solely by adjusting the weights of samples (without transport),

$$\frac{\partial p_t(x)}{\partial t} = -\left\langle \nabla, p_t(x)v_t(x) \right\rangle = \left(\frac{-1}{p_t(x)}\left\langle \nabla, p_t(x)v_t(x) \right\rangle\right)p_t(x)$$
$$\implies dw_t = (-\left\langle \nabla, v_t(x_t) \right\rangle - \left\langle \nabla \log p_t(x_t), v_t(x_t) \right\rangle)dt \quad (12)$$

**Diffusion → Reweighting** We further observe that diffusion terms may be captured in the weights using

$$\frac{\partial p_t(x)}{\partial t} = \frac{\sigma_t^2}{2}\Delta p_t(x) = \frac{\sigma_t^2}{2}p_t(x)\big(\Delta \log p_t(x) + \|\nabla \log p_t(x)\|^2\big)$$
$$\implies dw_t = \frac{\sigma_t^2}{2}\big(\Delta \log p_t(x_t) + \|\nabla \log p_t(x_t)\|^2\big)dt \quad (13)$$

In particular, using Eqs. (12) and (13) we now have an approach for translating arbitrary flow $v_t$ or diffusion $\sigma_t$ terms into the reweighting factors, assuming access to an exact score function $\nabla \log p_t$. Such manipulations will play a key role in deriving our proposed methods in Sec. 3.

## 3. Modifying Diffusion Inference using Feynman-Kac Correctors

In this section, we propose new sampling tools for combining or modifying diffusion models at inference time using the Feynman-Kac PDEs in Sec. 2.2. To this end, consider several different pretrained diffusion models with marginals $\{q_t^i\}_{i=1}^M$ following

$$\frac{\partial q_t^i}{\partial t} = -\left\langle \nabla, q_t^i\big(-f_t + \sigma_t^2\nabla \log q_t^i\big)\right\rangle + \frac{\sigma_t^2}{2}\Delta q_t^i, \quad (14a)$$
$$dx_t = \big(-f_t(x_t) + \sigma_t^2\nabla \log q_t^i(x_t)\big)dt + \sigma_t dW_t, \quad (14b)$$

which is the denoising SDE from Eq. (2). Note that $q_t^i$ may arise from training on different datasets or correspond to conditional models with different conditioning. Throughout this work, we assume access to an exact score model $s_t^i(x; \theta^i) = \nabla \log q_t^i(x)$, in part to facilitate the conversion rules introduced in Sec. 2.3 and summarized in Table 1.

At inference time, we would like to sample from a modified target distribution involving these given models. While other variants are possible, we focus on the following examples:

$$\textbf{Annealed:} \quad p_{t,\beta}^{\text{anneal}}(x) = \frac{1}{Z_t(\beta)}q_t(x)^\beta$$

$$\textbf{Product:} \quad p_t^{\text{prod}}(x) = \frac{1}{Z_t}q_t^1(x)q_t^2(x) \quad (15)$$

$$\textbf{Geometric Avg:} \quad p_{t,\beta}^{\text{geo}}(x) = \frac{1}{Z_t(\beta)}q_t^1(x)^{1-\beta}q_t^2(x)^\beta.$$

A common heuristic for sampling from the distributions in the form of Eq. (15) is to simulate according to the score function of the target density. For example, in classifier-free guidance (Ho & Salimans, 2021) we use the score of the ge-

ometric average $\nabla \log p_{t,\beta}^{\text{geo}} = (1 - \beta)\nabla \log q_t^1 + \beta \nabla \log q_t^2$ to simulate the following SDE

$$dx_t = (-f_t(x_t) + \sigma_t^2 \nabla \log p_{t,\beta}^{\text{geo}}(x_t))dt + \sigma_t dW_t \,. \quad (16)$$

However, despite the similarity to Eq. (2), this heuristic does not sample from the prescribed marginals (including the final distribution), except in special cases. We proceed by using the $p_{t,\beta}^{\text{geo}}$ example to illustrate our approach.

### 3.1. Outline of Our Approach

To remedy this, we inspect the PDE corresponding to $p_{t,\beta}^{\text{geo}}$, which can be written in terms of the evolution of $q_t^1$ and $q_t^2$

$$\frac{\partial p_{t,\beta}^{\text{geo}}(x)}{\partial t} = \frac{\partial}{\partial t} \frac{1}{Z_t(\beta)} q_t^1(x)^{(1-\beta)} q_t^2(x)^{\beta} \,. \quad (17)$$

Expanding and using our expressions for the Fokker-Planck equation of $q_t^i$ in (14), we proceed to locate terms corresponding to the simulation of an SDE with the drift $v_t(x_t) = -f_t(x_t) + \sigma_t^2 \nabla \log p_{t,\beta}^{\text{geo}}(x_t)$. Collecting all remaining terms of PDE (17) into weights $\bar{g}_t(x_t)$ we obtain the following Feynman-Kac PDE, which can be simulated using the weighted SDE in Eq. (8), along with the resampling schemes described in Sec. 4

$$\frac{\partial p_{t,\beta}^{\text{geo}}}{\partial t} = -\left\langle \nabla, p_{t,\beta}^{\text{geo}} v_t \right\rangle + \frac{\sigma_t^2}{2} \Delta p_{t,\beta}^{\text{geo}} + p_{t,\beta}^{\text{geo}} \bar{g}_t \,. \quad (18)$$

**Conversion Rules** To facilitate the construction of Feynman-Kac PDEs corresponding to existing simulation schemes, in Table 1 we present the conversion rules that describe how the corresponding PDEs change for the annealed densities and the product of densities. We use these rules as building blocks when deriving our practical schemes.

**Computational Considerations** Our recipe above can yield many different weighted PDEs for a given sequence of target distributions. In practice, we would like our simulation scheme to closely approximate the intermediate targets distributions to limit the need for correction. On the other hand, for computational efficiency, we hope to obtain weights which avoid expensive divergence $\left\langle \nabla, v_t(x) \right\rangle$ or Laplacian terms $\left\langle \nabla, \nabla \log q_t^i(x) \right\rangle$. Remarkably, for linear drift functions $f_t(x)$ commonly used in diffusion models (Song et al., 2021), we find that simulating according to the common heuristic in Eq. (16) yields a Feynman-Kac PDE whose weights can be estimated with no additional overhead. We focus on these schemes in our examples.

### 3.2. Classifier-Free Guidance (CFG)

CFG (Ho & Salimans, 2021) is a widely-used procedure that simulates an SDE combining the scores of conditional and unconditional models with a guidance weight $\beta$,

$$\nabla \log p_{t,\beta}(x) = (1 - \beta)\nabla \log q_t^1(x \,|\, \emptyset) + \beta \nabla \log q_t^2(x \,|\, c)$$

In practice, $q_t^1(x|\emptyset)$ may represent an unconditional model (or a model with an empty prompt) whereas $q_t^2(x|c)$ is conditioned on a text prompt, class, or other random variables (Ho & Salimans, 2021). Alternatively, in autoguidance techniques, $q_t^1$ may be an undertrained version of a stronger conditional or unconditional model $q_t^2$ (Karras et al., 2024a).

For our purposes, we will view CFG as an attempt to sample from the geometric average distributions $p_{t,\beta}^{\text{geo}}(x) \propto q_t^1(x)^{1-\beta} q_t^2(x)^{\beta}$. Using the conversion rules in Table 1, we derive the reweighting terms which facilitate consistent sampling along the trajectory.

---

**Proposition 3.1** (Classifier-Free Guidance + FKC). *Consider two diffusion models $q_t^1(x), q_t^2(x)$ defined via (14). The weighted SDE corresponding to the geometric average of the marginals $p_{t,\beta}^{\text{geo}}(x) \propto q_t^1(x)^{1-\beta} q_t^2(x)^{\beta}$ is*

$$dx_t = \sigma_t^2((1-\beta)\nabla \log q_t^1(x_t) + \beta \nabla \log q_t^2(x_t))dt$$
$$- f_t(x_t)dt + \sigma_t dW_t \,, \quad (19)$$
$$dw_t = \frac{\sigma_t^2}{2}\beta(\beta-1)\left\|\nabla \log q_t^1(x_t) - \nabla \log q_t^2(x_t)\right\|^2 dt \,.$$

---

In Prop. D.3, we provide a more general formulation of this proposition outlining a continuous family of weighted SDEs sampling from the geometric average $p_{t,\beta}^{\text{geo}}(x) \propto q_t^1(x)^{1-\beta} q_t^2(x)^{\beta}$. As a further example, we combine CFG with a product of experts in Prop. D.4.

### 3.3. Annealed Distribution

Next, we consider a single diffusion model with the learned score $\nabla \log q_t(x)$, which we use to sample from the *annealed* or *tempered* density

$$p_{t,\beta}^{\text{anneal}}(x) = q_t(x)^{\beta}/Z_t(\beta) \,. \quad (20)$$

For $\beta > 1$, this can be used to generate samples from modes or high-probability regions of given models (Karczewski et al., 2025), while in Sec. 5.2 we explore the use of annealed inference in learning diffusion samplers from Boltzmann densities. The annealed target can be shown to admit the following Feynman-Kac weighted simulation scheme.

---

**Proposition 3.2** (Annealed SDE + FKC). *Consider a diffusion model $q_t(x)$ defined via (14). Sampling from the annealed marginals $p_{t,\beta}^{\text{anneal}}(x) \propto q_t(x)^{\beta}$, $\beta > 0$ can be performed by simulating the following weighted SDE*

$$dx_t = (-f_t(x_t) + \eta\sigma_t^2 \nabla \log q_t(x_t))dt + \zeta\sigma_t dW_t \,,$$
$$dw_t = (\beta-1)\left(\left\langle \nabla, f_t(x_t) \right\rangle + \frac{\sigma_t^2}{2}\beta\|\nabla \log q_t(x_t)\|^2\right)dt \,,$$

*with the coefficients (for $(\beta + (1-\beta)2a)/\beta \geq 0$)*

$$\eta = \beta + (1-\beta)a \,, \quad \zeta = \sqrt{(\beta + (1-\beta)2a)/\beta} \,. \quad (21)$$

---

| Original FK-PDE | Original wSDE | Annealed PDE | Annealed SDE $dx_t =$ | FK Corrector $dw_t$ += | Proof |
|---|---|---|---|---|---|
| $-\langle\nabla, q_t v_t\rangle$ | $v_t(x_t)dt$ | $-\langle\nabla, p_{t,\beta}v_t\rangle$ | $v_t(x_t)dt$ | $-(\beta-1)\langle\nabla, v_t\rangle dt$ | Prop. C.1 |
| | | $-\langle\nabla, p_{t,\beta}\beta v_t\rangle$ | $\beta v_t(x_t)dt$ | $\beta(\beta-1)\langle\nabla\log q_t, v_t\rangle dt$ | Prop. C.2 |
| $\frac{\sigma_t^2}{2}\Delta q_t$ | $\sigma_t dW_t$ | $\frac{\sigma_t^2}{2}\Delta p_{t,\beta}$ | $\sigma_t dW_t$ | $-\beta(\beta-1)\frac{\sigma_t^2}{2}\|\nabla\log q_t\|^2 dt$ | Prop. C.3 |
| | | $\frac{\sigma_t^2}{2\beta}\Delta p_{t,\beta}$ | $\frac{\sigma_t}{\sqrt{\beta}}dW_t$ | $(\beta-1)\frac{\sigma_t^2}{2}\Delta\log q_t dt$ | Prop. C.4 |
| $g_t q_t$ | $dw_t = g_t dt$ | $\beta g_t p_{t,\beta}$ | — | $\beta g_t dt$ | Prop. C.5 |
| — | — | time-dependent annealing: $\beta\to\beta_t$ | | $\frac{\partial\beta_t}{\partial t}\log q_t dt$ | Prop. C.6 |

| Original FK-PDE | Original wSDE | Product PDE | Product SDE $dx_t =$ | FK Corrector $dw_t$ += | |
|---|---|---|---|---|---|
| $-\langle\nabla, q_t v_t^{1,2}\rangle$ | $v_t^{1,2}dt$ | $-\langle\nabla, p_t(v_t^1+v_t^2)\rangle$ | $(v_t^1+v_t^2)dt$ | $(\langle\nabla\log q_t^1, v_t^2\rangle+\langle\nabla\log q_t^2, v_t^1\rangle)dt$ | Prop. C.7 |
| $\frac{\sigma_t^2}{2}\Delta q_t^{1,2}$ | $\sigma_t dW_t$ | $\frac{\sigma_t^2}{2}\Delta p_t$ | $\sigma_t dW_t$ | $-\sigma_t^2\langle\nabla\log q_t^1, \nabla\log q_t^2\rangle dt$ | Prop. C.8 |
| $g_t^{1,2}q_t^{1,2}$ | $dw_t = g_t^{1,2}dt$ | $(g_t^1+g_t^2)p_t$ | — | $(g_t^1+g_t^2)dt$ | Prop. C.9 |

*Table 1.* Conversion rules for different terms of the original Feynman-Kac PDEs (FK-PDEs) and the corresponding weighted SDE (wSDE). For every term term corresponding to the original densities $q_t$ (first two columns), we present the terms corresponding to the annealed marginals $p_{t,\beta}(x) \propto q_t(x)^\beta$ (top part) and the terms corresponding to the product of marginals $p_t(x) \propto q_t^1(x)q_t^2(x)$ (bottom part). Importantly, the *correctors are additive* in the weight space, e.g. when transforming the Fokker-Planck equation, we transform both the continuity & diffusion equation terms and sum the corresponding correctors. References to proofs are provided in the right-most column.

See Prop. D.1 for proof, and note that linear drifts $f_t(x)$ will lead to constant divergence terms which cancel upon reweighting in (9) and (10). We detail two choices of $a$.

**Target Score Simulation** For $a = 0$, we have $\eta = \beta$ and $\zeta = 1$, which yields the *target score* SDE whose drift corresponds to the score of the annealed target,

$$dx_t = (-f_t(x_t) + \beta\sigma_t^2\nabla\log q_t(x_t))dt + \sigma_t dW_t\,. \quad (22)$$

**Tempered Noise Simulation** For $a = 1/2$, we have $\eta = (1 + \beta)/2, \zeta = 1/\sqrt{\beta}$. We refer to this as an SDE with *tempered noise*, namely

$$dx_t = (-f_t(x_t) + \frac{\beta + 1}{2}\sigma_t^2\nabla\log q_t(x_t))dt + \frac{\sigma_t}{\sqrt{\beta}}dW_t\,. \quad (23)$$

We focus on these two choices of $a$, but note that for different $\beta$, we found that either target score or tempered-noise simulation could perform better in practice (Sec. 5).

### 3.4. Product of Experts (PoE)

Intuitively, samples from the product of densities correspond to the generations that have high likelihood values under *both* models. The product can also be interpreted as unanimous vote of experts, since a sample is not accepted if one of the densities is zero. Formally, consider the density

$$p_t^{\text{prod}}(x) = q_t^1(x)q_t^2(x)/Z_t\,. \quad (24)$$

For conditional generative models, the product of densities can describe samples satisfying several conditions. For example, in image generation, we could use $q(x\,|\,\text{``horse''})q(x\,|\,\text{``a sandy beach''})$ to generate images of "a horse on a sandy beach" (Du et al., 2023). In Sec. 5.3, we demonstrate that the PoE target can be used to improve molecule generations which satisfy multiple conditions simultaneously.

Again, a natural heuristic is to use the score of the target product density in the reverse-time SDE (2),

$$\nabla\log p_t^{\text{prod}}(x) = \nabla\log q_t^1(x_t) + \nabla\log q_t^2(x_t)\,, \quad (25)$$

In the following proposition, we further combine these rules with the annealing procedure to present the weighted SDE that samples from the marginals $p_{t,\beta}^{\text{prod}}(x) \propto (q_t^1(x)q_t^2(x))^\beta$.

**Proposition 3.3** (Product of Experts + FKC)**.** *Consider two diffusion models $q_t^1(x), q_t^2(x)$ defined via (14). The weighted SDE corresponding to the product of the marginals $p_{t,\beta}^{prod}(x) \propto (q_t^1(x)q_t^2(x))^\beta$ , with $\beta > 0$ is*

$$dx_t = \sigma_t^2\eta\big(\nabla\log q_t^1(x_t) + \nabla\log q_t^2(x_t)\big)dt \\ - f_t(x_t)dt + \zeta\sigma_t dW_t\,, \quad (26)$$

$$dw_t = \beta(\beta-1)\frac{\sigma_t^2}{2}\big\|\nabla\log q_t^1(x_t) + \nabla\log q_t^2(x_t)\big\|^2 dt \\ + \beta\sigma_t^2\langle\nabla\log q_t^1(x_t), \nabla\log q_t^2(x_t)\rangle dt \\ + (2\beta-1)\langle\nabla, f_t(x_t)\rangle dt\,, \quad (27)$$

*with the coefficients (for $(\beta + (1 - \beta)2a)/\beta \geq 0$)*

$$\eta = \beta + (1 - \beta)a\,, \quad \zeta = \sqrt{(\beta + (1 - \beta)2a)/\beta}\,. \quad (28)$$

See proof in Prop. D.2. Again, note that for linear drifts, the divergence term $\langle\nabla, f_t(x)\rangle$ is constant and can be ignored. Further, for $\beta = 1$, the first term in the weight evolution vanishes to leave only the inner product of score vectors. Similarly to Eqs. (22) and (23) for annealing, we have the *target score* SDE ($a = 0, \eta = \beta, \zeta = 1$) and the *tempered noise* SDE ($a = 1/2, \eta = (\beta + 1)/2, \zeta = 1/\sqrt{\beta}$).

More generally, we derive the weighted SDE that samples from $p_{t,\beta}(x) \propto \prod_i q_t^i(x)^{\beta_i}$, i.e. the weighted product of marginal densities $q_t^i(x)$ for arbitrary number of diffusion models (see Prop. D.5).

### 3.5. Reward-tilted Target Density

Finally, our framework can easily incorporate a reward function $r(x)$ defined on the state-space at inference time. Namely, we assume that the function $\exp(\beta_t r(x))$ is normalizable and consider the reward-tilted density $p_t^{\text{reward}}(x) \propto q_t(x) \exp(\beta_t r(x))$. Despite its similarity to the product of densities, this case is different as we do not assume $\exp(\beta_t r(x))$ changes according to the diffusion process.

---

**Proposition 3.4** (Reward-tilted Target + FKC). *Consider a diffusion model $q_t(x)$ defined via (14). Sampling from the reward-tilted marginals $p_t^{\text{reward}}(x) \propto q_t(x) \exp(\beta_t r(x))$ is performed by the following weighted SDE*

$$dx_t = \sigma_t^2 (\nabla \log q_t(x_t) + \frac{\beta_t}{2} \nabla r(x_t)) dt - \quad (29)$$
$$- f_t(x_t) dt + \sigma_t dW_t,$$

$$dw_t = \frac{\partial \beta_t}{\partial t} r(x_t) dt - \langle \beta_t \nabla r(x_t), f_t(x_t) \rangle dt + \quad (30)$$
$$+ \left\langle \beta_t \nabla r(x_t), \frac{\sigma_t^2}{2} \nabla \log q_t(x_t) \right\rangle dt.$$

---

See proof in Prop. D.6. Here, the weights increase when the vector field of the diffusion models aligns with the gradient of the reward function.

## 4. Resampling Methods

In this section, we describe several options for utilizing the weights to improve sampling with a batch of $K$ particles. While the simplest technique would be to simulate the weighted SDE in Eq. (8) for $K$ independent particles across the full time interval $t \in [0, 1]$ and reweight using SNIS in (10), we expect these full-trajectory weights to have high variance in practice due to error accumulation.

**Sequential Monte Carlo** Since our weights provide a proper weighting scheme for all intermediate distributions ((Naesseth et al., 2019), App. A), we can leverage SMC techniques which reweight particles along our trajectories.

In practice, we find that resampling only over an 'active interval' $t \in [t_{\min}, t_{\max}]$ is useful for improving sample quality and preserving diversity, and set weights to zero outside of this interval. Within the active interval, we resample at each step based on the increment $w_t^{(k)} = g_t(x_t^{(k)}) dt$, using systematic sampling proportional to $\exp\{w_t^{(k)}\}$ (Douc & Cappé, 2005). For small discretizations $dt$, we might expect relatively low-variance weights. From this perspective, systematic resampling is an attractive selection mechanism as

all particles are preserved in the case of uniform weights.

**Jump Process Interpretation of Reweighting** Finally, by reframing the reweighting equation in terms of a Markov jump process (Ethier & Kurtz (2009, Ch. 4.2)), a variety of further simulation algorithms for Feynman-Kac PDEs are possible (Del Moral (2013, Ch. 1.2.2, 5); Rousset & Stoltz (2006); Angeli (2020)).

A Markov jump process is determined by a rate function $\lambda_t(x)$, which governs the frequency of jump events, and a Markov transition kernel $J_t(y|x)$, which is used to sample the next state when a jump occurs. The forward Kolmogorov equation for a jump process is given by

$$\frac{\partial p_t^{\text{jump}}(x)}{\partial t} = \left( \int \lambda_t(y) J_t(x|y) p_t(y) dy \right) - p_t(x) \lambda_t(x)$$

where the two terms can intuitively be seen to measure the inflow and outflow of probability due to jumps.

Our goal is to find $\lambda_t(x), J_t(y|x)$ such that $p_t^{\text{jump}}$ matches the evolution of $p_t^w$ in Eq. (6) for a given choice of $g_t$. In fact, there are many possible jump processes which satisfy this property (Del Moral (2013, Ch. 5); Angeli et al. (2019)) We present a particular choice here, with proof in App. B.2.

---

**Proposition 4.1.** *For a given $g_t$ in Eq. (6), define the jump process rate and transition as*

$$\lambda_t(x) = \left( g_t(x) - \mathbb{E}_{p_t}[g_t] \right)^- \quad (31a)$$

$$J_t(y|x) = \frac{\left( g_t(y) - \mathbb{E}_{p_t}[g_t] \right)^+ p_t(y)}{\int \left( g_t(z) - \mathbb{E}_{p_t}[g_t] \right)^+ p_t(z) dz} \quad (31b)$$

*where $(u)^- := max(0, -u)$ and $(u)^+ := max(0, u)$. Then,*

$$\frac{\partial p_t^{\text{jump}}(x)}{\partial t} = \frac{\partial p_t^w(x)}{\partial t} = p_t(x) \left( g_t(x) - \mathbb{E}_{p_t}[g_t] \right) \quad (32)$$

*which matches Eq. (6).*

---

In continuous time and the mean-field limit, this jump process formulation of reweighting corresponds to simulating

$$x_{t+dt} = \begin{cases} x_t & \text{w.p. } 1 - \lambda_t(x_t) dt + o(dt) \\ \sim J_t(y|x_t) & \text{w.p. } \lambda_t(x_t) dt + o(dt). \end{cases} \quad (33)$$

We expect this process to improve the sample population in efficient fashion, since jump events are triggered only in states where $(g_t(x) - \mathbb{E}_{p_t}[g_t])^- \geq 0 \implies g_t(x) \leq \mathbb{E}_{p_t}[g_t]$, and transitions are more likely to jump to states with high excess weight $(g_t(y) - \mathbb{E}_{p_t}[g_t])^+ > 0$.

In practice, we use an empirical approximation $p_t^K(z) = \frac{1}{K} \sum_{k=1}^K \delta_z(x^{(k)})$ to approximate the jump rate $\lambda_t(x)$ and transition $J_t(y|x)$. Instead of simulating Eq. (33) directly, one can also adopt an implementation based on birth-death 'exponential clocks' (BDC, Del Moral (2013, Ch. 5.3-4), see App. B.3).

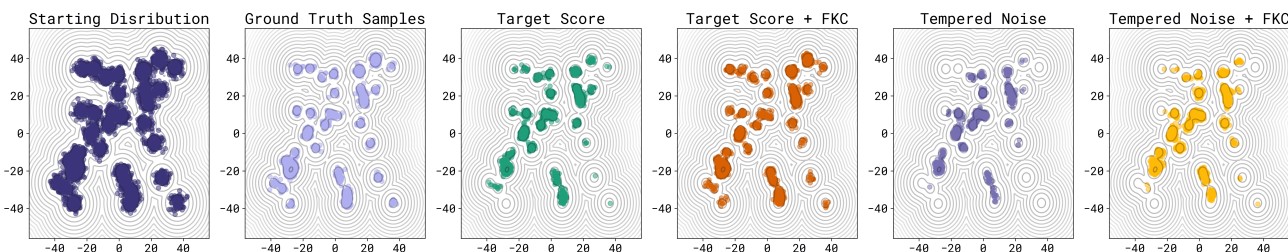

*Figure 2.* Samples from Mixture of 40 Gaussians.

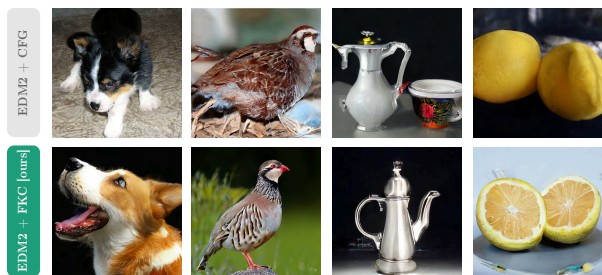

*Figure 3.* Samples from EDM2+CFG (top), EDM2+FKC (bottom).

# 5. Empirical Study

In this section, we compare our Feynman-Kac corrector (FKC) resampling schemes against their corresponding SDEs without resampling. We consider both target score and tempered noise SDEs. While we show results for BDC sampling in App. F.2 Table A1, we proceed with systematic resampling throughout the remainder of our experiments.

## 5.1. Image Generation with EDM2

In this section, we study the effect of FKC resampling for image generation in RGB pixel space, using CFG with an EDM2-XS model trained on ImageNet-512 (Karras et al., 2024b). In particular, we test whether resampling to more closely match the intermediate geometric average distributions translates to improvement in two downstream image quality metrics: CLIP Score (Radford et al., 2021) and ImageReward (Xu et al., 2024). CLIP Score measures the cosine similarity between the image and text prompt embeddings; ImageReward assigns a score that reflects human preferences (aesthetic quality and prompt adherence).

For a fixed simulation scheme, we compare the effect of adding FKC resampling (✅) versus the standard baseline without resampling (❌). We report results across various simulation parameters, namely the number of sampling steps $N$ and churn parameter $\gamma$ (which controls how the SDE integration scheme adds noise). For FKC, we additionally sweep over the batch size or number of particles $K$, whereas $K = 1$ corresponds to the no-resampling baseline (❌). To calculate metrics on a single image for FKC, we resample from among $K$ particles according to the weights since the last resampling step. Note that we often observe that final-step images from a single batch with FKC are nearly identical visually due to weight degeneracy.

In Table 2, we compare the quantitative performance of our

*Table 2.* Comparison of EDM2+FKC (✅) with EDM2+CFG (❌) for image generation using EDM2. We sweep over noise level ($\gamma$) and steps (N). For all metrics, we report CLIP and ImageReward (IR) scores averaged over 10,000 images.

| FKC | $\gamma$ | N | CLIP (↑) | IR (↑) | FKC | $\gamma$ | N | CLIP (↑) | IR (↑) |
|---|---|---|---|---|---|---|---|---|---|
| ❌ | 10 | 32 | 28.74 | −0.25 | ❌ | 40 | 16 | 28.67 | −0.30 |
| ✅ | 10 | 32 | 28.97 | 0.03 | ✅ | 40 | 16 | 29.12 | −0.01 |
| ❌ | 40 | 32 | 28.75 | −0.24 | ❌ | 40 | 32 | 28.75 | −0.24 |
| ✅ | 40 | 32 | **29.00** | **0.04** | ✅ | 40 | 32 | **29.14** | 0.05 |
| ❌ | 80 | 32 | 28.75 | −0.24 | ❌ | 40 | 64 | 28.81 | −0.19 |
| ✅ | 80 | 32 | 28.99 | **0.04** | ✅ | 40 | 64 | 29.12 | **0.07** |

FKC resampling against vanilla CFG. We find that adding FKC (✅) improves performance in both ImageReward and CLIP score, indicating both higher prompt adherence and aesthetically better images. While this comes at the cost of extra computation due to $K > 1$, we find that FKC demonstrates benefits even for $K = 2$, with $K = 8$ performing the best (Table A3). Qualitative results in Fig. 3 further support the finding that FKC can improve image quality.

In App. F.5, we provide an additional analysis on using FKC with latent diffusion image models.

## 5.2. Samplers from the Boltzmann Density

As described in the Sec. 1, our FKC inference techniques suggest flexible schemes for learning diffusion samplers at a given temperature and sampling according to a different temperature. Since we are given an energy function in these settings, we are not restricted to learning with temperature 1 for for our base model $q_t$. Thus, we use $(T_L, T_S)$ to refer to the learning ($q_t$) and sampling target ($p_{t,\beta}$) distributions, with $\beta = T_L/T_S$ in the notation of Sec. 3.3.

**Mixture of 40 Gaussians with Ground-Truth $q_t^\beta$**  To verify our tools in a tractable setting, we consider a highly multimodal distribution where we can calculate the optimal $q_t$ and $\nabla \log q_t$ for (small) integer $T_L$. We show qualitative results in Fig. 2. We find that target score + FKC performs best, while tempered noise has a tendency to drop modes. We also find that FKC outperforms SDE-only simulation in both tempered noise and target score settings. This is further supported by quantitative results in Table A1.

**Sampling LJ-13**  To demonstrate the utility of first learning a sampler at a high temperature then annealing to a lower temperature vs. directly learning at a lower temperature, we consider a Lennard-Jones (LJ) system of 13 particles at a base temperature $T_L = 2$. We train a Denoising Energy

*Table 3.* LJ-13 sampling task with various SDEs, with performance measured by mean $\pm$ standard deviation over 3 seeds. The starting temperature is $T_L = 2$, annealed to target temperatures $T_S = 0.8$ and $T_S = 1.5$. The DEM samples are generated with a model trained at those corresponding target temperatures.

| Target Temp. | SDE Type | FKC | Distance-$\mathcal{W}_2$ | Energy-$\mathcal{W}_1$ | Energy-$\mathcal{W}_2$ |
|---|---|---|---|---|---|
| 0.8 ($\beta = 2.5$) | Target Score | ✗ | $0.189 \pm 0.002$ | $14.730 \pm 0.029$ | $15.556 \pm 0.045$ |
| | | ✓ | $\mathbf{0.048 \pm 0.019}$ | $\mathbf{6.252 \pm 2.710}$ | $\mathbf{6.356 \pm 2.673}$ |
| | Tempered Noise | ✗ | $0.108 \pm 0.007$ | $\mathbf{6.487 \pm 0.056}$ | $8.501 \pm 0.283$ |
| | | ✓ | $\mathbf{0.047 \pm 0.006}$ | $7.016 \pm 0.538$ | $7.111 \pm 0.535$ |
| | DEM | — | $0.103 \pm 0.001$ | $9.794 \pm 0.100$ | $9.804 \pm 0.101$ |
| 1.5 ($\beta = 1.33$) | Target Score | ✗ | $0.168 \pm 0.009$ | $5.340 \pm 0.054$ | $6.210 \pm 0.254$ |
| | | ✓ | $0.083 \pm 0.003$ | $3.366 \pm 0.083$ | $3.386 \pm 0.090$ |
| | Tempered Noise | ✗ | $0.095 \pm 0.006$ | $2.154 \pm 0.048$ | $3.920 \pm 0.258$ |
| | | ✓ | $\mathbf{0.066 \pm 0.002}$ | $\mathbf{0.765 \pm 0.156}$ | $\mathbf{0.939 \pm 0.171}$ |
| | DEM | — | $0.268 \pm 0.005$ | $4.471 \pm 0.105$ | $5.211 \pm 0.017$ |

Matching (DEM) model (Akhound-Sadegh et al., 2024) at this base temperature and perform temperature-annealed inference to lower temperatures. In Table 3 and A2 we compare the performance of a DEM model trained at a lower temperature against a DEM model trained at a higher temperature and annealed to the lower temperature using various SDEs. We evaluate methods using the 2-Wasserstein metric between distance distributions, and the 1- and 2-Wasserstein metrics between energy histograms to a reference (App. F.3). We find that tempered noise+FKC performs best at higher temperatures. However, at lower temperatures, the target score SDE+FKC performs best. Both methods outperform DEM directly trained at the lower temperature for temperatures $T_S \in [2.0, 0.8]$ (Fig. 4). We find DEM is qualitatively easier to learn at higher temperatures requiring much less tuning compared to lower temperatures (Fig. A1). This makes the train-then-anneal approach attractive in this setting. For extended results and discussion see App. F.

### 5.3. Multi-Target Structure-Based Drug Design

We apply FKC to the setting of structure-based drug design (SBDD), where the goal is to design molecules (or ligands) using the three-dimensional structure of a biological target—typically a protein—as a guide (Anderson, 2003). The ligands are then evaluated based on how well they fit into the protein's binding site. We focus on dual-target drug design, where a molecule should interact with two proteins simultaneously. Dual-target drug design has become increasingly investigated for targeting complex disease pathways such as in various cancers and neurodegeneration (Ramsay et al., 2018), as well as for diminishing drug resistance mechanisms (Yang et al., 2024).

We investigate the performance of PoE using both target score and tempered noise SDEs at various $\beta$, with (✓) and without (✗) FKC. Ligand performance is determined by docking scores to each protein target using AutoDock Vina (Eberhardt et al., 2021). We evaluate 100 protein pairs and average our results over tasks. We sampled 5 molecule sizes from the original training set from Guan et al. (2023): $\{15, 19, 23, 27, 35\}$ generating 32 molecules per size. We

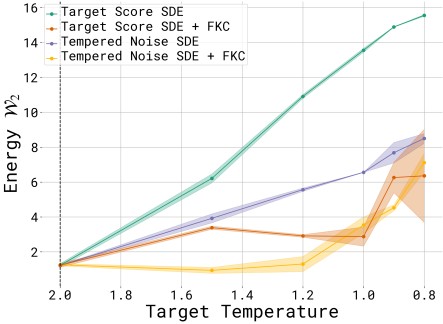

*Figure 4.* 2-Wasserstein between energy distributions of MCMC samples from the annealed target distribution and our methods at different temperatures. Note the training temperature $T_L = 2$.

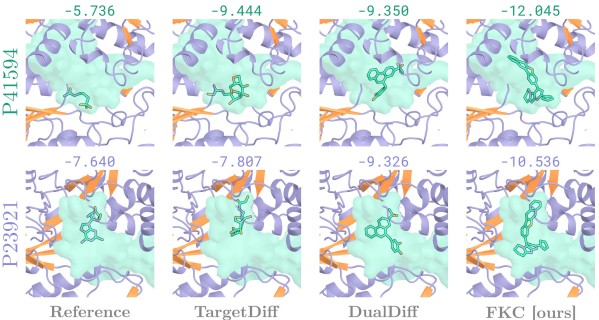

*Figure 5.* Molecules generated from our method (target score SDE with $\beta = 2.0$ and FKC resampling) and baselines in the binding pockets of two proteins: GRM5 (top row, UniProt ID P41594) and RRM1 (bottom row, UniProt ID P23921). Docking scores for each molecule and target are above each image; lower docking scores are better. Here, we display molecules with the best docking scores that have a QED $\geq 0.4$; more generations are in App. F.6. The binding pocket is shaded in light green.

showcase our best results in Table 4 and the full ablation in App. F.6. We evaluate the generated molecules on their docking scores to a protein pair, $\mathbb{P}_1$ and $\mathbb{P}_2$. We report the average of docking score products for each target, as well as the average maximum docking score for a pair. Lower docking scores are better, and so lower maximum docking scores indicate the molecule is better at binding to *both* targets. We compute the percentage of molecules that have better docking scores than known binders, as well as the number of valid and unique molecules generated, their diversity, their drug-likeness (QED (Bickerton et al., 2012)), and their synthetic accessibility (SA (Ertl & Schuffenhauer, 2009)).

We find that the target noise SDE at $\beta > 0.5$ generates molecules with better average docking scores for each of the target proteins compared with both baselines DualDiff (Zhou et al., 2024) and TargetDiff (Guan et al., 2023). When we incorporate FKC, the average docking scores improve further. In Fig. A7, we observe a positive correlation between the FKC weights and docking scores. There is a slight sacrifice in terms of diversity and uniqueness when resampling with FKC, although this is a common trade-off for an increase in quality. Notably, our method achieves

*Table 4.* Docking scores of generated ligands for 100 protein target pairs (P1, P2). We generate 32 ligands for 5 molecule lengths for each protein pair using the Target Score SDE. Lower docking scores are better. Values are reported as averages over all generated molecules in each run. "Better than ref." is the percentage of ligands with better docking scores than known reference molecules for *both* targets (the mean docking score for the reference molecules is $-7.915_{\pm 2.841}$). We also report the diversity, validity & uniqueness, SA score, and QED. [1]TargetDiff from Guan et al. (2023), [2]DualDiff from Zhou et al. (2024).

| | | $(P_1 * P_2)$ (↑) | $\max(P_1, P_2)$ (↓) | $P_1$ (↓) | $P_2$ (↓) | Better than ref. (↑) | Div. (↑) | Val. & Uniq. (↑) | SA (↓) | QED (↑) |
|---|---|---|---|---|---|---|---|---|---|---|
| $P_1$ only[1] | | $59.355_{\pm 30.169}$ | $-6.961_{\pm 2.774}$ | $-8.090_{\pm 1.783}$ | $-7.213_{\pm 2.746}$ | $0.321_{\pm 0.371}$ | $\mathbf{0.886_{\pm 0.013}}$ | $0.918_{\pm 0.107}$ | $\mathbf{0.588_{\pm 0.086}}$ | $0.531_{\pm 0.150}$ |
| $\beta$ | FKC | $(P_1 * P_2)$ (↑) | $\max(P_1, P_2)$ (↓) | $P_1$ (↓) | $P_2$ (↓) | Better than ref. (↑) | Div. (↑) | Val. & Uniq. (↑) | SA (↓) | QED (↑) |
| 0.5 | ✗[2] | $64.554_{\pm 28.225}$ | $-7.030_{\pm 2.556}$ | $-7.950_{\pm 2.212}$ | $-8.028_{\pm 2.154}$ | $0.306_{\pm 0.346}$ | $0.883_{\pm 0.012}$ | $0.943_{\pm 0.124}$ | $0.609_{\pm 0.084}$ | $0.575_{\pm 0.134}$ |
| | ✓ | $66.380_{\pm 35.747}$ | $-6.966_{\pm 3.291}$ | $-8.085_{\pm 2.832}$ | $-8.098_{\pm 2.638}$ | $0.341_{\pm 0.377}$ | $0.870_{\pm 0.021}$ | $0.951_{\pm 0.096}$ | $0.596_{\pm 0.094}$ | $0.587_{\pm 0.129}$ |
| 1.0 | ✗ | $68.851_{\pm 30.153}$ | $-7.256_{\pm 2.622}$ | $-8.206_{\pm 2.385}$ | $-8.287_{\pm 2.123}$ | $0.363_{\pm 0.375}$ | $0.880_{\pm 0.013}$ | $\mathbf{0.964_{\pm 0.100}}$ | $0.611_{\pm 0.090}$ | $0.589_{\pm 0.126}$ |
| | ✓ | $76.036_{\pm 33.835}$ | $-7.649_{\pm 2.605}$ | $-8.658_{\pm 2.347}$ | $-8.660_{\pm 2.349}$ | $0.434_{\pm 0.416}$ | $0.844_{\pm 0.029}$ | $0.939_{\pm 0.106}$ | $0.627_{\pm 0.095}$ | $0.591_{\pm 0.128}$ |
| 2.0 | ✗ | $71.186_{\pm 30.799}$ | $-7.421_{\pm 2.497}$ | $-8.365_{\pm 2.336}$ | $-8.401_{\pm 2.051}$ | $0.383_{\pm 0.389}$ | $0.877_{\pm 0.015}$ | $0.961_{\pm 0.115}$ | $0.642_{\pm 0.086}$ | $\mathbf{0.594_{\pm 0.124}}$ |
| | ✓ | $\mathbf{77.271_{\pm 34.268}}$ | $\mathbf{-7.720_{\pm 2.562}}$ | $\mathbf{-8.682_{\pm 2.488}}$ | $-8.735_{\pm 2.187}$ | $\mathbf{0.450_{\pm 0.438}}$ | $0.806_{\pm 0.048}$ | $0.862_{\pm 0.174}$ | $0.641_{\pm 0.112}$ | $0.592_{\pm 0.146}$ |

the lowest maximum docking score, meaning that generated ligands are able to better bind to both proteins (on average across tasks). Our method also generates the highest fraction of molecules that are better than known binders (reference molecules), which could motivate using our model in *de novo* drug design settings (the mean docking score of reference molecules is $-7.915_{\pm 2.841}$). We visualize ligands for a sample target pair in Fig. 5 and Fig. A6.

In App. F.7, we further investigate the utility of PoE in generating molecule SMILES using a latent diffusion model, and show that FKC resampling improves generation for small molecules satisfying multiple functional properties.

## 6. Related Work

Sequential Monte Carlo methods have proven useful across a wide range of tasks involving diffusion models, including for reward-guided generation (Uehara et al., 2024; 2025; Singhal et al., 2025; Kim et al., 2025; Chen et al., 2025a), conditional generation (Wu et al., 2024), or inverse problems (Dou & Song, 2024; Cardoso et al., 2024).

For compositional generation, Du et al. (2023) learn an energy-based score function and use the energy within MCMC procedures. Thornton et al. (2025) improve training of the energy-based score function by distilling an unconditional score model, where the resulting energy can be used for SMC resampling from annealed or product densities.

Within the context of diffusion samplers from Boltzmann densities, Phillips et al. (2024) consider SMC for energy-based score parameterizations. Chen et al. (2025b); Albergo & Vanden-Eijnden (2024) consider SMC resampling along trajectories with respect to a prescribed geometric annealing path, where Albergo & Vanden-Eijnden (2024) is presented through the Feynman-Kac perspective. The approaches in Vargas et al. (2024); Albergo & Vanden-Eijnden (2024) correspond to the *escorted* Jarynski equality (Vaikuntanathan & Jarzynski, 2008; 2011), where additional transport terms are learned to more closely match the evolution of a given density path (Arbel et al., 2021; Chemseddine et al., 2025; Máté & Fleuret, 2023; Tian et al., 2024; Fan et al., 2024; Maurais & Marzouk, 2024; Vargas et al., 2024).

Indeed, the celebrated Jarzynski equality (Jarzynski, 1997; Crooks, 1999) and its variants admit an elegant proof using the Feynman-Kac formula (Lelièvre et al. (2010, Ch. 4), Vaikuntanathan & Jarzynski (2008)).

Predictor-corrector simulation (Song et al., 2021) performs additional Langevin steps to promote matching the intermediate marginals of $p_t$ of a diffusion model. These schemes can be adapted for annealed or product targets, although Du et al. (2023) found best performance using Metropolis corrections. Bradley & Nakkiran (2024) interpret standard CFG SDE simulation (19) as a predictor-corrector where the corrector targets a different guidance or geometric mixture weight $\beta' = \frac{1}{2}(1 + \beta)$. Our resampling correctors are instead tailored to the original guidance weight $\beta$.

Finally, SMC methods have recently been extended to discrete diffusion models (Singhal et al., 2025; Li et al., 2024; Uehara et al., 2025; Lee et al., 2025a), where the approach of Lee et al. (2025a) is analogous to FKC for discrete settings.

## 7. Conclusion

In this work, we proposed FEYNMAN-KAC CORRECTORS, an array of tools allowing for fine control over the sample distributions of diffusion processes. These target distributions may arise in compositional generative modeling (Du & Kaelbling, 2024), where we seek to combine specialist models capturing various chemical properties of molecules or different aspects of a complex prompt. Geometric averaging appears in widely-used CFG techniques while, via annealing, we demonstrate that an approach of first learning an amortized sampler at a higher temperature and then annealing using FKCs down to a lower temperature opens up a new dimension for the construction of amortized samplers.

Finally, our framework allows for the use of reward models (Prop. D.6) and for a time-dependent annealing schedule $\beta_t$ (Prop. C.6), where the log-density terms needed for weights can be estimated using methods from Skreta et al. (2025).

## Impact Statement

This paper presents work whose goal is to advance the field of Machine Learning. There are many potential societal consequences of our work, none of which we feel must be specifically highlighted here.

## Acknowledgments

This project was partially sponsored by Google through the Google & Mila projects program. The authors acknowledge funding from UNIQUE, CIFAR, NSERC, Intel, and Samsung. The research was enabled in part by computational resources provided by the Digital Research Alliance of Canada (https://alliancecan.ca), Mila (https://mila.quebec), the Acceleration Consortium (https://acceleration.utoronto.ca/), and NVIDIA. KN was supported by IVADO and Institut Courtois. MS thanks Ella Rajaonson for assistance with docking visualizations, as well as Austin Cheng and Cher-Tian Ser for providing feedback on molecule generation.

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

## A. Expectation Estimation under Feynman-Kac PDEs

We proceed in two steps, first finding a Kolmogorov backward equation corresponding to evolution under a weighted Feynman-Kac SDE. We then use this identity to derive the expectation estimator in Eq. (10). Throughout, we consider the evolution of density $p_t$ defined via the following Feynman-Kac PDE,

$$\frac{\partial}{\partial t} p_t(x_t) = -\langle \nabla, p_t(x_t) v_t(x_t) \rangle + \frac{\sigma_t^2}{2} \Delta p_t(x_t) + p_t(x_t) \left( g_t(x_t) - \int g_t(x_t) p_t(x_t) dx_t \right) \tag{34}$$

Our proof follows similar derivations as in Lelièvre et al. (2010, Prop 4.1, Ch. 4.1.4.3) (see also (Vaikuntanathan & Jarzynski, 2008; 2011) and references therein), where the authors are interested in sampling from a sequence of unnormalized distributions $\tilde{p}_t$ specified via a time-varying energy or Hamiltonian. The proofs often rely on Langevin dynamics that leave $p_t$ invariant. We adopt a similar proof technique, but focus directly on simulation with arbitrary $v_t, g_t$ derived via our methods in Sec. 3.

**Proposition A.1.** *For a bounded test function $\phi : \mathcal{X} \to \mathbb{R}$ and $p_t$ satisfying Eq. (34), we have*

$$\mathbb{E}_{p_T(x_T)}[\phi(x_T)] = \frac{1}{Z_T} \mathbb{E}\left[ e^{\int_0^T g_s(x_s) ds} \phi(x_T) \right] \tag{35}$$

$$where \quad dx_t = v_t(x_t) dt + \sigma_t dW_t, \qquad x_0 \sim p_0$$

*where $Z_T$ is a normalization constant independent of x. Eq. (35) which suggests that the self-normalized importance sampling approximation in Eq. (10) is consistent as $K \to \infty$.*

*Proof.* The proof proceeds in three steps, delineated with bold paragraph headers. We first derive the backward Kolmogorov equation for appropriate functions, then specify the evolution of the Feynman-Kac PDE for the unnormalized density, before combining these results to prove the result in Prop. A.1.

**Backward PDE:** For a given test function $\phi(x)$, consider *defining* the following function

$$\Phi_T(x, t) = \mathbb{E}\left[ e^{\int_t^T g_s(x_s) ds} \phi(x_T) \mid x_t = x \right], \qquad \Phi_T(x, T) = \phi(x) \tag{36}$$

where expectations are taken under the evolution of the SDE $dx_t = v_t(x_t) dt + \sigma_t dW_t$.

In particular, for $\tau > t$, we have

$$\Phi_T(x, t) = \mathbb{E}\left[ e^{\int_t^\tau g_s(x_s) ds} e^{\int_\tau^T g_s(x_s) ds} \phi(x_T) \mid x_t = x \right] = \mathbb{E}\left[ e^{\int_t^\tau g_s(x_s) ds} \Phi_T(x_\tau, \tau) \mid x_t = x \right] \tag{37}$$

We will leverage this identity to derive a PDE which $\Phi_T(x, t)$ must satisfy. Note, to link $\Phi_T(x, t)$ and (the expected value of) $\Phi_T(x_\tau, \tau)$, we should account for the weights $e^{\int_t^\tau g_s(x_s) ds}$. Thus, we apply Ito's product rule and Ito's lemma to capture how $e^{\int_t^\tau g_s(x_s) ds} \Phi_T(x_\tau, \tau)$ evolves with $\tau$,

$$d\left( e^{\int_t^\tau g_s(x_s) ds} \Phi_T(x_\tau, \tau) \right) = e^{\int_t^\tau g_s(x_s) ds} d\Phi_T(x_\tau, \tau) + \Phi_T(x_\tau, \tau) de^{\int_t^\tau g_s(x_s) ds} + d\langle \Phi_T(x_\tau, \tau), e^{\int_t^\tau g(x_s) ds} \rangle \tag{38}$$

In the final term, $e^{\int_t^\tau g_s(x_s) ds}$ is non-stochastic and, assuming it has finite variation, the term $d\langle \Phi_T(x, t), e^{\int_t^\tau g_s(x_s) ds} \rangle$ vanishes. We can use Ito's lemma to expand $d\Phi_T(x_\tau, \tau)$ and simple differentiation for $de^{\int_t^\tau g_s(x_s) ds}$,

$$d\left( e^{\int_t^\tau g_s(x_s) ds} \Phi_T(x_\tau, \tau) \right) = e^{\int_t^\tau g_s(x_s) ds} \left( \frac{\partial \Phi_T(x_\tau, \tau)}{\partial \tau} + \langle v_\tau(x_\tau), \nabla \Phi_T(x_\tau, \tau) \rangle + \frac{\sigma_\tau^2}{2} \Delta \Phi_T(x_\tau, \tau) \right) d\tau$$

$$+ e^{\int_t^\tau g_s(x_s) ds} \sigma_t \langle \nabla \Phi_T(x_\tau, \tau), dW_t \rangle + \Phi_T(x_\tau, \tau) e^{\int_t^\tau g_s(x_s) ds} \left( g_\tau(x_\tau) \right) d\tau \tag{39}$$

$$= e^{\int_t^\tau g_s(x_s) ds} \left( \frac{\partial \Phi_T(x_\tau, \tau)}{\partial t} + \langle v_t(x), \nabla \Phi_T(x_\tau, \tau) \rangle + \frac{\sigma_t^2}{2} \Delta \Phi_T(x_\tau, \tau) + \Phi_T(x_\tau, \tau) g_t(x) \right) dt$$

$$+ e^{\int_t^\tau g_s(x_s) ds} \sigma_t \langle \nabla \Phi_T(x_\tau, \tau), dW_t \rangle \tag{40}$$

Integrating Eq. (40) $\tau = t$ to $\tau = T$ and taking expectations under the simulated process from initial point $x_t = x$, the

stochastic term vanishes and we obtain

$$\mathbb{E}\left[e^{\int_t^T g_s(x_s)ds}\Phi_T(x_T,T) \mid x_t = x\right] - \mathbb{E}\left[e^{\int_t^t g_s(x_s)ds}\Phi_T(x,t) \mid x_t = x\right] \tag{41}$$

$$= \mathbb{E}\left[\int_{\tau=t}^T e^{\int_t^\tau g_s(x_s)ds}\left(\frac{\partial\Phi_T(x_\tau,\tau)}{\partial\tau} + \left\langle v_\tau(x), \nabla\Phi_T(x_\tau,\tau)\right\rangle + \frac{\sigma_\tau^2}{2}\Delta\Phi_T(x_\tau,\tau) + \Phi_T(x_\tau,\tau)g_\tau(x)\right)d\tau\right]$$

Finally, we simplify the first line in Eq. (41). Considering the definition and endpoint condition in Eq. (36), we have

$$\mathbb{E}\left[e^{\int_t^T g_s(x_s)ds}\Phi_T(x_T,T) \mid x_t = x\right] - \mathbb{E}\left[e^{\int_t^t g_s(x_s)ds}\Phi_T(x,t) \mid x_t = x\right]$$

$$= \mathbb{E}\left[e^{\int_t^T g_s(x_s)ds}\phi(x_T) \mid x_t = x\right] - \Phi_T(x,t) = 0 \tag{42}$$

by definition in Eq. (36). Since $e^{\int_t^\tau g_s(x_s)ds} > 0$, this implies that the integrand in the second line of Eq. (41) should be zero for any $\tau$. Thus, we obtain a backward PDE which is often used directly in the statement of the Feynman-Kac formula,

$$\frac{\partial\Phi_T(x_\tau,\tau)}{\partial\tau} + \left\langle v_\tau(x), \nabla\Phi_T(x_\tau,\tau)\right\rangle + \frac{\sigma_\tau^2}{2}\Delta\Phi_T(x_\tau,\tau) + \Phi_T(x_\tau,\tau)g_\tau(x) = 0 \tag{43}$$

**Evolution of Unnormalized Density**   In practice, we cannot exactly calculate $\int g_t(x_t)p_t(x_t)dx_t$, which appears in the reweighting equation in Eq. (6) (or Eq. (46) below) to ensure normalization. Eventually, we will account for normalization using SNIS as in Eq. (10).

For now, consider the evolution of unnormalized density $\tilde{p}_t(x) = p_t(x)Z_t$ for a particular $v_t, \sigma_t, g_t$ and some normalization constant $Z_t$. With foresight, we define

$$\frac{\partial}{\partial t}\tilde{p}_t(x_t) = -\left\langle\nabla, \tilde{p}_t(x_t)v_t(x_t)\right\rangle + \frac{\sigma_t^2}{2}\Delta\tilde{p}_t(x_t) + \tilde{p}_t(x_t)g_t(x_t) \tag{44}$$

which we justify by noting that only the reweighting term does not preserve normalization. In particular, let

$$\partial_t \log Z_t := \int p_t(x)g_t(x)dx. \tag{45}$$

which seems to be a natural candidate from inspecting a general, reweighting-only evolution $\partial_t p_t^w(x) = p_t^w(x)\left(g_t(x) - \int p_t^w(x)g_t(x)dx\right)$, which implies $\partial_t \log p_t^w(x) = g_t(x) - \int p_t^w(x)g_t(x)dx$. Defining terms such that $\partial_t \log p_t^w(x) = \partial_t \log \tilde{p}_t^w(x) - \partial_t \log Z_t$ yields Eq. (45). We finally confirm that the definitions in Eq. (44) and Eq. (45) are consistent with the original Feynman-Kac PDE,

$$\frac{\partial}{\partial t}p_t(x_t) = -\left\langle\nabla, p_t(x_t)v_t(x_t)\right\rangle + \frac{\sigma_t^2}{2}\Delta p_t(x_t) + p_t(x_t)\left(g_t(x_t) - \int g_t(x_t)p_t(x_t)dx_t\right) \tag{46}$$

Namely, since $p_t(x_t) = \tilde{p}_t(x_t)Z_t^{-1}$, the definitions in Eq. (44)-(45) should satisfy

$$\frac{\partial}{\partial t}p_t(x_t) = \frac{\partial}{\partial t}\left(\tilde{p}_t(x_t)Z_t^{-1}\right) \tag{47a}$$

$$= Z_t^{-1}\frac{\partial}{\partial t}\tilde{p}_t(x_t) + \tilde{p}_t(x_t)Z_t^{-1}\,\partial_t\log(Z_t^{-1}) \tag{47b}$$

$$= Z_t^{-1}\frac{\partial}{\partial t}\tilde{p}_t(x_t) - \tilde{p}_t(x_t)Z_t^{-1}\,\partial_t\log Z_t \tag{47c}$$

$$= Z_t^{-1}\left(-\left\langle\nabla, \tilde{p}_t(x_t)v_t(x_t)\right\rangle + \frac{\sigma_t^2}{2}\Delta\tilde{p}_t(x_t) + \tilde{p}_t(x_t)g_t(x_t)\right) - \tilde{p}_t(x_t)Z_t^{-1}\int p_t(x_t)g_t(x_t)dx \tag{47d}$$

Noting that $\nabla_{x_t}Z_t = 0$, we can pull $Z_t^{-1}$ inside differential operators to obtain

$$= -\left\langle\nabla, \frac{\tilde{p}_t(x_t)}{Z_t}v_t(x_t)\right\rangle + \frac{\sigma_t^2}{2}\Delta\frac{\tilde{p}_t(x_t)}{Z_t} + \frac{\tilde{p}_t(x_t)}{Z_t}g_t(x_t) - \frac{\tilde{p}_t(x_t)}{Z_t}\int p_t(x_t)g_t(x_t)dx \tag{47e}$$

$$= -\left\langle\nabla, p_t(x_t)v_t(x_t)\right\rangle + \frac{\sigma_t^2}{2}\Delta p_t(x_t) + p_t(x_t)\left(g_t(x_t) - \int p_t(x_t)g_t(x_t)dx\right) \tag{47f}$$

as desired.

**Expectation Estimation:** Now, we use Eq. (43) to write the total derivative of the following integral under the unnormalized density $\tilde{p}_t(x)$,

$$\frac{d}{dt}\left(\int \Phi_T(x,t)\tilde{p}_t(x)dx\right) = \int\left(\frac{\partial \Phi_T(x,t)}{\partial t}\right)\tilde{p}_t(x)dx + \int \Phi_T(x,t)\left(\frac{\partial \tilde{p}_t(x)}{\partial t}\right)dx \tag{48a}$$

Using Eq. (43) and Eq. (44), we have

$$= \int\left(-\langle v_t(x), \nabla\Phi_T(x,t)\rangle - \frac{\sigma_\tau^2}{2}\Delta\Phi_T(x,t) - \Phi_T(x,t)g_\tau(x)\right)\tilde{p}_t(x)dx \tag{48b}$$

$$+ \int \Phi_T(x,t)\left(-\langle\nabla, \tilde{p}_t(x)v_t(x)\rangle + \frac{\sigma_t^2}{2}\Delta\tilde{p}_t(x) + \tilde{p}_t(x)g_t(x)\right)dx$$

Integrating by parts in the second line, we have

$$= \int\left(-\langle v_t(x), \nabla\Phi_T(x,t)\rangle - \frac{\sigma_\tau^2}{2}\Delta\Phi_T(x,t) - \Phi_T(x,t)g_\tau(x)\right)\tilde{p}_t(x)dx \tag{48c}$$

$$+ \int\left(\langle v_t(x), \nabla\Phi_T(x,t)\rangle + \frac{\sigma_t^2}{2}\Delta\Phi_T(x,t) + \Phi_T(x,t)g_t(x)\right)\tilde{p}_t(x)dx$$

$$= 0 \tag{48d}$$

Integrating on the interval $t = 0$ to $t = T$, we obtain

$$\int \Phi_T(x_T, T)\tilde{p}_T(x_T)dx_T - \int \Phi_T(x_0, 0)\tilde{p}_0(x_0)dx_0 = \int_0^T \frac{d}{dt}\left(\int \Phi_T(x,t)\tilde{p}_t(x)dx\right)dt = 0 \tag{49}$$

Thus, we can set these two quantities equal to each other. Using the identity $\tilde{p}_t(x) = p_t(x)Z_t$ and assuming we initialize simulation with normalized $p_0(x) = \tilde{p}_0(x)$ with $Z_0 = 1$, we can finally use the definitions in Eq. (36) (namely $\Phi_t(x_T, T) = \phi(x_T)$) to write

$$\int \Phi_T(x_0, 0)\tilde{p}_0(x_0)dx_0 = \int \Phi_T(x_T, T)\tilde{p}_T(x_T)dx_T \tag{50}$$

$$Z_0 \int\left(\mathbb{E}[e^{\int_0^T g_s(x_s)ds}\phi(x_T) \mid x_0]\right)p_0(x_0)dx_0 = Z_T \int \phi(x_T)p_T(x_T)dx_T \tag{51}$$

$$\frac{1}{Z_T}\mathbb{E}\left[e^{\int_0^T g_s(x_s)ds}\phi(x_T)\right] = \mathbb{E}_{p_T(x_T)}[\phi(x_T)] \tag{52}$$

which is the desired identity. In practice, we could estimate $Z_T \approx \frac{1}{K}\sum_{k=1}^K e^{\int_0^T g_s(x_s^{(k)})ds} = \frac{1}{K}\sum_{k=1}^K e^{w_T^{(k)}}$ and $\mathbb{E}[e^{\int_0^T g_s(x_s)ds}\phi(x_T)] \approx \frac{1}{K}\sum_{k=1}^K e^{w_T^{(k)}}\phi(x_T^{(k)})$, which yields Eq. (10). $\square$

Note that our choice of upper limit $T$ in $\Phi_T$ was arbitrary, suggesting that we could repurpose the same reasoning for estimating expectations at intermediate $t$ from initialization at time 0. This suggests that our samples are properly weighted for estimating expectations and normalization constants $Z_t$ for intermediate $p_t$ (Naesseth et al., 2019).

Similarly, changing the lower limit of integration from $t = 0$ to intermediate $t$, the analogue of Eq. (51) suggests estimating expectations using $Z_t\mathbb{E}_{p_t}[\Phi_T(x_t, t)] = Z_T\mathbb{E}_{p_T}[\phi(x_T)]$. Given properly-weighted particle approximations of $p_t, Z_t$, we can continue calculating the appropriate weights along the trajectory to estimate $Z_T$ or terminal expectations under $p_T$. These arguments can be similarly adapted to justify SMC resampling at intermediate steps, as we do in practice (Sec. 4).

## B. Feynman-Kac Processes

### B.1. Markov Generators for Feynman-Kac Processes

In Sec. 2, we described the adjoint generators $\mathcal{L}_t^{*(v)}[p_t], \mathcal{L}_t^{*(\sigma)}[p_t], \mathcal{L}_t^{*(g)}[p_t]$ corresponding to flows with vector field $v_t$, diffusions with coefficient $\sigma_t$, and reweighting with respect to $g_t$. In particular, the Kolmogorov forward equation $\frac{\partial p_t}{\partial t}(x) = \mathcal{L}_t^*[p_t](x)$ corresponds to our PDEs presented in Eqs. (3), (5) and (6). In the lemma below, we recall the generators which are adjoint to those in Sec. 2 and operate over smooth, bounded test functions with compact support, e.g. $\mathcal{L}_t^{(v)}[\phi]$.

**Lemma B.1** (Adjoint Generators). *Using the identity $\int \phi(x)\,\mathcal{L}_t^*[p_t](x)\,dx = \int \mathcal{L}_t[\phi](x)\,p_t(x)\,dx$*

$$
\textbf{\textit{Flow:}} \qquad \mathcal{L}_t^{(v)}[\phi](x) = \langle \nabla\phi(x), v_t(x)\rangle \qquad\qquad\qquad \mathcal{L}_t^{*(v)}[p_t](x) = -\langle \nabla, p_t(x)\,v_t(x)\rangle
$$

$$
\textbf{\textit{Diffusion:}} \qquad \mathcal{L}_t^{(\sigma)}[\phi](x) = \frac{\sigma_t^2}{2}\Delta\phi(x) \qquad\qquad\qquad \mathcal{L}_t^{*(\sigma)}[p_t](x) = \frac{\sigma_t^2}{2}p_t(x) \tag{53}
$$

$$
\textbf{\textit{Reweighting:}} \qquad \mathcal{L}_t^{(g,p)}[\phi](x) = \phi_t(x)\Big(g_t(x) - \int g_t(x)\,p_t(x)\,dx\Big) \qquad \mathcal{L}_t^{*(g)}[p_t](x) = p_t(x)\Big(g_t(x) - \int g_t(x)\,p_t(x)dx\Big)
$$

*Proof.* The proofs for flows and diffusions follow using integration by parts, with proofs found in, for example Holderrieth et al. (2025, Sec. A.5). For the reweighting generator, we have

$$
\int \phi(x)\mathcal{L}_t^{*(g)}[p_t](x)dx = \int \phi(x)\Big(p_t(x)\Big(g_t(x) - \int g_t(y)\,p_t(y)dy\Big)\Big)dx
$$

$$
= \int p_t(x)\Big(\phi(x)\Big(g_t(x) - \int g_t(y)\,p_t(y)dy\Big)\Big)dx
$$

$$
=: \int p_t(x)\,\mathcal{L}_t^{(g,p)}[\phi](x)\,dx
$$

Note that the weights $g_t$ are often chosen in relation to the unnormalized density of $p_t$ (Lelièvre et al. (2010, Sec. 4)), and our attention will be focused on the pair of generator actions $\mathcal{L}_t^{*(g)}[p_t], \mathcal{L}_t^{(g,p)}[\phi]$ for possibly time-dependent $\phi$. $\qquad\square$

## B.2. Jump Process Interpretation of Reweighting

One way to perform simulation of the reweighting equation will be to rewrite it in terms of a jump process. We first recall the definition of the Markov generator of a jump process (Ethier & Kurtz (2009, 4.2), Del Moral (2013, 1.1), Holderrieth et al. (2025, A.5.3)) and derive its adjoint generator.

**Lemma B.2** (Jump Process Generators). *Using the definition of the jump process generator and the identity $\int \phi(x)\,\mathcal{J}_t^*[p_t](x)\,dx = \int \mathcal{J}_t[\phi](x)\,p_t(x)\,dx$. Letting $W_t(x,y) = \lambda_t(x)J_t(y|x)$ for normalized $J_t(y|x)$,*

$$
\textbf{\textit{Jump Process:}} \qquad \mathcal{J}_t^{(W)}[\phi](x) := \int \Big(\phi(y) - \phi(x)\Big)\lambda_t(x)J_t(y|x)dy \tag{54a}
$$

$$
\mathcal{J}_t^{*(W)}[p_t](x) = \Big(\int \lambda_t(y)J_t(x|y)p_t(y)dy\Big) - p_t(x)\lambda_t(x) \tag{54b}
$$

*Proof.* Through simple manipulations and changing the variables of integration, we obtain

$$
\int \phi(x)\,\mathcal{J}_t^*[p_t](x)\,dx = \int \mathcal{J}_t[\phi](x)\,p_t(x)\,dx
$$

$$
= \int\Big(\int \Big(\phi(y) - \phi(x)\Big)\lambda_t(x)J_t(y|x)dy\Big)p_t(x)\,dx
$$

$$
= \int\int \phi(y)\lambda_t(x)J_t(y|x)p_t(x)\,dydx - \int\int \phi(x)\lambda_t(x)J_t(y|x)p_t(x)\,dydx
$$

$$
= \int\int \phi(x)\lambda_t(y)J_t(x|y)p_t(y)\,dxdy - \int\int \phi(x)\lambda_t(x)J_t(y|x)p_t(x)\,dydx
$$

$$
= \int \phi(x)\Big(\Big(\int \lambda_t(y)J_t(x|y)p_t(y)dy\Big) - p_t(x)\lambda_t(x)\Big(\int J_t(y|x)dy\Big)\Big)dx
$$

$$
\implies \quad \mathcal{J}_t^*[p_t](x) = \Big(\int \lambda_t(y)J_t(x|y)p_t(y)dy\Big) - p_t(x)\lambda_t(x)
$$

using the assumption that $J_t(y|x)$ is normalized. $\qquad\square$

**Reweighting $\to$ Jump Process**   Our goal is to derive a jump process such that the adjoint generators are equivalent $\mathcal{J}_t^{*(W)}[p_t](x) = \mathcal{L}_t^{*(g)}[p_t](x)$ for a given reweighting generator with weights $g_t$ (Eq. (53)).

While Del Moral (2013); Angeli (2020) emphasize the freedom of choice in such generators,[1] Sec. 4 of (Angeli et al., 2019)

---

[1] For example, see Rousset (2006); Rousset & Stoltz (2006) for a particular instantiation combining separate birth and death processes.

argues for a particular choice to reduce the expected number of resampling events. To define this process, consider the following thresholding operations,

$$(u)^- := \max(0, -u) \qquad (u)^+ := \max(0, u), \qquad \text{which satisfy:} \quad (u)^+ - (u)^- = u. \tag{55}$$

We can now define the Markov generator using

$$W_t(x, y) = \lambda_t(x) J_t(y|x) \qquad \lambda_t(x) := \left(g_t(x) - \mathbb{E}_{p_t}[g_t]\right)^- \qquad J_t(y|x) := \frac{\left(g_t(y) - \mathbb{E}_{p_t}[g_t]\right)^+ p_t(y)}{\int (g_t(z) - \mathbb{E}_{p_t}[g_t])^+ p_t(z) dz} \tag{56}$$

Since jump events are triggered based on $\lambda_t(x_t) = (g_t(x) - \mathbb{E}_{p_t}[g_t])^-$ and are more likely to transition to events with high excess weight $(g_t(y) - \mathbb{E}_{p_t}[g_t])^+ p_t(y)$, we expect this process to improve the sample population in efficient fashion (Angeli et al., 2019).

---

**Proposition B.3.** *For a given weighting function $g_t$ and the adjoint generator $\mathcal{L}_t^{*(g)}$, the adjoint generator $\mathcal{J}_t^{*(W)}$ derived using in Eq. (56) satisfies $\mathcal{J}_t^{*(W)}[p_t](x) = \mathcal{L}_t^{*(g)}[p_t](x)$. More explicitly, we have*

$$\mathcal{L}_t^{*(g)}[p_t](x) = \mathcal{J}_t^{*(W)}[p_t](x) \tag{57}$$

$$p_t(x)\left(g_t(x) - \int g_t(x)\, p_t(x) dx\right) = \left(\int \left(g_t(y) - \mathbb{E}_{p_t}[g_t]\right)^- \frac{(g_t(x) - \mathbb{E}_{p_t}[g_t])^+ p_t(x)}{\int (g_t(z) - \mathbb{E}_{p_t}[g_t])^+ p_t(z) dz} p_t(y) dy\right) - p_t(x)\left(g_t(x) - \mathbb{E}_{p_t}[g_t]\right)^-.$$

---

*Proof.* We start by expanding the definition of $\mathcal{J}_t^{*(W)}[p_t](x)$

$$\mathcal{J}_t^{*(W)}[p_t](x) = \left(\int \lambda_t(y) J_t(x|y) p_t(y) dy\right) - p_t(x)\lambda_t(x) \tag{58a}$$

$$= \left(\int \left(g_t(y) - \mathbb{E}_{p_t}[g_t]\right)^- \frac{(g_t(x) - \mathbb{E}_{p_t}[g_t])^+ p_t(x)}{\int (g_t(z) - \mathbb{E}_{p_t}[g_t])^+ p_t(z) dz} p_t(y) dy\right) - p_t(x)\left(g_t(x) - \mathbb{E}_{p_t}[g_t]\right)^- \tag{58b}$$

$$= \left(\int \left(g_t(y) - \mathbb{E}_{p_t}[g_t]\right)^- p_t(y) dy\right)\left(\frac{(g_t(x) - \mathbb{E}_{p_t}[g_t])^+ p_t(x)}{\int (g_t(z) - \mathbb{E}_{p_t}[g_t])^+ p_t(z) dz}\right) - p_t(x)\left(g_t(x) - \mathbb{E}_{p_t}[g_t]\right)^- \tag{58c}$$

$$= \left(\frac{\int (g_t(y) - \mathbb{E}_{p_t}[g_t])^- p_t(y) dy}{\int (g_t(z) - \mathbb{E}_{p_t}[g_t])^+ p_t(z) dz}\right) p_t(x)\left(g_t(x) - \mathbb{E}_{p_t}[g_t]\right)^+ - p_t(x)\left(g_t(x) - \mathbb{E}_{p_t}[g_t]\right)^- \tag{58d}$$

Using Eq. (55), note that

$$\int \left(g_t(z) - \mathbb{E}_{p_t}[g_t]\right)^+ p_t(z) dz - \int dp_t(z)\left(g_t(z) - \mathbb{E}_{p_t}[g_t]\right)^- = \int (g_t(z) - \mathbb{E}_{p_t}[g_t]) p_t(z) dz = 0 \tag{59}$$

which implies $\int (g_t(z) - \mathbb{E}_{p_t}[g_t])^+ p_t(z) dz = \int (g_t(z) - \mathbb{E}_{p_t}[g_t])^- p_t(z) dz$. We proceed in two cases, handling separately the trivial case where the denominator in Eq. (58d) is zero.

*Case 1 ($\lambda_t(x) = 0 \ \forall z \in supp(p_t)$):*   Note that $\int \left(g_t(z) - \mathbb{E}_{p_t}[g_t]\right)^- p_t(z) dz = 0$ if and only if $g_t(z) = \mathbb{E}_{p_t}[g_t], \ \forall z$, since $(u)^- \geq 0$. In this case, the generators become trivial and we can confirm

$$\mathcal{L}_t^{*(g)}[p_t](x) = p_t(x)\left(g_t(x) - \int g_t(x)\, p_t(x) dx\right) = p_t(x)(\mathbb{E}_{p_t}[g_t] - \mathbb{E}_{p_t}[g_t]) = 0$$

$$\mathcal{J}_t^{*(W)}[p_t](x) = \int 0 \cdot 0\, p_t(y) dy - p_t(x) \cdot 0 = 0 \tag{60}$$

and thus Eq. (57) holds, as desired.

*Case 2 ($\exists x \in supp(p_t)$ s.t. $\lambda_t(x) > 0$):*   Under the assumption, $\exists x \in supp(\mu_t)$ s.t. $\left(g_t(x) - \mathbb{E}_{p_t}[g_t]\right)^- > 0$. This implies $\int \left(g_t(z) - \mathbb{E}_{p_t}[g_t]\right)^- p_t(z) dz = \int \left(g_t(z) - \mathbb{E}_{p_t}[g_t]\right)^+ p_t(z) dz > 0$.

In this case, we can conclude using Eq. (59) that $\frac{\int dp_t(z)\left(g_t(z) - \mathbb{E}_{p_t}[g_t]\right)^-}{\int dp_t(z)\left(g_t(z) - \mathbb{E}_{p_t}[g_t]\right)^+} = 1$.

Continuing from Eq. (58d)

$$\mathcal{J}_t^{*(W)}[p_t](x) = \left( \frac{\int (g_t(y) - \mathbb{E}_{p_t}[g_t])^- p_t(y)dy}{\int (g_t(z) - \mathbb{E}_{p_t}[g_t])^+ p_t(z)dz} \right) p_t(x) \Big( g_t(x) - \mathbb{E}_{p_t}[g_t] \Big)^+ - p_t(x) \Big( g_t(x) - \mathbb{E}_{p_t}[g_t] \Big)^- \tag{61a}$$

$$= p_t(x) \left( \Big( g_t(x) - \mathbb{E}_{p_t}[g_t] \Big)^+ - \Big( g_t(x) - \mathbb{E}_{p_t}[g_t] \Big)^- \right) \tag{61b}$$

$$= p_t(x)(g_t(x) - \mathbb{E}_{p_t}[g_t]) \tag{61c}$$

$$= \mathcal{L}_t^{*(g)}[p_t](x) \tag{61d}$$

as desired. Note that, in the second to last line, we used the identity in Eq. (55) that $(u)^+ - (u)^- = u$. $\qquad\square$

### B.3. Simulation Schemes

In practice, we use an empirical mean over $K$ particles with as an approximation to the expectation $\mathbb{E}_{p_t}[g_t]$, with

$$\Big( g_t(x^{(k)}) - \mathbb{E}_{p_t}[g_t] \Big)^- \approx \Big( g_t(x^{(k)}) - \frac{1}{K} \sum_{i=1}^K g_t(x^{(i)}) \Big)^-, \quad \Big( g_t(x^{(k)}) - \mathbb{E}_{p_t}[g_t] \Big)^+ \approx \Big( g_t(x^{(k)}) - \frac{1}{K} \sum_{i=1}^K g_t(x^{(i)}) \Big)^+$$

See Del Moral (2013, Sec. 5.4) for discussion.

**Discretization of the Continuous-Time Jump Process** To simulate a jump process with generator $\mathcal{J}_t^{(J,p)}[\phi]$, we can consider the following infinitesimal sampling procedure (Gardiner (2009, Ch. 12); Davis (1984); Holderrieth et al. (2025)). With rate $\lambda_t(x) = \Big( g_t(x) - \mathbb{E}_{p_t}[g_t] \Big)^-$, the particle jumps to a new configuration,

$$x_{t+dt} = \begin{cases} x_t & \text{with probability } 1 - dt \cdot \lambda_t(x_t) + o(dt) \\ y_{t+dt} \sim \text{Categorical} \left\{ \frac{\Big( g_t(x^{(k)}) - \frac{1}{K} \sum_{i=1}^K g_t(x^{(i)}) \Big)^+}{\sum_{j=1}^K \Big( g_t(x^{(j)}) - \frac{1}{K} \sum_{i=1}^K g_t(x^{(i)}) \Big)^+} \right\}_{k=1}^K & \text{with probability } dt \cdot \lambda_t(x_t) + o(dt) \end{cases}$$

The new configuration is sampled according to an empirical approximation of $J_t(y|x)$ using $p_t^K(y) = \frac{1}{K} \sum_{k=1}^K \delta_y(x^{(k)})$, where the outer $\frac{1}{K}$ factor cancels.

Note that the jump rate is zero for particles with $g_t(x) \geq \mathbb{E}_{p_t}[g_t]$. Resampling a new particle proportional to $(g_t(x^{(k)}) - \frac{1}{K} \sum_j g_t(x^{(j)}))^+$ thus promotes the replacement of low importance-weight samples with more promising samples.

**Interacting Particle System** Following Del Moral (2013, Sec 5.4), the process may also be simulated using 'exponential clocks'. In particular, we sample an exponential random variable with rate 1, $\tau^{(k)} \sim \text{exponential}(1)$ as the time when the next jump event will occur (see Gardiner (2009, Ch. 12)). We record artificial time by accumulating the rate function $\lambda_{t_{\text{last}}:s} = \sum_{t=t_{\text{last}}}^s \lambda_t(x_t)dt$ for samples $x_t$ along our simulated diffusion. Upon exceeding the threshold time $\lambda_{t_{\text{last}}:s}^{(k)} \geq \tau^{(k)}$, we sample a transition according the empirical approximaton of $J_t(y|x)$ in Eq. (62). We report results using this scheme in App. F.2 Table A1, but found it to underperform relative to systematic resampling in these initial experiments.

## C. Proofs for Table 1

### C.1. Annealing

**Proposition C.1** (Annealed Continuity Equation). *Consider the marginals generated by the continuity equation*

$$\frac{\partial q_t(x)}{\partial t} = -\big\langle \nabla, q_t(x)v_t(x) \big\rangle. \tag{62}$$

*The marginals* $p_{t,\beta}(x) \propto q_t^\beta(x)$ *satisfy the following PDE*

$$\frac{\partial}{\partial t} p_{t,\beta}(x) = - \big\langle \nabla, p_{t,\beta}(x)v_t(x) \big\rangle + p_{t,\beta}(x)\big[ g_t(x) - \mathbb{E}_{p_{t,\beta}} g_t(x) \big], \tag{63}$$

$$g_t(x) = - (\beta - 1)\big\langle \nabla, v_t(x) \big\rangle. \tag{64}$$

*Proof.* We want to find the partial derivative of the annealed density

$$p_{t,\beta}(x) = \frac{q_t(x)^\beta}{\int dx\, q_t(x)^\beta}\,, \quad \frac{\partial}{\partial t}p_{t,\beta}(x) = ? \tag{65}$$

By the straightforward calculations we have

$$\frac{\partial}{\partial t}\log p_{t,\beta} = \beta\frac{\partial}{\partial t}\log q_t - \int dx\, p_{t,\beta}\beta\frac{\partial}{\partial t}\log q_t \tag{66}$$

$$= -\beta\langle\nabla, v_t\rangle - \beta\langle\nabla\log q_t, v_t\rangle - \int dx\, p_{t,\beta}\big[-\beta\langle\nabla, v_t\rangle - \beta\langle\nabla\log q_t, v_t\rangle\big] \tag{67}$$

$$= -\langle\nabla, v_t\rangle - \langle\nabla\log p_{t,\beta}, v_t\rangle + (1-\beta)\langle\nabla, v_t\rangle - \int dx\, p_{t,\beta}\big[-\beta\langle\nabla, v_t\rangle - \langle\nabla\log p_{t,\beta}, v_t\rangle\big] \tag{68}$$

$$= -\langle\nabla, v_t\rangle - \langle\nabla\log p_{t,\beta}, v_t\rangle + (1-\beta)\langle\nabla, v_t\rangle - \int dx\, p_{t,\beta}\big[(1-\beta)\langle\nabla, v_t\rangle\big]\,. \tag{69}$$

Thus, we have

$$\frac{\partial}{\partial t}p_{t,\beta}(x) = -\langle\nabla, p_{t,\beta}(x)v_t(x)\rangle + p_{t,\beta}(x)\big[(1-\beta)\langle\nabla, v_t(x)\rangle - \mathbb{E}_{p_{t,\beta}}(1-\beta)\langle\nabla, v_t(x)\rangle\big]\,, \tag{70}$$

which can be simulated as

$$dx_t = v_t(x_t)dt\,, \tag{71}$$

$$dw_t = -(\beta-1)\langle\nabla, v_t(x_t)\rangle dt\,. \tag{72}$$

$\square$

---

**Proposition C.2** (Scaled Annealed Continuity Equation). *Consider the marginals generated by the continuity equation*

$$\frac{\partial q_t(x)}{\partial t} = -\langle\nabla, q_t(x)v_t(x)\rangle\,. \tag{73}$$

*The marginals $p_{t,\beta}(x) \propto q_t^\beta(x)$ satisfy the following PDE*

$$\frac{\partial}{\partial t}p_{t,\beta}(x) = -\langle\nabla, p_{t,\beta}(x)\beta v_t(x)\rangle + p_{t,\beta}(x)\big[g_t(x) - \mathbb{E}_{p_{t,\beta}}g_t(x)\big]\,, \tag{74}$$

$$g_t(x) = (\beta-1)\langle\nabla\log p_{t,\beta}(x), v_t(x)\rangle\,. \tag{75}$$

---

*Proof.* We want to find the partial derivative of the annealed density

$$p_{t,\beta}(x) = \frac{q_t(x)^\beta}{\int dx\, q_t(x)^\beta}\,, \quad \frac{\partial}{\partial t}p_{t,\beta}(x) = ? \tag{76}$$

By the straightforward calculations we have

$$\frac{\partial}{\partial t}\log p_{t,\beta} = \beta\frac{\partial}{\partial t}\log q_t - \int dx\, p_{t,\beta}\beta\frac{\partial}{\partial t}\log q_t \tag{77}$$

$$= -\beta\langle\nabla, v_t\rangle - \beta\langle\nabla\log q_t, v_t\rangle - \int dx\, p_{t,\beta}\big[-\beta\langle\nabla, v_t\rangle - \beta\langle\nabla\log q_t, v_t\rangle\big] \tag{78}$$

$$= -\langle\nabla, \beta v_t\rangle - \langle\nabla\log p_{t,\beta}, v_t\rangle - \int dx\, p_{t,\beta}\big[-\beta\langle\nabla, v_t\rangle - \langle\nabla\log p_{t,\beta}, v_t\rangle\big] \tag{79}$$

$$= -\langle\nabla, \beta v_t\rangle - \langle\nabla\log p_{t,\beta}, \beta v_t\rangle - (1-\beta)\langle\nabla\log p_{t,\beta}, v_t\rangle - \int dx\, p_{t,\beta}\big[-(1-\beta)\langle\nabla\log p_{t,\beta}, v_t\rangle\big]\,. \tag{80}$$

Thus, we have

$$\frac{\partial}{\partial t}p_{t,\beta}(x) = -\langle\nabla, p_{t,\beta}(x)\beta v_t(x)\rangle + p_{t,\beta}(x)\big[g_t(x) - \mathbb{E}_{p_{t,\beta}}g_t(x)\big]\,, \tag{81}$$

$$g_t(x) = -(1-\beta)\langle\nabla\log p_{t,\beta}, v_t\rangle\,, \tag{82}$$

which can be simulated as

$$dx_t = \beta v_t(x_t)dt \,, \tag{83}$$

$$dw_t = \beta(\beta - 1)\langle \nabla \log q_t(x_t), v_t(x_t)\rangle dt \,. \tag{84}$$

$\square$

**Proposition C.3** (Annealed Diffusion Equation). *Consider the marginals generated by the diffusion equation*

$$\frac{\partial q_t(x)}{\partial t} = \frac{\sigma_t^2}{2} \Delta q_t(x) \,. \tag{85}$$

*Then the marginals $p_{t,\beta}(x) \propto q_t^\beta(x)$ satisfy the following PDE*

$$\frac{\partial}{\partial t}p_{t,\beta}(x) = \frac{\sigma_t^2}{2}\Delta p_{t,\beta}(x) + p_{t,\beta}(x)\big[g_t(x) - \mathbb{E}_{p_{t,\beta}}g_t(x)\big] \,, \tag{86}$$

$$g_t(x) = -\beta(\beta - 1)\frac{\sigma_t^2}{2}\|\nabla \log q_t(x)\|^2 \,. \tag{87}$$

*Proof.* We want to find the partial derivative of the annealed density

$$p_{t,\beta}(x) = \frac{q_t(x)^\beta}{\int dx \, q_t(x)^\beta} \,, \quad \frac{\partial}{\partial t}p_{t,\beta}(x) =? \tag{88}$$

By the straightforward calculations we have

$$\frac{\partial}{\partial t}\log p_{t,\beta} = \beta\frac{\partial}{\partial t}\log q_t - \int dx \, p_{t,\beta}\beta\frac{\partial}{\partial t}\log q_t \tag{89}$$

$$= \beta\frac{\sigma_t^2}{2}\Delta \log q_t + \beta\frac{\sigma_t^2}{2}\|\nabla \log q_t\|^2 - \int dx \, p_{t,\beta}\left[\beta\frac{\sigma_t^2}{2}\Delta \log q_t + \beta\frac{\sigma_t^2}{2}\|\nabla \log q_t\|\right] \tag{90}$$

$$= \frac{\sigma_t^2}{2}\Delta \log p_{t,\beta} + \frac{\sigma_t^2}{2\beta}\|\nabla \log p_{t,\beta}\|^2 - \int dx \, p_{t,\beta}\left[\frac{\sigma_t^2}{2}\Delta \log p_{t,\beta} + \frac{\sigma_t^2}{2\beta}\|\nabla \log p_{t,\beta}\|^2\right] \tag{91}$$

$$= \frac{\sigma_t^2}{2}\Delta \log p_{t,\beta} + \frac{\sigma_t^2}{2}\|\nabla \log p_{t,\beta}\|^2 - \left(1 - \frac{1}{\beta}\right)\frac{\sigma_t^2}{2}\|\nabla \log p_{t,\beta}\|^2 \tag{92}$$

$$- \int dx \, p_{t,\beta}\left[-\left(1 - \frac{1}{\beta}\right)\frac{\sigma_t^2}{2}\|\nabla \log p_{t,\beta}\|^2\right] \,. \tag{93}$$

Thus, we have

$$\frac{\partial}{\partial t}p_{t,\beta}(x) = \frac{\sigma_t^2}{2}\Delta p_{t,\beta}(x) + p_{t,\beta}(x)\big[g_t(x) - \mathbb{E}_{p_{t,\beta}}g_t(x)\big] \,, \tag{94}$$

$$g_t(x) = -\beta(\beta - 1)\frac{\sigma_t^2}{2}\|\nabla \log q_t(x)\|^2 \,, \tag{95}$$

which can be simulated (for $\beta > 0$) as

$$dx_t = \sigma_t dW_t \,, \tag{96}$$

$$dw_t = -\beta(\beta - 1)\frac{\sigma_t^2}{2}\|\nabla \log q_t(x_t)\|^2 dt \,. \tag{97}$$

$\square$

**Proposition C.4** (Scaled Annealed Diffusion Equation). *Consider the marginals generated by the diffusion equation*

$$\frac{\partial q_t(x)}{\partial t} = \frac{\sigma_t^2}{2} \Delta q_t(x) \,. \tag{98}$$

*Then the marginals $p_{t,\beta}(x) \propto q_t^\beta(x)$ satisfy the following PDE*

$$\frac{\partial}{\partial t} p_{t,\beta}(x) = \frac{\sigma_t^2}{2\beta} \Delta p_{t,\beta}(x) + p_{t,\beta}(x) \big[ g_t(x) - \mathbb{E}_{p_{t,\beta}} g_t(x) \big] \,, \tag{99}$$

$$g_t(x) = (\beta - 1) \frac{\sigma_t^2}{2} \Delta \log q_t(x) \,. \tag{100}$$

*Proof.* We want to find the partial derivative of the annealed density

$$p_{t,\beta}(x) = \frac{q_t(x)^\beta}{\int dx \, q_t(x)^\beta} \,, \quad \frac{\partial}{\partial t} p_{t,\beta}(x) = ? \tag{101}$$

By the straightforward calculations we have

$$\frac{\partial}{\partial t} \log p_{t,\beta} = \beta \frac{\partial}{\partial t} \log q_t - \int dx \, p_{t,\beta} \beta \frac{\partial}{\partial t} \log q_t \tag{102}$$

$$= \beta \frac{\sigma_t^2}{2} \Delta \log q_t + \beta \frac{\sigma_t^2}{2} \|\nabla \log q_t\|^2 - \int dx \, p_{t,\beta} \left[ \beta \frac{\sigma_t^2}{2} \Delta \log q_t + \beta \frac{\sigma_t^2}{2} \|\nabla \log q_t\| \right] \tag{103}$$

$$= \frac{\sigma_t^2}{2} \Delta \log p_{t,\beta} + \frac{\sigma_t^2}{2\beta} \|\nabla \log p_{t,\beta}\|^2 - \int dx \, p_{t,\beta} \left[ \frac{\sigma_t^2}{2} \Delta \log p_{t,\beta} + \frac{\sigma_t^2}{2\beta} \|\nabla \log p_{t,\beta}\|^2 \right] \tag{104}$$

$$= \frac{\sigma_t^2}{2\beta} \Delta \log p_{t,\beta} + \frac{\sigma_t^2}{2\beta} \|\nabla \log p_{t,\beta}\|^2 + \left( 1 - \frac{1}{\beta} \right) \frac{\sigma_t^2}{2} \Delta \log p_{t,\beta} \tag{105}$$

$$- \int dx \, p_{t,\beta} \left[ \left( 1 - \frac{1}{\beta} \right) \frac{\sigma_t^2}{2} \Delta \log p_{t,\beta} \right] \,. \tag{106}$$

Thus, we have

$$\frac{\partial}{\partial t} p_{t,\beta}(x) = \frac{\sigma_t^2}{2\beta} \Delta p_{t,\beta}(x) + p_{t,\beta}(x) \big[ g_t(x) - \mathbb{E}_{p_{t,\beta}} g_t(x) \big] \,, \tag{107}$$

$$g_t(x) = (\beta - 1) \frac{\sigma_t^2}{2} \Delta \log q_t(x) \,, \tag{108}$$

which can be simulated (for $\beta > 0$) as

$$dx_t = \frac{\sigma_t}{\sqrt{\beta}} dW_t \,, \tag{109}$$

$$dw_t = (\beta - 1) \frac{\sigma_t^2}{2} \Delta \log q_t(x_t) dt \,. \tag{110}$$

$\square$

**Proposition C.5** (Annealed Re-weighting). *Consider the marginals generated by the re-weighting equation*

$$\frac{\partial q_t(x)}{\partial t} = q_t(x) \big( g_t(x) - \mathbb{E}_{q_t(x)} g_t(x) \big) \,. \tag{111}$$

*The marginals $p_{t,\beta}(x) \propto q_t^\beta(x)$ satisfy the following PDE*

$$\frac{\partial}{\partial t} p_{t,\beta}(x) = p_{t,\beta} \big[ \beta g_t(x) - \mathbb{E}_{p_{t,\beta}} \beta g_t(x) \big] \,. \tag{112}$$

*Proof.* We want to find the partial derivative of the annealed density

$$p_{t,\beta}(x) = \frac{q_t(x)^\beta}{\int dx\ q_t(x)^\beta}\,, \quad \frac{\partial}{\partial t}p_{t,\beta}(x) = ? \tag{113}$$

By the straightforward calculations we have

$$\frac{\partial}{\partial t}\log p_{t,\beta} = \beta\frac{\partial}{\partial t}\log q_t - \int dx\ p_{t,\beta}\beta\frac{\partial}{\partial t}\log q_t \tag{114}$$

$$= \beta\big(g_t(x) - \mathbb{E}_{q_t(x)}g_t(x)\big) - \int dx\ p_{t,\beta}\big[\beta\big(g_t(x) - \mathbb{E}_{q_t(x)}g_t(x)\big)\big] \tag{115}$$

$$= \beta g_t(x) - \int dx\ p_{t,\beta}\beta g_t(x)\,. \tag{116}$$

Thus, we have

$$\frac{\partial}{\partial t}p_{t,\beta}(x) = p_{t,\beta}\big[\beta g_t(x) - \mathbb{E}_{p_{t,\beta}}\beta g_t(x)\big]\,, \tag{117}$$

which can be simulated as

$$dx_t = 0\,, \tag{118}$$

$$dw_t = \beta g_t(x_t)\,. \tag{119}$$

$\square$

---

**Proposition C.6** (Time-dependent annealing). *Consider the annealed marginals $p_{t,\beta}(x) \propto q_t(x)^\beta$ following some F*

$$dx_t = v_{t,\beta}(x_t) + \sigma_{t,\beta}dW_t\,, \tag{120}$$

$$dw_t = g_{t,\beta}(x_t)\,. \tag{121}$$

*Then, for the time-dependent schedule $\beta_t$, we have*

$$dx_t = v_{t,\beta_t}(x_t) + \sigma_{t,\beta_t}dW_t\,, \tag{122}$$

$$dw_t = g_{t,\beta_t}(x_t) + \frac{\partial\beta_t}{\partial t}\log q_t(x_t)\,, \tag{123}$$

*sampling from $p_{t,\beta_t}(x) \propto q_t(x)^{\beta_t}$.*

---

*Proof.* First, let's note that for the annealed marginals $p_{t,\beta}(x) \propto q_t(x)^\beta$ with constant $\beta$, we have

$$\frac{\partial}{\partial t}\log p_{t,\beta} = \beta\frac{\partial}{\partial t}\log q_t - \int dx\ p_{t,\beta}\left[\beta\frac{\partial}{\partial t}\log q_t\right] \tag{124}$$

$$= -\frac{1}{p_{t,\beta}}\big\langle\nabla, p_{t,\beta}v_{t,\beta}\big\rangle + \frac{1}{p_{t,\beta}}\frac{\sigma_{t,\beta}^2}{2}\Delta p_{t,\beta} + \big(g_{t,\beta} - \mathbb{E}_{p_{t,\beta}}g_{t,\beta}\big)\,. \tag{125}$$

Thus, for the time-dependent $\beta_t$, we have

$$\frac{\partial}{\partial t}\log p_{t,\beta_t} = \beta_t\frac{\partial}{\partial t}\log q_t + \frac{\partial\beta_t}{\partial t}\log q_t - \int dx\ p_{t,\beta_t}\left[\beta_t\frac{\partial}{\partial t}\log q_t + \frac{\partial\beta_t}{\partial t}\log q_t\right] \tag{126}$$

$$= -\frac{1}{p_{t,\beta_t}}\big\langle\nabla, p_{t,\beta_t}v_{t,\beta_t}\big\rangle + \frac{1}{p_{t,\beta_t}}\frac{\sigma_{t,\beta_t}^2}{2}\Delta p_{t,\beta_t} + \left[\left(g_{t,\beta_t} + \frac{\partial\beta_t}{\partial t}\log q_t\right) - \mathbb{E}_{p_{t,\beta_t}}\left(g_{t,\beta_t} + \frac{\partial\beta_t}{\partial t}\log q_t\right)\right]\,. \tag{127}$$

From which we have the statement of the proposition. $\square$

## C.2. Product

**Proposition C.7** (Product of Continuity Equations). *Consider marginals $q_t^{1,2}(x)$ generated by two different continuity equations*

$$\frac{\partial q_t^1(x)}{\partial t} = -\left\langle \nabla, q_t^1(x)v_t^1(x)\right\rangle, \quad \frac{\partial q_t^2(x)}{\partial t} = -\left\langle \nabla, q_t^2(x)v_t^2(x)\right\rangle. \tag{128}$$

*The product of densities $p_t(x) \propto q^1(x)q^2(x)$ satisfies the following PDE*

$$\frac{\partial}{\partial t}p_t(x) = -\left\langle \nabla, p_t(x)\big(v_t^1(x) + v_t^2(x)\big)\right\rangle + p_t(x)\big(g_t(x) - \mathbb{E}_{p_t(x)}g_t(x)\big), \tag{129}$$

$$g_t(x) = \left\langle \nabla \log q_t^1(x), v_t^2(x)\right\rangle + \left\langle \nabla \log q_t^2(x), v_t^1(x)\right\rangle. \tag{130}$$

*Proof.* For the continuity equations

$$\frac{\partial}{\partial t}q_t^{1,2}(x) = -\left\langle \nabla, q_t^{1,2}(x)v_t^{1,2}(x)\right\rangle, \tag{131}$$

we want to find the partial derivative of the annealed density

$$p_t(x) = \frac{q_t^1(x)q_t^2(x)}{\int dx\, q_t^1(x)q_t^2(x)}, \quad \frac{\partial}{\partial t}p_t(x) = ? \tag{132}$$

By the straightforward calculations we have

$$\frac{\partial}{\partial t}\log p_t = \frac{\partial}{\partial t}\log q_t^1 + \frac{\partial}{\partial t}\log q_t^2 - \int dx\, p_t\left[\frac{\partial}{\partial t}\log q_t^1 + \frac{\partial}{\partial t}\log q_t^2\right] \tag{133}$$

$$= -\left\langle \nabla, v_t^1 + v_t^2\right\rangle - \left\langle \nabla \log q_t^1, v_t^1\right\rangle - \left\langle \nabla \log q_t^2, v_t^2\right\rangle - \tag{134}$$

$$\quad - \int dx\, p_t\left[-\left\langle \nabla, v_t^1 + v_t^2\right\rangle - \left\langle \nabla \log q_t^1, v_t^1\right\rangle - \left\langle \nabla \log q_t^2, v_t^2\right\rangle\right] \tag{135}$$

$$= -\left\langle \nabla, v_t^1 + v_t^2\right\rangle - \left\langle \nabla \log p_t, v_t^1 + v_t^2\right\rangle + \left\langle \nabla \log q_t^1, v_t^2\right\rangle + \left\langle \nabla \log q_t^2, v_t^1\right\rangle - \tag{136}$$

$$\quad - \int dx\, p_t\left[\left\langle \nabla \log q_t^1, v_t^2\right\rangle + \left\langle \nabla \log q_t^2, v_t^1\right\rangle\right]. \tag{137}$$

Thus, we have

$$\frac{\partial}{\partial t}p_t(x) = -\left\langle \nabla, p_t(x)\big(v_t^1(x) + v_t^2(x)\big)\right\rangle + p_t(x)\big(g_t(x) - \mathbb{E}_{p_t(x)}g_t(x)\big), \tag{138}$$

$$g_t(x) = \left\langle \nabla \log q_t^1(x), v_t^2(x)\right\rangle + \left\langle \nabla \log q_t^2(x), v_t^1(x)\right\rangle, \tag{139}$$

which can be simulated as

$$dx_t = \big(v_t^1(x_t) + v_t^2(x_t)\big)dt, \tag{140}$$

$$dw_t = \left[\left\langle \nabla \log q_t^1(x_t), v_t^2(x_t)\right\rangle + \left\langle \nabla \log q_t^2(x_t), v_t^1(x_t)\right\rangle\right]dt. \tag{141}$$

$\square$

**Proposition C.8** (Product of Diffusion Equations). *Consider marginals $q_t^{1,2}(x)$ generated by two different diffusion equations*

$$\frac{\partial q_t^1(x)}{\partial t} = \frac{\sigma_t^2}{2}\Delta q_t^1(x), \quad \frac{\partial q_t^2(x)}{\partial t} = \frac{\sigma_t^2}{2}\Delta q_t^2(x). \tag{142}$$

*The product of densities $p_t(x) \propto q^1(x)q^2(x)$ satisfies the following PDE*

$$\frac{\partial}{\partial t}p_t(x) = \frac{\sigma_t^2}{2}\Delta p_t(x) + p_t(x)\big(g_t(x) - \mathbb{E}_{p_t(x)}g_t(x)\big), \tag{143}$$

$$g_t(x) = -\sigma_t^2\left\langle \nabla \log q_t^1(x), \nabla \log q_t^2(x)\right\rangle. \tag{144}$$

*Proof.* We want to find the partial derivative of the annealed density

$$p_t(x) = \frac{q_t^1(x)q_t^2(x)}{\int dx\, q_t^1(x)q_t^2(x)}, \quad \frac{\partial}{\partial t}p_t(x) = ? \tag{145}$$

By the straightforward calculations we have

$$\frac{\partial}{\partial t}\log p_t = \frac{\partial}{\partial t}\log q_t^1 + \frac{\partial}{\partial t}\log q_t^2 - \int dx\, p_t\left[\frac{\partial}{\partial t}\log q_t^1 + \frac{\partial}{\partial t}\log q_t^2\right] \tag{146}$$

$$= \frac{\sigma_t^2}{2}\Delta\log q_t^1 + \frac{\sigma_t^2}{2}\|\nabla\log q_t^1\|^2 + \frac{\sigma_t^2}{2}\Delta\log q_t^2 + \frac{\sigma_t^2}{2}\|\nabla\log q_t^2\|^2 - \tag{147}$$

$$- \int dx\, p_t\left[\frac{\sigma_t^2}{2}\Delta\log q_t^1 + \frac{\sigma_t^2}{2}\|\nabla\log q_t^1\|^2 + \frac{\sigma_t^2}{2}\Delta\log q_t^2 + \frac{\sigma_t^2}{2}\|\nabla\log q_t^2\|^2\right] \tag{148}$$

$$= \frac{\sigma_t^2}{2}\Delta\log p_t + \frac{\sigma_t^2}{2}\|\nabla\log p_t\|^2 - \sigma_t^2\langle\nabla\log q_t^1, \nabla\log q_t^2\rangle - \int dx\, p_t\left[-\sigma_t^2\langle\nabla\log q_t^1, \nabla\log q_t^2\rangle\right]. \tag{149}$$

Thus, we have

$$\frac{\partial}{\partial t}p_t(x) = \frac{\sigma_t^2}{2}\Delta p_t(x) + p_t(x)\big(g_t(x) - \mathbb{E}_{p_t(x)}g_t(x)\big), \tag{150}$$

$$g_t(x) = -\sigma_t^2\langle\nabla\log q_t^1(x), \nabla\log q_t^2(x)\rangle, \tag{151}$$

which can be simulated as

$$dx_t = \sigma_t dW_t, \tag{152}$$

$$dw_t = \left[-\sigma_t^2\langle\nabla\log q_t^1(x_t), \nabla\log q_t^2(x_t)\rangle\right]dt. \tag{153}$$

$\square$

---

**Proposition C.9** (Product of Re-weightings). *Consider marginals $q_t^{1,2}(x)$ generated by two different diffusion equations*

$$\frac{\partial q_t^1(x)}{\partial t} = \big(g_t^1(x) - \mathbb{E}_{q_t^1}g_t^1(x)\big)q_t^1(x), \quad \frac{\partial q_t^2(x)}{\partial t} = \big(g_t^2(x) - \mathbb{E}_{q_t^2}g_t^2(x)\big)q_t^2(x). \tag{154}$$

*The product of densities $p_t(x) \propto q^1(x)q^2(x)$ satisfies the following PDE*

$$\frac{\partial}{\partial t}p_t(x) = p_t(x)\big(g_t(x) - \mathbb{E}_{p_t(x)}g_t(x)\big), \tag{155}$$

$$g_t(x) = g_t^1(x) + g_t^2(x), \tag{156}$$

---

*Proof.* We want to find the partial derivative of the annealed density

$$p_t(x) = \frac{q_t^1(x)q_t^2(x)}{\int dx\, q_t^1(x)q_t^2(x)}, \quad \frac{\partial}{\partial t}p_t(x) = ? \tag{157}$$

By the straightforward calculations we have

$$\frac{\partial}{\partial t}\log p_t = \frac{\partial}{\partial t}\log q_t^1 + \frac{\partial}{\partial t}\log q_t^2 - \int dx\, p_t\left[\frac{\partial}{\partial t}\log q_t^1 + \frac{\partial}{\partial t}\log q_t^2\right] \tag{158}$$

$$= \big(g_t^1(x) - \mathbb{E}_{q_t^1}g_t^1(x)\big) + \big(g_t^2(x) - \mathbb{E}_{q_t^2}g_t^2(x)\big) - \tag{159}$$

$$- \int dx\, p_t\left[\big(g_t^1(x) - \mathbb{E}_{q_t^1}g_t^1(x)\big) + \big(g_t^2(x) - \mathbb{E}_{q_t^2}g_t^2(x)\big)\right] \tag{160}$$

$$= g_t^1(x) + g_t^2(x) - \int dx\, p_t\left[g_t^1(x) + g_t^2(x)\right]. \tag{161}$$

Thus, we have

$$\frac{\partial}{\partial t}p_t(x) = p_t(x)\big(g_t(x) - \mathbb{E}_{p_t(x)}g_t(x)\big),$$ (162)

$$g_t(x) = g_t^1(x) + g_t^2(x),$$ (163)

which can be simulated as

$$dx_t = 0,$$ (164)

$$dw_t = g_t^1(x_t) + g_t^2(x_t).$$ (165)

$\square$

# D. Proofs of Propositions

**Proposition D.1** (Annealed SDE). *Consider the SDE*

$$dx_t = \big(-f_t(x_t) + \sigma_t^2 \nabla \log q_t(x_t)\big)dt + \sigma_t dW_t,$$ (166)

*then the samples from the annealed marginals $p_{t,\beta}(x) \propto q_t(x)^\beta$ can be obtained via the following family of SDEs*

$$dx_t = \big(-f_t(x_t) + (\beta + (1-\beta)a)\sigma_t^2 \nabla \log q_t(x_t)\big)dt + \sqrt{\frac{\sigma_t^2(\beta + (1-\beta)2a)}{\beta}}dW_t,$$ (167)

$$dw_t = \left[(\beta - 1)\langle\nabla, f_t(x_t)\rangle + \frac{1}{2}\sigma_t^2\beta(\beta - 1)\|\nabla \log q_t(x_t)\|^2\right]dt,$$ (168)

*where the parameter $a$ satisfies $0 \le (\beta + (1-\beta)2a)/\beta$.*

*Proof.* For the following SDE

$$dx_t = \big(-f_t(x_t) + \sigma_t^2 \nabla \log q_t(x_t)\big)dt + \sigma_t dW_t,$$ (169)

let's consider everything but the drift $f_t$. Thus, we can write the following PDE

$$\frac{\partial q_t}{\partial t} = \big\langle\nabla, q_t\big[(1-a)\sigma_t^2 \nabla \log q_t(x_t) + a\sigma_t^2 \nabla \log q_t(x_t)\big]\big\rangle + (1-b)\frac{\sigma_t^2}{2}\Delta q_t + b\frac{\sigma_t^2}{2}\Delta q_t.$$ (170)

We apply Prop. C.2, Prop. C.1, Prop. C.4, Prop. C.3 (rules from Table 1) to the corresponding terms of the PDE above. Hence, the formulas for the weights are

$$g_t(x) = (1-a)\sigma_t^2\beta(\beta - 1)\|\nabla \log q_t(x)\|^2 - a\sigma_t^2(\beta - 1)\Delta \log q_t(x)+$$ (171)

$$+ (\beta - 1)\frac{(1-b)\sigma_t^2}{2}\Delta \log q_t(x_t) - \beta(\beta - 1)\frac{b\sigma_t^2}{2}\|\nabla \log q_t(x_t)\|^2.$$ (172)

Let's cancel out the term with the Laplacians, hence, we have $2a = 1 - b$ and

$$g_t(x) = (1 - a - b/2)\sigma_t^2\beta(\beta - 1)\|\nabla \log q_t(x)\|^2 = \frac{1}{2}\sigma_t^2\beta(\beta - 1)\|\nabla \log q_t(x)\|^2.$$ (173)

The PDE for the density is

$$\frac{\partial p_{t,\beta}}{\partial t} = -\big\langle\nabla, p_{t,\beta}\big(-f_t + (\beta(1-a) + a)\sigma_t^2 \nabla \log q_t\big)\big\rangle + \left(\frac{1-b}{\beta} + b\right)\frac{\sigma_t^2}{2}\Delta p_{t,\beta} + p_{t,\beta}\big(g_t - \mathbb{E}_{p_{t,\beta}}g_t\big)$$ (174)

$$= -\big\langle\nabla, p_{t,\beta}\big(-f_t + (\beta + (1-\beta)a)\sigma_t^2 \nabla \log q_t\big)\big\rangle + \frac{\beta + (1-\beta)2a}{\beta}\frac{\sigma_t^2}{2}\Delta p_{t,\beta} + p_{t,\beta}\big(g_t - \mathbb{E}_{p_{t,\beta}}g_t\big)$$ (175)

This corresponds to the following family of SDEs ($0 \leq \beta + (1-\beta)2a$)

$$dx_t = \left(-f_t(x_t) + (\beta + (1-\beta)a)\sigma_t^2 \nabla \log q_t(x_t)\right)dt + \sqrt{\frac{\sigma_t^2(\beta + (1-\beta)2a)}{\beta}}dW_t \,, \tag{176}$$

$$dw_t = \left[(\beta - 1)\langle \nabla, f_t(x_t)\rangle + \frac{1}{2}\sigma_t^2 \beta(\beta-1)\|\nabla \log q_t(x_t)\|^2\right]dt \,. \tag{177}$$

$\square$

**Proposition D.2** (Product of Experts). *Consider two PDEs corresponding to the following SDEs*

$$dx_t = (-f_t(x_t) + \sigma_t^2 \nabla \log q_t^{1,2}(x_t))dt + \sigma_t dW_t \,, \tag{178}$$

*which marginals we denote as $q_t^1(x_t)$ and $q_t^2(x_t)$. The following family of SDEs (for $0 \leq (\beta + (1-\beta)2a)/\beta$) corresponds to the product of the marginals $p_{t,\beta}(x) \propto (q_t^1(x)q_t^2(x))^\beta$*

$$dx_t = \left(-f_t(x_t) + \sigma_t^2(\beta + (1-\beta)a)(\nabla \log q_t^1(x_t) + \nabla \log q_t^2(x_t))\right)dt + \sqrt{\frac{\sigma_t^2(\beta + (1-\beta)2a)}{\beta}}dW_t \,, \tag{179}$$

$$dw_t = \left[\beta\sigma_t^2\langle \nabla \log q_t^1(x_t), \nabla \log q_t^2(x_t)\rangle + \beta(\beta-1)\frac{\sigma_t^2}{2}\|\nabla \log q_t^1(x_t) + \nabla \log q_t^2(x_t)\|^2 + (2\beta - 1)\langle \nabla, f_t(x_t)\rangle\right]dt \,. \tag{180}$$

*Proof.* First, according to Table 1, we have the following PDE for the product density $p_t(x) \propto q_t^1(x)q_t^2(x)$ is

$$\frac{\partial p_t(x)}{\partial t} = -\left\langle \nabla, p_t(x)\left(-2f_t(x) + \sigma_t^2(\nabla \log q_t^1(x) + \nabla \log q_t^2(x))\right)\right\rangle + \frac{\sigma_t^2}{2}\Delta p_t(x) + \tag{181}$$

$$+ p_t(x)(g_t(x) - \mathbb{E}_{p_t}g_t(x)) \,, \tag{182}$$

where

$$g_t(x) = \left\langle \nabla \log q_t^1(x), -f_t(x) + \sigma_t^2 \nabla \log q_t^2(x)\right\rangle + \left\langle \nabla \log q_t^2(x), -f_t(x) + \sigma_t^2 \nabla \log q_t^1(x)\right\rangle - \tag{183}$$

$$- \sigma_t^2\left\langle \nabla \log q_t^1(x), \nabla \log q_t^2(x)\right\rangle \tag{184}$$

$$= \sigma_t^2\left\langle \nabla \log q_t^1(x), \nabla \log q_t^2(x)\right\rangle - \left\langle f_t(x), \nabla \log q_t^1(x) + \nabla \log q_t^2(x)\right\rangle \,. \tag{185}$$

Now, combining Prop. D.1 and Prop. C.5, for the annealed density $p_{t,\beta} \propto p_t(x)^\beta$ we have

$$\frac{\partial p_{t,\beta}(x)}{\partial t} = -\left\langle \nabla, p_{t,\beta}(x)\left(-2f_t(x) + \sigma_t^2(\beta + (1-\beta)a)(\nabla \log q_t^1(x) + \nabla \log q_t^2(x))\right)\right\rangle + \tag{186}$$

$$+ \frac{\beta + (1-\beta)2a}{\beta}\frac{\sigma_t^2}{2}\Delta p_{t,\beta}(x) + p_{t,\beta}(x)\left(g_t(x) - \mathbb{E}_{p_{t,\beta}}g_t(x)\right) \,, \tag{187}$$

$$g_t(x) = \beta\sigma_t^2\left\langle \nabla \log q_t^1(x), \nabla \log q_t^2(x)\right\rangle - \beta\left\langle f_t(x), \nabla \log q_t^1(x) + \nabla \log q_t^2(x)\right\rangle + \tag{188}$$

$$+ (\beta - 1)\left\langle \nabla, 2f_t(x)\right\rangle + \beta(\beta-1)\frac{\sigma_t^2}{2}\|\nabla \log q_t^1(x) + \nabla \log q_t^2(x)\|^2 \,. \tag{189}$$

The last step is interpreting $\left\langle \nabla, p_{t,\beta}(x)f_t(x)\right\rangle$ as the weight term, i.e.

$$\frac{\partial p_{t,\beta}(x)}{\partial t} = -\left\langle \nabla, p_{t,\beta}(x)\left(-f_t(x) + \sigma_t^2(\beta + (1-\beta)a)(\nabla \log q_t^1(x) + \nabla \log q_t^2(x))\right)\right\rangle + \tag{190}$$

$$+ \frac{\beta + (1-\beta)2a}{\beta}\frac{\sigma_t^2}{2}\Delta p_{t,\beta}(x) + p_{t,\beta}(x)\left(g_t(x) - \mathbb{E}_{p_{t,\beta}}g_t(x)\right) \,, \tag{191}$$

$$g_t(x) = \beta\sigma_t^2\left\langle \nabla \log q_t^1(x), \nabla \log q_t^2(x)\right\rangle + \beta(\beta-1)\frac{\sigma_t^2}{2}\|\nabla \log q_t^1(x) + \nabla \log q_t^2(x)\|^2 + \tag{192}$$

$$+ (2\beta - 1)\left\langle \nabla, f_t(x)\right\rangle \,. \tag{193}$$

Thus, we get the following family of SDEs (for $0 \le (\beta + (1-\beta)2a)/\beta$)

$$dx_t = \left(-f_t(x_t) + \sigma_t^2(\beta + (1-\beta)a)(\nabla \log q_t^1(x_t) + \nabla \log q_t^2(x_t))\right)dt + \sqrt{\frac{\sigma_t^2(\beta + (1-\beta)2a)}{\beta}}dW_t\,, \tag{194}$$

$$dw_t = \left[\beta\sigma_t^2\langle \nabla \log q_t^1(x_t), \nabla \log q_t^2(x_t)\rangle + \beta(\beta-1)\frac{\sigma_t^2}{2}\left\|\nabla \log q_t^1(x_t) + \nabla \log q_t^2(x_t)\right\|^2 + (2\beta-1)\langle \nabla, f_t(x_t)\rangle\right]dt\,. \tag{195}$$

$\square$

---

**Proposition D.3** (Classifier-free Guidance). *Consider two PDEs corresponding to the following SDEs*

$$dx_t = (-f_t(x_t) + \sigma_t^2 \nabla \log q_t^{1,2}(x_t))dt + \sigma_t dW_t\,, \tag{196}$$

*which marginals we denote as $q_t^1(x_t)$ and $q_t^2(x_t)$. The SDE corresponding to the geometric average of the marginals $p_{t,\beta}(x) \propto q_t^1(x)^{1-\beta}q_t^2(x)^\beta$, for $(\beta + 2a(1-\beta))/\beta \ge 0$, is*

$$dx_t = \left(-f_t(x_t) + \sigma_t^2\left(1 + \frac{a(1-\beta)}{\beta}\right)\left((1-\beta)\nabla \log q_t^1(x_t) + \beta \nabla \log q_t^2(x_t)\right)\right)dt + \sigma_t\sqrt{1 + \frac{2a(1-\beta)}{\beta}}dW_t\,, \tag{197}$$

$$dw_t = \frac{1}{2}\sigma_t^2 \beta(\beta-1)\left\|\nabla \log q_t^1(x_t) - \nabla \log q_t^2(x_t)\right\|^2 + 2a\sigma_t^2(\beta-1)^2\langle \nabla \log q_t^1(x_t), \nabla \log q_t^2(x_t)\rangle\,. \tag{198}$$

---

*Proof.* First, according to Prop. D.1, we perform annealing $p_{t,1-\beta}^1(x) \propto q_t^1(x)^{1-\beta}$ and $p_{t,\beta}^2(x) \propto q_t^2(x)^\beta$, i.e.

$$\frac{\partial p_{t,1-\beta}^1(x)}{\partial t} = -\langle \nabla, p_{t,1-\beta}^1(x)\left(-f_t(x) + \sigma_t^2(1-\beta+\beta a_1)\nabla \log q_t^1(x)\right)\rangle + \frac{1-\beta+2\beta a_1}{1-\beta}\frac{\sigma_t^2}{2}\Delta p_{t,1-\beta}^1(x)+ \tag{199}$$

$$+ p_{t,1-\beta}^1(x)\left(g_t(x) - \mathbb{E}_{p_{t,1-\beta}^1}g_t(x)\right), \tag{200}$$

$$g_t(x) = -\beta\langle \nabla, f_t(x)\rangle + \frac{1}{2}\sigma_t^2 \beta(\beta-1)\left\|\nabla \log q_t^1(x_t)\right\|^2, \tag{201}$$

and

$$\frac{\partial p_{t,\beta}^2(x)}{\partial t} = -\langle \nabla, p_{t,\beta}^2(x)\left(-f_t(x) + \sigma_t^2(\beta + (1-\beta)a_2)\nabla \log q_t^2(x)\right)\rangle + \frac{\beta + (1-\beta)2a_2}{\beta}\frac{\sigma_t^2}{2}\Delta p_{t,\beta}^2(x)+ \tag{202}$$

$$+ p_{t,\beta}^2(x)\left(g_t(x) - \mathbb{E}_{p_{t,\beta}^2}g_t(x)\right), \tag{203}$$

$$g_t(x) = (\beta-1)\langle \nabla, f_t(x)\rangle + \frac{1}{2}\sigma_t^2\beta(\beta-1)\left\|\nabla \log q_t^2(x_t)\right\|^2, \tag{204}$$

Now, we would like to match the diffusion coefficient (to directly apply Prop. C.8 for diffusion equations in the product case, and avoid the evaluation of additional Laplacian terms in the weights).

$$\frac{1-\beta+2\beta a_1}{1-\beta} = \frac{\beta + (1-\beta)2a_2}{\beta} \implies \beta - \beta^2 + 2\beta^2 a_1 = \beta - \beta^2 + (1-\beta)^2 2a_2 \tag{205}$$

$$\beta^2 a_1 = (1-\beta)^2 a_2 \implies a_2 := a\,, \quad a_1 = \frac{a(1-\beta)^2}{\beta^2}\,. \tag{206}$$

Now, according to Table 1, for the product density $p_{t,\beta} \propto p_{t,1-\beta}^1(x)p_{t,\beta}^2(x)$, we have

$$\frac{\partial p_{t,\beta}(x)}{\partial t} = -\left\langle \nabla, p_{t,\beta}(x)\left(-2f_t(x) + \sigma_t^2(\beta + (1-\beta)a)\left(\frac{1-\beta}{\beta}\nabla \log q_t^1(x) + \nabla \log q_t^2(x)\right)\right)\right\rangle + \tag{207}$$

$$+ \frac{\beta + (1-\beta)2a}{\beta}\frac{\sigma_t^2}{2}\Delta p_{t,\beta}(x) + p_{t,\beta}(x)\left(g_t(x) - \mathbb{E}_{p_{t,\beta}}g_t(x)\right), \tag{208}$$

$$g_t(x) = \underbrace{-\beta\langle \nabla, f_t(x)\rangle + \frac{1}{2}\sigma_t^2\beta(\beta-1)\left\|\nabla \log q_t^1(x)\right\|^2}_{Eq.~(201)} + \tag{209}$$

$$+ \underbrace{(\beta-1)\langle \nabla, f_t(x)\rangle + \frac{1}{2}\sigma_t^2\beta(\beta-1)\left\|\nabla \log q_t^2(x)\right\|^2}_{Eq.~(204)} + \tag{210}$$

$$+ (1-\beta)\langle \nabla \log q_t^1(x), -f_t(x) + \sigma_t^2(\beta + (1-\beta)a)\nabla \log q_t^2(x)\rangle + \tag{211}$$

$$+ \beta\left\langle \nabla \log q_t^2(x), -f_t(x) + \sigma_t^2\frac{(1-\beta)}{\beta}(\beta + (1-\beta)a)\nabla \log q_t^1(x)\right\rangle - \tag{212}$$

$$- \sigma_t^2\beta(1-\beta)\langle \nabla \log q_t^1(x), \nabla \log q_t^2(x)\rangle \tag{213}$$

where the terms in the last three lines arise from the conversion rules from the product in Table 1. Finally, we obtain

$$g_t(x) = \frac{1}{2}\sigma_t^2\beta(\beta-1)\left(\left\|\nabla \log q_t^1(x) - \nabla \log q_t^2(x)\right\|^2 - 4\frac{a(1-\beta)}{\beta}\langle \nabla \log q_t^1(x), \nabla \log q_t^2(x)\rangle\right) - \tag{214}$$

$$- \langle \nabla, f_t(x)\rangle - \langle (1-\beta)\nabla \log q_t^1(x) + \beta\nabla \log q_t^2(x), f_t(x)\rangle. \tag{215}$$

Finally, we re-interpret $\langle \nabla, p_{t,\beta}(x)f_t(x)\rangle$ as the weighting term, and get

$$\frac{\partial p_{t,\beta}(x)}{\partial t} = -\left\langle \nabla, p_{t,\beta}(x)\left(-f_t(x) + \sigma_t^2\left(1 + \frac{a(1-\beta)}{\beta}\right)\left((1-\beta)\nabla \log q_t^1(x) + \beta\nabla \log q_t^2(x)\right)\right)\right\rangle + \tag{216}$$

$$+ \frac{\sigma_t^2}{2}\left(1 + \frac{2a(1-\beta)}{\beta}\right)\Delta p_{t,\beta}(x) + p_{t,\beta}(x)\left(g_t(x) - \mathbb{E}_{p_{t,\beta}}g_t(x)\right), \tag{217}$$

$$g_t(x) = \frac{1}{2}\sigma_t^2\beta(\beta-1)\left\|\nabla \log q_t^1(x) - \nabla \log q_t^2(x)\right\|^2 + 2a\sigma_t^2(\beta-1)^2\langle \nabla \log q_t^1(x), \nabla \log q_t^2(x)\rangle. \tag{218}$$

Thus, for $(\beta + 2a(1-\beta))/\beta \geq 0$, we have

$$dx_t = \left(-f_t(x_t) + \sigma_t^2\left(1 + \frac{a(1-\beta)}{\beta}\right)\left((1-\beta)\nabla \log q_t^1(x_t) + \beta\nabla \log q_t^2(x_t)\right)\right)dt + \sigma_t\sqrt{1 + \frac{2a(1-\beta)}{\beta}}dW_t, \tag{219}$$

$$dw_t = \frac{1}{2}\sigma_t^2\beta(\beta-1)\left\|\nabla \log q_t^1(x_t) - \nabla \log q_t^2(x_t)\right\|^2 + 2a\sigma_t^2(\beta-1)^2\langle \nabla \log q_t^1(x_t), \nabla \log q_t^2(x_t)\rangle. \tag{220}$$

$\square$

**Proposition D.4** (PoE + CFG). *Consider two PDEs corresponding to the following SDEs*

$$dx_t = (-f_t(x_t) + \sigma_t^2 \nabla \log q_t(x_t))dt + \sigma_t dW_t\,, \tag{221}$$

$$dx_t = (-f_t(x_t) + \sigma_t^2 \nabla \log q_t^{1,2}(x_t))dt + \sigma_t dW_t\,, \tag{222}$$

*with corresponding marginals* $q_t(x_t)$, $q_t^1(x_t)$ *and* $q_t^2(x_t)$. *The SDE corresponding to the product of the marginals* $p_{t,\beta}(x) \propto q_t(x)^{2(1-\beta)}(q_t^1(x)q_t^2(x))^\beta$ *is*

$$dx_t = \left(-f_t(x_t) + \sigma_t^2(v_t^1(x_t) + v_t^2(x_t))\right)dt + \sigma_t dW_t\,, \tag{223}$$

$$dw_t = \frac{1}{2}\sigma_t^2\beta(\beta-1)\left(\left\|\nabla \log q_t(x_t) - \nabla \log q_t^1(x_t)\right\|^2 + \left\|\nabla \log q_t(x_t) - \nabla \log q_t^2(x_t)\right\|^2\right)+ \tag{224}$$

$$+ \sigma_t^2\langle v_t^1(x_t), v_t^2(x_t)\rangle + \langle \nabla, f_t(x_t)\rangle\,, \tag{225}$$

*where we denote* $v_t^{1,2}(x) = (1-\beta)\nabla \log q_t(x) + \beta\nabla \log q_t^{1,2}(x)$.

*Proof.* Using Prop. D.3, we start from the SDEs simulating the product $q_t(x)^{(1-\beta)}q_t^1(x)^\beta$ and $q_t(x)^{(1-\beta)}q_t^2(x)^\beta$, i.e.

$$dx_t = \left(-f_t(x_t) + \sigma_t^2(\underbrace{(1-\beta)\nabla \log q_t(x_t) + \beta\nabla \log q_t^1(x_t)}_{v_t^1(x_t)})\right)dt + \sigma_t dW_t\,, \tag{226}$$

$$dw_t = \frac{1}{2}\sigma_t^2\beta(\beta-1)\left\|\nabla \log q_t(x_t) - \nabla \log q_t^1(x_t)\right\|^2\,, \tag{227}$$

$$dx_t = \left(-f_t(x_t) + \sigma_t^2(\underbrace{(1-\beta)\nabla \log q_t(x_t) + \beta\nabla \log q_t^2(x_t)}_{v_t^2(x_t)})\right)dt + \sigma_t dW_t\,, \tag{228}$$

$$dw_t = \frac{1}{2}\sigma_t^2\beta(\beta-1)\left\|\nabla \log q_t(x_t) - \nabla \log q_t^2(x_t)\right\|^2\,. \tag{229}$$

Then we consider the product of these SDEs, i.e.

$$\frac{\partial p_{t,\beta}(x)}{\partial t} = -\left\langle\nabla, p_{t,\beta}(x)\left(-2f_t(x) + \sigma_t^2(v_t^1(x) + v_t^2(x))\right)\right\rangle + \frac{\sigma_t^2}{2}\Delta p_{t,\beta}(x) + p_{t,\beta}(x)\left(g_t(x) - \mathbb{E}_{p_{t,\beta}}g_t(x)\right)\,, \tag{230}$$

$$g_t(x) = \frac{1}{2}\sigma_t^2\beta(\beta-1)\left(\left\|\nabla \log q_t(x) - \nabla \log q_t^1(x)\right\|^2 + \left\|\nabla \log q_t(x) - \nabla \log q_t^2(x)\right\|^2\right)+ \tag{231}$$

$$+ \left\langle v_t^1(x), -f_t(x) + \sigma_t^2 v_t^2(x)\right\rangle + \left\langle v_t^2(x), -f_t(x) + \sigma_t^2 v_t^1(x)\right\rangle - \sigma_t^2\left\langle v_t^1(x), v_t^2(x)\right\rangle \tag{232}$$

$$= \frac{1}{2}\sigma_t^2\beta(\beta-1)\left(\left\|\nabla \log q_t(x) - \nabla \log q_t^1(x)\right\|^2 + \left\|\nabla \log q_t(x) - \nabla \log q_t^2(x)\right\|^2\right)+ \tag{233}$$

$$+ \sigma_t^2\left\langle v_t^1(x), v_t^2(x)\right\rangle - \left\langle f_t(x), v_t^1(x) + v_t^2(x)\right\rangle\,. \tag{234}$$

Re-interpreting $\langle\nabla, p_{t,\beta}(x)f_t(x)\rangle$, we get

$$\frac{\partial p_{t,\beta}(x)}{\partial t} = -\left\langle\nabla, p_{t,\beta}(x)\left(-f_t(x) + \sigma_t^2(v_t^1(x) + v_t^2(x))\right)\right\rangle + \frac{\sigma_t^2}{2}\Delta p_{t,\beta}(x) + p_{t,\beta}(x)\left(g_t(x) - \mathbb{E}_{p_{t,\beta}}g_t(x)\right)\,, \tag{235}$$

$$g_t(x) = \frac{1}{2}\sigma_t^2\beta(\beta-1)\left(\left\|\nabla \log q_t(x) - \nabla \log q_t^1(x)\right\|^2 + \left\|\nabla \log q_t(x) - \nabla \log q_t^2(x)\right\|^2\right)+ \tag{236}$$

$$+ \sigma_t^2\left\langle v_t^1(x), v_t^2(x)\right\rangle + \left\langle\nabla, f_t(x)\right\rangle\,, \tag{237}$$

which corresponds to

$$dx_t = \left(-f_t(x_t) + \sigma_t^2(v_t^1(x_t) + v_t^2(x_t))\right)dt + \sigma_t dW_t\,, \tag{238}$$

$$dw_t = \frac{1}{2}\sigma_t^2\beta(\beta-1)\left(\left\|\nabla \log q_t(x_t) - \nabla \log q_t^1(x_t)\right\|^2 + \left\|\nabla \log q_t(x_t) - \nabla \log q_t^2(x_t)\right\|^2\right)+ \tag{239}$$

$$+ \sigma_t^2\left\langle v_t^1(x_t), v_t^2(x_t)\right\rangle + \left\langle\nabla, f_t(x_t)\right\rangle\,. \tag{240}$$

$\square$

**Proposition D.5** (Target Score Product SDE). *Consider two PDEs corresponding to the following SDEs*

$$dx_t = (-f_t(x_t) + \sigma_t^2 \nabla \log q_t^i(x_t))dt + \sigma_t dW_t \,, \tag{241}$$

*with corresponding marginals $q_t^i(x_t)$. The SDE corresponding to the product of the marginals $p_{t,\beta}(x) \propto \prod_i q_t^i(x)^{\beta_i}$, can be simulated as follows*

$$dx_t = \left(-f_t(x_t) + \sigma_t^2 \sum_i \beta_i \nabla \log q_t^i(x)\right)dt + \sigma_t dW_t \,, \tag{242}$$

$$dw_t = \left[\left(\sum_i \beta_i - 1\right)\langle \nabla, f_t(x_t)\rangle + \frac{\sigma_t^2}{2}\left\|\sum_i \beta_i \nabla \log q_t^i(x_t)\right\|^2 - \frac{\sigma_t^2}{2}\sum_i \beta_i\|\nabla \log q_t^i(x_t)\|^2\right]dt \,. \tag{243}$$

*Proof.* The target log-density is defined as

$$\log p_t(x) = \sum_i \beta_i \log q_t^i(x) - \log Z_t \,. \tag{244}$$

The time-derivative of the target log-density is

$$\frac{\partial}{\partial t}\log p_t(x) = \sum_i \beta_i \frac{\partial}{\partial t}\log q_t^i(x) - \mathbb{E}_{p_t(x)}\sum_i \beta_i \frac{\partial}{\partial t}\log q_t^i(x) \,. \tag{245}$$

We focus on the first term, and write for all the marginals their corresponding PDEs

$$\frac{\partial}{\partial t}\log q_t^i(x) = -\langle \nabla, -f_t(x) + \sigma_t^2 \nabla \log q_t^i(x)\rangle - \langle \nabla \log q_t^i(x), -f_t(x) + \sigma_t^2 \nabla \log q_t^i(x)\rangle + \tag{246}$$

$$+ \frac{\sigma_t^2}{2}\Delta \log q_t^i(x) + \frac{\sigma_t^2}{2}\|\nabla \log q_t^i(x)\|^2 \tag{247}$$

$$= -\langle \nabla, -f_t(x) + \sigma_t^2 \nabla \log q_t^i(x)\rangle - \langle \nabla \log q_t^i(x), -f_t(x)\rangle + \frac{\sigma_t^2}{2}\Delta \log q_t^i(x) - \frac{\sigma_t^2}{2}\|\nabla \log q_t^i(x)\|^2 \,.$$

$$\sum_i \beta_i \frac{\partial}{\partial t}\log q_t^i(x) = -\left\langle \nabla, -\sum_i \beta_i f_t(x) + \sigma_t^2 \nabla \log p_t(x)\right\rangle - \langle \nabla \log p_t(x), -f_t(x)\rangle + \tag{248}$$

$$+ \frac{\sigma_t^2}{2}\Delta \log p_t(x) - \frac{\sigma_t^2}{2}\sum_i \beta_i\|\nabla \log q_t^i(x)\|^2 \tag{249}$$

$$= -\langle \nabla, -f_t(x) + \sigma_t^2 \nabla \log p_t(x)\rangle - \langle \nabla \log p_t(x), -f_t(x) + \sigma_t^2 \nabla \log p_t(x)\rangle + \tag{250}$$

$$+ \frac{\sigma_t^2}{2}\Delta \log p_t(x) + \frac{\sigma_t^2}{2}\|\nabla \log p_t(x)\|^2 + \tag{251}$$

$$+ \left(\sum_i \beta_i - 1\right)\langle \nabla, f_t(x)\rangle + \frac{\sigma_t^2}{2}\|\nabla \log p_t(x)\|^2 - \frac{\sigma_t^2}{2}\sum_i \beta_i\|\nabla \log q_t^i(x)\|^2 \,. \tag{252}$$

Writing down the PDE $p_t(x)$, we get

$$\frac{\partial p_t(x)}{\partial t} = -\langle \nabla, p_t(x)(-f_t(x) + \sigma_t^2 \nabla \log p_t(x))\rangle + \frac{\sigma_t^2}{2}\Delta p_t(x) + p_t(x)(g_t(x) - \mathbb{E}_{p_t(x)}g_t(x)) \,, \tag{253}$$

$$g_t(x) = \left(\sum_i \beta_i - 1\right)\langle \nabla, f_t(x)\rangle + \frac{\sigma_t^2}{2}\|\nabla \log p_t(x)\|^2 - \frac{\sigma_t^2}{2}\sum_i \beta_i\|\nabla \log q_t^i(x)\|^2 \,, \tag{254}$$

which ends the proof.

In particular, for the target $p_t(x) \propto q_t(x)^{(1-\beta)}(q_t^1(x)q_t^2(x))^{\beta/2}$, we have

$$dx_t = \left(-f_t(x_t) + \sigma_t^2\left((1-\beta)\nabla\log q_t(x) + \frac{\beta}{2}\nabla\log q_t^1(x) + \frac{\beta}{2}\nabla\log q_t^2(x)\right)\right)dt + \sigma_t dW_t, \tag{255}$$

$$dw_t = \frac{\sigma_t^2}{2}\left[\left\|(1-\beta)\nabla\log q_t(x) + \frac{\beta}{2}\nabla\log q_t^1(x) + \frac{\beta}{2}\nabla\log q_t^2(x)\right\|^2 - \tag{256}\right.$$

$$\left. - \left((1-\beta)\|\nabla\log q_t(x_t)\|^2 + \frac{\beta}{2}\|\nabla\log q_t^1(x_t)\|^2 + \frac{\beta}{2}\|\nabla\log q_t^2(x_t)\|^2\right)\right]dt. \tag{257}$$

$\square$

---

**Proposition D.6** (Reward-tilted SDE). *Consider the following SDE*

$$dx_t = v_t(x)dt + \sigma_t dW_t, \tag{258}$$

*which samples from the marginals $q_t(x)$. The samples from the marginals $p_t(x) \propto q_t(x)\exp(\beta_t r(x))$ can be simulated according to the following SDE*

$$dx_t = (v_t(x_t) + a\nabla r(x_t))dt + \sigma_t dW_t, \tag{259}$$

$$dw_t = \left[\left\langle\nabla r(x_t), \beta_t(v_t(x_t) - \sigma_t^2\nabla\log q_t(x_t) - \frac{\sigma_t^2}{2}\beta_t\nabla r(x_t)) + a(\nabla\log q_t(x_t) + \beta_t\nabla r(x))\right\rangle + \tag{260}\right.$$

$$\left. + \left(a - \beta_t\frac{\sigma_t^2}{2}\right)\Delta r(x_t) + \frac{\partial\beta_t}{\partial t}r(x_t)\right]dt. \tag{261}$$

*For the reverse-time SDE with drift $v_t(x_t) = -f_t(x_t) + \sigma_t^2\nabla\log q_t(x_t)$ corresponding to the original diffusion generative model and $a = \beta_t\sigma_t^2/2$, we obtain the following weighted SDE*

$$dx_t = (-f_t(x_t) + \sigma_t^2\nabla\log q_t(x_t) + \beta_t\frac{\sigma_t^2}{2}\nabla r(x_t))dt + \sigma_t dW_t, \tag{262}$$

$$dw_t = \left[\frac{\partial\beta_t}{\partial t}r(x) + \left\langle\beta_t\nabla r(x), \frac{\sigma_t^2}{2}\nabla\log q_t(x) - f_t(x)\right\rangle\right]dt \tag{263}$$

---

*Proof.* First, consider the density $q_t(x)$ that follows the PDE

$$\frac{\partial q_t(x)}{\partial t} = -\langle\nabla, q_t(x)v_t(x)\rangle + \frac{\sigma_t^2}{2}\Delta q_t(x). \tag{264}$$

We want to find the PDE for the reward-tilted density

$$p_t(x) = \frac{q_t(x)\exp(\beta_t r(x))}{\int dx\, q_t(x)\exp(\beta_t r(x))}. \tag{265}$$

Straightforwardly, we get

$$\frac{\partial}{\partial t}\log p_t(x) = \frac{\partial}{\partial t}\log q_t(x) + \frac{\partial\beta_t}{\partial t}r(x) - \int dx\, p_t(x)\left[\frac{\partial}{\partial t}\log q_t(x) + \frac{\partial\beta_t}{\partial t}r(x)\right] \tag{266}$$

For the first term, we have

$$\frac{\partial}{\partial t}\log q_t(x) = -\langle\nabla, v_t(x)\rangle - \langle\nabla\log q_t(x), v_t(x)\rangle + \frac{\sigma_t^2}{2}\Delta\log q_t(x) + \frac{\sigma_t^2}{2}\|\nabla\log q_t(x)\|^2 \tag{267}$$

$$= -\langle\nabla, v_t(x)\rangle - \langle\nabla\log p_t(x), v_t(x)\rangle + \frac{\sigma_t^2}{2}\Delta\log p_t(x) + \frac{\sigma_t^2}{2}\|\nabla\log p_t(x)\|^2 + \tag{268}$$

$$+ \left\langle\beta_t\nabla r(x), v_t(x) - \sigma_t^2\nabla\log q_t(x) - \frac{\sigma_t^2}{2}\beta_t\nabla r(x)\right\rangle - \beta_t\frac{\sigma_t^2}{2}\Delta r(x).$$

Thus, we have

$$\frac{\partial p_t(x)}{\partial t} = -\left\langle \nabla, p_t(x)v_t(x) \right\rangle + \frac{\sigma_t^2}{2}\Delta p_t(x) + p_t(x)\big(g_t(x) - \mathbb{E}_{p_t(x)}g_t(x)\big) \tag{269}$$

$$g_t(x) = \left\langle \beta_t\nabla r(x), v_t(x) - \sigma_t^2\nabla \log q_t(x) - \frac{\sigma_t^2}{2}\beta_t\nabla r(x) \right\rangle - \beta_t\frac{\sigma_t^2}{2}\Delta r(x) + \frac{\partial \beta_t}{\partial t}r(x)\,. \tag{270}$$

Furthermore, we can add the gradient of the reward as additional drift term $a\nabla r(x)$, i.e.

$$\frac{\partial p_t(x)}{\partial t} = -\left\langle \nabla, p_t(x)(v_t(x) + a\nabla r(x)) \right\rangle + \frac{\sigma_t^2}{2}\Delta p_t(x) + p_t(x)\big(g_t(x) - \mathbb{E}_{p_t(x)}g_t(x)\big) \tag{271}$$

$$g_t(x) = a\Delta r(x) + a\left\langle \nabla \log p_t(x), \nabla r(x) \right\rangle - \beta_t\frac{\sigma_t^2}{2}\Delta r(x) + \frac{\partial \beta_t}{\partial t}r(x) + \tag{272}$$

$$+ \left\langle \beta_t\nabla r(x), v_t(x) - \sigma_t^2\nabla \log q_t(x) - \frac{\sigma_t^2}{2}\beta_t\nabla r(x) \right\rangle.$$

Taking $v_t(x) = -f_t(x) + \sigma_t^2\nabla \log q_t(x)$ and $a = \beta_t\sigma_t^2/2$, we have

$$\frac{\partial p_t(x)}{\partial t} = -\left\langle \nabla, p_t(x)(-f_t(x) + \sigma_t^2\nabla \log q_t(x) + \beta_t\frac{\sigma_t^2}{2}\nabla r(x)) \right\rangle + \frac{\sigma_t^2}{2}\Delta p_t(x) + p_t(x)\big(g_t(x) - \mathbb{E}_{p_t(x)}g_t(x)\big)$$

$$g_t(x) = \left\langle \beta_t\nabla r(x), \frac{\sigma_t^2}{2}\nabla \log p_t(x) \right\rangle + \frac{\partial \beta_t}{\partial t}r(x) + \left\langle \beta_t\nabla r(x), -f_t(x) - \frac{\sigma_t^2}{2}\beta_t\nabla r(x) \right\rangle \tag{273}$$

$$= \frac{\partial \beta_t}{\partial t}r(x) + \left\langle \beta_t\nabla r(x), \frac{\sigma_t^2}{2}\nabla \log q_t(x) - f_t(x) \right\rangle. \tag{274}$$

This can be simulated as

$$dx_t = (-f_t(x_t) + \sigma_t^2\nabla \log q_t(x_t) + \beta_t\frac{\sigma_t^2}{2}\nabla r(x_t))dt + \sigma_t dW_t\,, \tag{275}$$

$$dw_t = \left[\frac{\partial \beta_t}{\partial t}r(x) + \left\langle \beta_t\nabla r(x), \frac{\sigma_t^2}{2}\nabla \log q_t(x) - f_t(x) \right\rangle\right]dt \tag{276}$$

$$\square$$

# E. Additional Related Work

**Amortized Sampling**   Recently, there has been renewed interested in learning amortized samplers, and particularly diffusion-based amortized samplers particularly towards molecular systems. Midgley et al. (2023) explored learning a normalizing flow using an $\alpha$-divergence trained with samples using annealed importance sampling (Neal, 2001). Zhang & Chen (2022); Vargas et al. (2023; 2024); Richter & Berner (2024); Akhound-Sadegh et al. (2024); Albergo & Vanden-Eijnden (2024); De Bortoli et al. (2024) learn diffusion annealed bridges between distributions using various methods.

While we use DEM in this work as it achieves state of the art results for our LJ-13 setting, there are several works that build upon DEM using bootstrapping (OuYang et al., 2024) and learning the energy function instead of the score (Woo & Ahn, 2024). We note that our FKC approach can be applied to *any* diffusion based sampler.

**(Wasserstein)-Fisher-Rao Gradient Flows**   The reweighting portion of our Feynman-Kac weighted SDEs corresponds to a non-parametric Fisher-Rao gradient flow of a linear functional $\mathcal{G}[p_t] = \int g_t\, p_t dx$, whereas gradient flows in the Wasserstein Fisher-Rao metric (Kondratyev et al., 2016; Chizat et al., 2018; Liero et al., 2018) have a form similar to our weighted PDEs (Lu et al., 2019) for an appropriate ODE simulation term $v_t = \nabla g_t$. In sampling applications, Chemseddine et al. (2025) study the problem of when a given tangent direction in the Fisher-Rao space can be simulated using transport via a tangent direction in the Wasserstein space.

# F. Additional Experimental Details and Results

### F.1. Sampling Metrics

We use a number of metrics to asses the quality of generated samples. These metrics capture different aspects of the distribution. Before computing metrics we filter out samples with energy $> 100$. This only affects non-resampled metrics

and prevents numerical. We find this filters out no samples for DEM or with FKC, and filters less than 3% of samples with target score SDE or tempered noise SDE sampling. We justify this as it is easy to set these filters for generated samples of very poor quality.

**Distance-$\mathcal{W}_2$** For the LJ-13 task we compute the 2-Wasserstein distance between pairwise distance histograms. For this metric we take all pairwise distances for all samples and flatten them into a single distribution. For a sample of 10,000 points this leads to distributions of size 700,000 as there are 70 pairwise distances for a 13 particle system. This is useful metric as it is equivariant and measures the global fidelity of the generated samples. It however is not useful for assessing fine grained details of the generated samples. For that we turn to the Energy-$\mathcal{W}_{1/2}$ distances.

**Energy-$\mathcal{W}_{1/2}$** The Energy-$\mathcal{W}_1$ and Energy-$\mathcal{W}_2$ measures the deviation in the energy value distribution of samples from the reference distribution and the generated distribution. We find this metric is useful to assess the overall fit of a model, although it cannot assess whether a sampler drops modes well. A model that has a reasonably small Energy Wasserstein distance may still have missed a mode of a similar energy value.

We note that for the LJ-13 task we exclude samples with energy $> 100$ for all methods and metrics. In practice this only affects Target Score and Tempered Noise SDEs without FKC and DEM trained at lower temperatures. This excludes roughly 2-3% of samples for those models, which helps these baselines.

**Maximum Mean Discrepancy (MMD)** We use a radial-basis function MMD with multiple scales to assess distribution fit. This measures how well the reference distribution matches the generated distribution locally.

**Total Variation distance** For low dimensional sampling problems, it is useful to consider the total variation distance between empirical distributions that are discretized into a grid. This measures fit in terms of density, ignoring the underlying metric, and is less sensitive to global reweighting of modes.

**1-Wasserstein and 2-Wasserstein distances ($\mathcal{W}_1$ / $\mathcal{W}_2$)** On 40 GMM we also measure the 1-Wasserstein and 2-Wasserstein distances between the generated and reference distributions with respect to the Euclidean metric. We note that while this is possible to measure in the LJ-13 case, it is not as useful as particles in the LJ-13 setting are SE(3) equivariant, and therefore the Euclidean distance is not a suitable ground metric.

## F.2. Mixture of 40 Gaussians

The mixture of 40 Gaussians setting is a 2D energy function with 40 randomly initialized modes with equal standard deviation. This serves as a useful experimental setting where we are able to calculate true densities and scores efficiently without modelling error.

### F.2.1. ADDITIONAL RESULTS

We include quantitative results for the tractable GMM example in Sec. 5.2, where we start at temperature $T_L = 3$ and anneal to target temperature $T_S = 1/3$. We used a geometric noise schedule with $\sigma_{\min} = 0.01$ and $\sigma_{\max} = 500$. We sample 10k samples with 1000 integration steps, with $dt = 0.001$. We observe that Target Score sampling ($a = 0$) from Eq. (22) with systematic resampling performs best in more metrics. We also use this example as an ablation study for the impact of the resampling scheme, where we find that systematic resampling appears to outperform the birth-death exponential clocks implementation of the jump process resampling. See Sec. 4 and App. B.2.

**On ground truth $q_t^\beta$** A subtle point to note is that $q_t^{T_L}$ is not a mixture of $|\pi|$ Gaussians, but rather $|\pi|^{T_L}$ Gaussians for integer $T_L$. This means that we are restricted to small integer $T_L$. We use $T_L = 3$ for all experiments in the 40 Gaussians setting. Note that we reserve $\beta = T_S/T_L$ for the ratio of learning and sampling/target temperatures.

## F.3. LJ-13 Sampling Task

**The Lennard-Jones Potential.** The Lennard-Jones (LJ) potential is an intermolecular potential, modelling interactions of non-bonding particles. This system is studied to evaluate the performance of various neural samplers. The energy for the system is based on the interatomic distance between the particles is given by:

$$\mathcal{E}^{\mathrm{LJ}}(x) = \frac{\varepsilon}{2\tau} \sum_{ij} \left( \left( \frac{r_m}{d_{ij}} \right)^6 - \left( \frac{r_m}{d_{ij}} \right)^{12} \right) \tag{277}$$

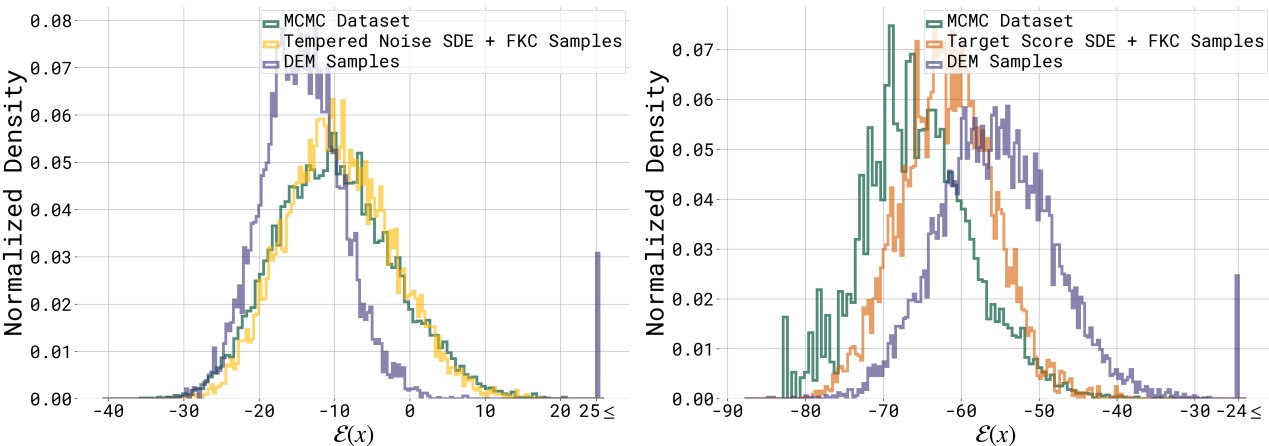

*Figure A1.* Comparison between the energy distribution of the MCMC dataset, samples generated using a DEM model trained at the target temperature, and samples generated using temperature annealing from a model trained at starting distribution $T = 2$. **Left:** the target temperature is 1.5 and **Right**: the target temperature is 0.8.

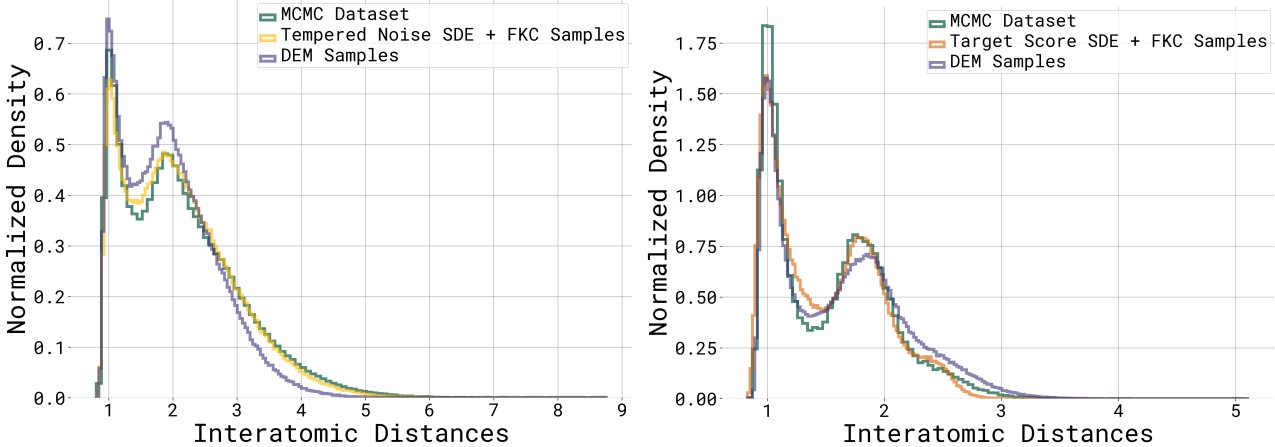

*Figure A2.* Comparison between the distribution of the interatomic distances of the particles in the MCMC dataset, samples generated using a DEM model trained at the target temperature, and samples generated using temperature annealing from a model trained at starting distribution $T = 2$. **Left:** the target temperature is 1.5 and **Right**: the target temperature is 0.8.

*Table A1.* Mixture of 40 Gaussians. Sampling from an annealed distribution with inverse temperature $\beta = 3$. Metrics are calculated over 5 runs with 10k samples.

| SDE Type | FKC | Energy-$\mathcal{W}_2$ | MMD | Total Var | $\mathcal{W}_1$ | $\mathcal{W}_2$ |
|---|---|---|---|---|---|---|
| Target Score | ✗ | $0.943 \pm 0.026$ | $0.020 \pm 0.001$ | $0.487 \pm 0.007$ | $11.304 \pm 0.296$ | $15.671 \pm 0.269$ |
| Tempered Noise | ✗ | $1.032 \pm 0.012$ | $0.058 \pm 0.001$ | $0.638 \pm 0.002$ | $16.051 \pm 0.123$ | $19.627 \pm 0.101$ |
| Target Score | ✓ BDC | $1.064 \pm 0.369$ | $0.010 \pm 0.004$ | $0.402 \pm 0.029$ | $7.797 \pm 3.990$ | $12.451 \pm 5.417$ |
| Tempered Noise | ✓ BDC | $1.228 \pm 0.401$ | $0.056 \pm 0.029$ | $0.572 \pm 0.055$ | $12.598 \pm 4.155$ | $17.679 \pm 4.178$ |
| Target Score | ✓ systematic | $1.098 \pm 0.418$ | $\mathbf{0.007 \pm 0.005}$ | $\mathbf{0.372 \pm 0.020}$ | $\mathbf{6.256 \pm 3.960}$ | $\mathbf{11.265 \pm 5.629}$ |
| Tempered Noise | ✓ systematic | $\mathbf{0.926 \pm 0.248}$ | $0.027 \pm 0.011$ | $0.512 \pm 0.017$ | $9.974 \pm 1.229$ | $14.045 \pm 1.308$ |

*Table A2.* Additional results for LJ-13 at different target temperatures. The model is trained at starting temperature $T_L = 2.0$ and metrics are computed over 3 runs. DEM is run for one seed only as the standard-deviation over seeds is negligible.

| Target Temp. | SDE Type | FKC | distance-$\mathcal{W}_2$ | Energy-$\mathcal{W}_1$ | Energy-$\mathcal{W}_2$ |
|---|---|---|---|---|---|
| 0.9 ($\beta$=2.2) | Target Score | ✗ | $0.215 \pm 0.001$ | $13.886 \pm 0.040$ | $14.893 \pm 0.012$ |
| | | ✓ | $\mathbf{0.042 \pm 0.009}$ | $6.218 \pm 0.896$ | $6.259 \pm 0.873$ |
| | Tempered Noise | ✗ | $0.110 \pm 0.016$ | $5.633 \pm 0.090$ | $7.682 \pm 0.585$ |
| | | ✓ | $\mathbf{0.042 \pm 0.004}$ | $\mathbf{4.384 \pm 0.135}$ | $\mathbf{4.530 \pm 0.167}$ |
| | DEM | — | $0.168 \pm$ — | $14.516 \pm$ — | $14.606 \pm$ — |
| 1.0 ($\beta$=2.0) | Target Score | ✗ | $0.221 \pm 0.001$ | $12.915 \pm 0.054$ | $13.558 \pm 0.112$ |
| | | ✓ | $\mathbf{0.039 \pm 0.008}$ | $2.629 \pm 0.665$ | $2.876 \pm 0.548$ |
| | Tempered Noise | ✗ | $0.094 \pm 0.002$ | $5.215 \pm 0.095$ | $6.560 \pm 0.000$ |
| | | ✓ | $0.053 \pm 0.008$ | $3.205 \pm 0.462$ | $3.538 \pm 0.468$ |
| | DEM | — | $0.127 \pm$ — | $\mathbf{1.352 \pm}$ — | $\mathbf{2.050 \pm}$ — |
| 1.2 ($\beta$=1.67) | Target Score | ✗ | $0.234 \pm 0.004$ | $10.414 \pm 0.036$ | $10.910 \pm 0.110$ |
| | | ✓ | $\mathbf{0.026 \pm 0.001}$ | $2.831 \pm 0.155$ | $2.915 \pm 0.074$ |
| | Tempered Noise | ✗ | $0.098 \pm 0.002$ | $4.258 \pm 0.069$ | $5.564 \pm 0.095$ |
| | | ✓ | $0.076 \pm 0.006$ | $\mathbf{1.017 \pm 0.494}$ | $\mathbf{1.300 \pm 0.433}$ |
| | DEM | — | $0.143 \pm$ — | $9.669 \pm$ — | $9.736 \pm$ — |

where we denote the Euclidean distance between two particles $i$ and $j$ by $d_{ij} = \|x_i - x_j\|_2$ and $r_m$, $\tau$, $\epsilon$ and $c$ are physical constants. As in Köhler et al. (2020), we also add a harmonic potential to the energy so that $\mathcal{E}^{LJ-system} = \mathcal{E}^{\mathrm{LJ}}(x) + c\mathcal{E}^{\mathrm{osc}}(x)$ The harmonic potential is given by:

$$\mathcal{E}^{\mathrm{osc}}(x) = \frac{1}{2} \sum_i \|x_i - x_{\mathrm{COM}}\|^2 \tag{278}$$

where $x_{\mathrm{COM}}$ refers to the center of mass of the system. We set $r_m = 1$, $\tau = 1$, $\varepsilon = 2.0$ and $c = 1.0$.

**Training details.** All DEM models are trained for 166 epochs on 4 NVIDIA A100 80GB GPUs. For all models, the best checkpoint with the lowest energy-$\mathcal{W}_2$ is used for inference. The model is an EGNN with the same architecture as in Akhound-Sadegh et al. (2024). Similar to Akhound-Sadegh et al. (2024), we use a geometric noise schedule for all experiments. We set $\sigma_{\min} = 0.01$ and $\sigma_{\max} = 4.0$. We clip the score to a maximum norm of 1000 (per particle). For sampling, we use 1000 integration steps with $dt = 0.001$. For inference with FKC, we assume a Gaussian distribution at time $t_{\mathrm{start}} = 0.99$ and start integration with the annealed SDE and resampling at that time. We found that this helps significantly to reduce the variance of the results over different runs. For visualizations in Figs. A1 and A2, we selected the best run for all methods for consistency.

In line with previous work, we find the DEM scores are noisy at high times, based on the score of the energy. This can be seen from the score estimator in DEM, which depends on the average gradient direction from a normal distribution sampled around $x_t$. The variance of this estimate grows with both time and gradient of the energy. This makes DEM style objective significantly easier to train on smooth energies, as quantified by norm of the score of the energy.

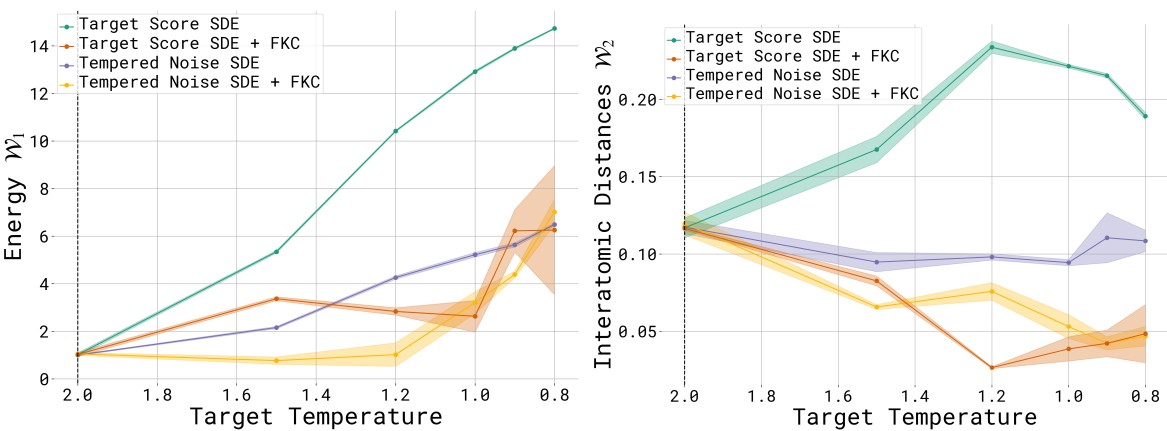

*Figure A3.* **Left:** 1-Wasserstein between energy distributions and **Right:** 2-Wasserstein between distributions of interatomic distances of MCMC samples from the annealed distribution and generated samples.

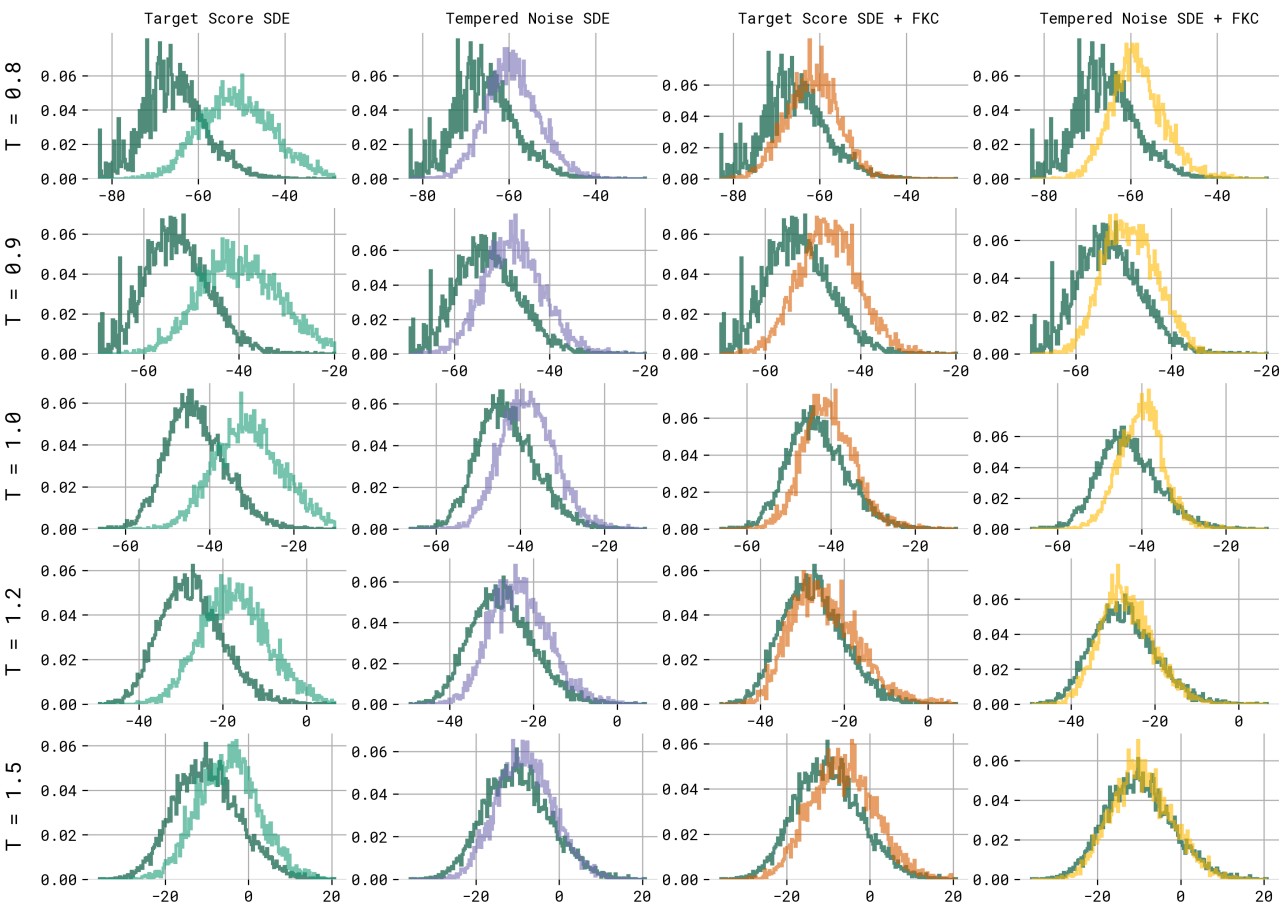

*Figure A4.* Energy distributions of samples generated with temperature annealing compared to the MCMC samples (in dark green), at different target temperatures. The starting temperature is $T_L = 2.0$.

*Table A3.* Comparison of EDM2+FKC (✅) with EDM2+CFG (❌) for image generation using EDM2. With sweep over batchsize $K$. For all metrics, we report CLIP and ImageReward (IR) scores averaged over 10,000 images.

| FKC | $\gamma$ | BS | N | CLIP (↑) | IR (↑) |
|-----|----------|----|----|----------|--------|
| ✅ | 40 | 1 | 32 | 28.75 | $-0.24$ |
| ✅ | 40 | 2 | 32 | 29.03 | $-0.07$ |
| ✅ | 40 | 4 | 32 | 29.11 | 0.02 |
| ✅ | 40 | 8 | 32 | **29.14** | **0.05** |
| ✅ | 40 | 16 | 32 | 29.04 | **0.05** |
| ✅ | 40 | 32 | 32 | 29.00 | 0.04 |
| ✅ | 40 | 64 | 32 | 28.95 | 0.01 |

**Sampling Reference distributions** To generate reference distributions from the Lennard-Jones-13 potential we use Pyro (Bingham et al., 2019) and a No-U-Turn sampler (Hoffman & Gelman, 2014) with default arguments. We use 20k warmup steps and collect 20k samples from the 10 chains for each temperature.

**Additional results** In Table A2 we can see additional results extending Table 3 for intermediate temperatures. Here we can see that generally the same patterns hold with one exception—DEM on a target temperature of 1.0 is better than FKC with $\beta = 2.0$ on Energy-$\mathcal{W}_1$ and Energy-$\mathcal{W}_2$ metrics. This means that DEM has on this temperature has better local fidelity but slightly worse global fidelity. This is quite interesting as we know DEM was originally developed and therefore tuned with a temperature of 1.0 on this dataset. Our hypothesis is that some hyperparameters are specifically tailored to this setting. It is quite interesting then that FKC can still perform better than DEM on global metrics for this temperature.

In Fig. A3 we can see the Energy-$\mathcal{W}_1$ and Interatomic Distance $\mathcal{W}_2$ metrics plotted against the target temperature using a model trained at temperature 2.0. Here we see that FKC performs well across all temperatures for our global metric of interatomic distances and across energy $\mathcal{W}_1$ distances, although at low target temperatures the Energy-$\mathcal{W}_1$ metric gets worse for all methods. We note that this is after excluding roughly 2-3% of samples with unacceptably bad energy from the Target Score SDE and Tempered Noise SDE. Therefore even though the lines are close here, we still prefer the FKC samplers.

### F.4. EDM2 image generation

**Additional experimental details** For all experiments with EDM2 we use the default classifier free guidance weight $\beta = 1.4$. For the $\gamma$ ablation we use batch size 32 for FKC and for the steps experiment we use batch size 8 for FKC.

**Additional Results** In Table A3 we experiment with FKC batch size. Theoretically larger batch sizes should be better as it corresponds to larger sets for the importance sampling step. We see that FKC gets higher scores with larger batch sizes, even with batch size 2 with performance plateauing after batch size $\approx 8$.

### F.5. Latent vs. ambient space image models

We apply the FKC method for generating images with Stable Diffusion-XL (SDXL), a latent diffusion model. We show performance of SDXL+FKC on the GenEval benchmark in Table A4, but we do not observe any significant increase from SDXL with vanilla CFG. While there are examples where FKC improves the semantic accuracy of SD-XL generations, (see Fig. A5), the gain is not consistent when evaluated on 1000 prompts. There multiple potential reasons for this behaviour, e.g. sampling from the geometric average is not better than CFG for these metrics, or the dimensionality of the latent space is too high, resulting in weights with too much variance. Investigating the effectiveness of these methods with latent diffusion models could be an interesting area of future study; in the present study, we find FKC to be consistently helpful when applied to models in the ambient space.

### F.6. Multi-Target Structure-Based Drug Design

**Additional experimental details for main experiments** Following Zhou et al. (2024), we align the target pockets in 3D space and generate sample coordinates for each pocket using an SE(3)-equivariant graph neural network over 1000 integration steps. We then use our Product of Experts (PoE) scheme (Prop. 3.3) to guide ligand generation.

We note that the PoE weight computation in Eq. (27) necessitates equal sample dimensionality, otherwise resampling would be skewed to favor samples of higher dimensions. This requires the molecules within a batch to have the same number of atoms. This motivates our use of fixed size molecule bins for generation.

*Table A4.* Image generation using SDXL with classifier-free guidance (CFG) on the GenEval benchmark (Ghosh et al., 2023). For all metrics mean values are reported. * indicates values directly taken from leaderboard.

| Model | $\beta$ | Overall | Single object | Two object | Counting | Colors | Position | Color attribution |
|---|---|---|---|---|---|---|---|---|
| **GenEval original 553 prompts** | | | | | | | | |
| CLIP retrieval* | – | 0.35 | 0.89 | 0.22 | 0.37 | 0.62 | 0.03 | 0.00 |
| SD-1.5* | – | 0.43 | 0.97 | 0.38 | 0.35 | 0.76 | 0.04 | 0.06 |
| SDXL* | – | 0.55 | 0.98 | 0.74 | 0.39 | 0.85 | 0.15 | 0.23 |
| SDXL (our run) | 7.5 | 0.57 | 0.99 | 0.80 | 0.46 | 0.86 | 0.11 | 0.22 |
| SDXL+FKC | 5.5 | 0.58 | 0.99 | 0.77 | 0.49 | 0.87 | 0.10 | 0.22 |
| SDXL+FKC | 7.5 | 0.57 | 0.99 | 0.78 | 0.46 | 0.83 | 0.13 | 0.23 |
| **GenEval 1000 prompts** | | | | | | | | |
| SDXL (our run) | 7.5 | 0.58 | 0.99 | 0.79 | 0.45 | 0.88 | 0.11 | 0.21 |
| SDXL+FKC | 5.5 | 0.57 | 0.99 | 0.79 | 0.42 | 0.86 | 0.13 | 0.21 |
| SDXL+FKC | 7.5 | 0.57 | 0.99 | 0.80 | 0.45 | 0.83 | 0.13 | 0.22 |

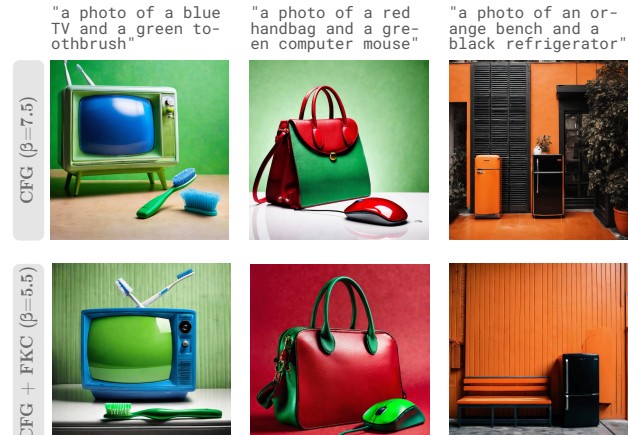

*Figure A5.* Examples where SDXL+FKC (bottom) outperforms EDM2+CFG (top).

Our baseline is the target score SDE with $\beta = 0.5$, which is equivalent to DualDiff from Zhou et al. (2024) and also corresponds to an averaging of scores (Liu et al., 2022). We also generate molecules conditioned on a single protein pocket using TargetDiff from Guan et al. (2023), but dock the molecules to both targets in a protein pair to understand the need for conditioning on two pockets simultaneously.

**SDE Component Analysis** In Table A5, we show the performance of varying the following SDE settings for dual-target drug design: SDE type, $\beta$, and the presence/absece of FKC resampling. Here, we report metrics for generating molecules on a single protein pocket pair (UniProt IDs P23786/P05023) as the validation set. We generate molecules a batch 32 molecules for 5 different molecule lengths, which were sampled from the original training distribution (Guan et al., 2023): $\{15, 19, 23, 27, 35\}$.

We study the impact of the following changing the following settings:

**Inverse temperature ($\beta$)** We find that as we increase $\beta$ from 0.5 to 2.0 the product of the docking scores of the protein pair increases, though the delta increase is larger at smaller $\beta$s.

**FKC.** Next, we try turning FKC on at a fixed $\beta$. We find that performance improves at both $\beta = 1.5$ and $\beta = 2.0$, although the improvement at $\beta = 2.0$ is larger. However, this comes at a cost of diversity and the uniqueness of molecules generated.

**$t_{\max}$** Given that resampling is helpful in terms of improving the quality of the final molecules but decreases molecular diversity, we investigate setting a $t_{\max}$ for our best $\beta$ settings, where we resample only when $\tau <= t_{\max}$. We find that

*Table A5.* Performance of generated molecules with different SDE settings. We generate 32 molecules for 5 molecule sizes for one protein pair for each setting. Lower docking scores are better. Values are reported as averages over all generated molecules in each run. "Better than ref." is the percentage of ligands with better docking scores than known reference molecules for *both* targets (the mean docking score for the reference molecules is $-8.255_{\pm 1.849}$). We also report the diversity, validity & uniqueness, and quality, which refers to the percentage of molecules that are valid, unique, have QED $\geq 0.6$ and SA $\leq 4.0$ (Lee et al., 2025b). Bolded values are the best metrics within each set of midlines. For $\beta = 1$, target score and tempering noise match (Prop. 3.3).

| $\beta$ | FKC | $t_{max}$ | SDE Type | $(P_1 * P_2)$ (↑) | $\max(P_1, P_2)$ (↓) | $P_1$ (↓) | $P_2$ (↓) | Better than ref. (↑) | Div. (↑) | Val. & Uniq. (↑) | Qual. (↑) |
|---|---|---|---|---|---|---|---|---|---|---|---|
| 0.5 | ✗ | — | Target Score | $67.657_{\pm11.985}$ | $-7.667_{\pm0.687}$ | $-8.377_{\pm0.661}$ | $-7.986_{\pm0.948}$ | $0.251_{\pm0.199}$ | $\mathbf{0.886_{\pm0.006}}$ | $0.969_{\pm0.062}$ | $0.244_{\pm0.161}$ |
| 1.0 | ✗ | — | Target Score | $73.366_{\pm14.423}$ | $-7.929_{\pm0.763}$ | $-8.843_{\pm0.899}$ | $-8.174_{\pm0.989}$ | $0.378_{\pm0.311}$ | $0.884_{\pm0.008}$ | $0.962_{\pm0.023}$ | $0.231_{\pm0.170}$ |
| 1.5 | ✗ | — | Target Score | $75.213_{\pm15.779}$ | $-8.085_{\pm0.856}$ | $-8.980_{\pm0.935}$ | $-8.258_{\pm1.024}$ | $\mathbf{0.402_{\pm0.339}}$ | $0.880_{\pm0.012}$ | $0.988_{\pm0.015}$ | $0.250_{\pm0.159}$ |
| 2.0 | ✗ | — | Target Score | $\mathbf{75.551_{\pm16.345}}$ | $\mathbf{-8.089_{\pm0.899}}$ | $\mathbf{-8.966_{\pm0.884}}$ | $\mathbf{-8.309_{\pm1.112}}$ | $0.391_{\pm0.331}$ | $0.881_{\pm0.011}$ | $\mathbf{0.994_{\pm0.012}}$ | $\mathbf{0.288_{\pm0.179}}$ |
| 1.5 | ✗ | — | Target Score | $75.213_{\pm15.779}$ | $\mathbf{-8.085_{\pm0.856}}$ | $\mathbf{-8.980_{\pm0.935}}$ | $-8.258_{\pm1.024}$ | $0.402_{\pm0.339}$ | $\mathbf{0.880_{\pm0.012}}$ | $\mathbf{0.988_{\pm0.015}}$ | $\mathbf{0.250_{\pm0.159}}$ |
| 1.5 | ✓ | — | Target Score | $\mathbf{75.798_{\pm32.984}}$ | $-7.438_{\pm2.507}$ | $-8.582_{\pm3.200}$ | $\mathbf{-8.829_{\pm1.193}}$ | $\mathbf{0.446_{\pm0.454}}$ | $0.651_{\pm0.102}$ | $0.475_{\pm0.169}$ | $0.100_{\pm0.157}$ |
| 2.0 | ✗ | — | Target Score | $75.551_{\pm16.345}$ | $-8.089_{\pm0.899}$ | $-8.966_{\pm0.884}$ | $-8.309_{\pm1.112}$ | $0.391_{\pm0.331}$ | $\mathbf{0.881_{\pm0.011}}$ | $\mathbf{0.994_{\pm0.012}}$ | $\mathbf{0.288_{\pm0.179}}$ |
| 2.0 | ✓ | — | Target Score | $\mathbf{91.845_{\pm28.421}}$ | $\mathbf{-8.977_{\pm1.433}}$ | $\mathbf{-9.984_{\pm1.533}}$ | $\mathbf{-8.978_{\pm1.434}}$ | $\mathbf{0.671_{\pm0.419}}$ | $0.617_{\pm0.049}$ | $0.475_{\pm0.132}$ | $0.044_{\pm0.073}$ |
| 1.5 | ✓ | 0.4 | Target Score | $74.558_{\pm14.361}$ | $-7.961_{\pm0.785}$ | $-8.883_{\pm0.885}$ | $-8.322_{\pm1.083}$ | $0.372_{\pm0.328}$ | $\mathbf{0.883_{\pm0.021}}$ | $0.981_{\pm0.038}$ | $0.262_{\pm0.222}$ |
| 1.5 | ✓ | 0.5 | Target Score | $80.421_{\pm16.567}$ | $-8.365_{\pm0.905}$ | $-9.314_{\pm0.882}$ | $-8.539_{\pm1.077}$ | $0.494_{\pm0.378}$ | $0.867_{\pm0.014}$ | $\mathbf{0.994_{\pm0.012}}$ | $\mathbf{0.288_{\pm0.233}}$ |
| 1.5 | ✓ | 0.6 | Target Score | $\mathbf{83.405_{\pm19.024}}$ | $\mathbf{-8.530_{\pm1.070}}$ | $\mathbf{-9.516_{\pm1.083}}$ | $-8.646_{\pm1.098}$ | $0.485_{\pm0.441}$ | $0.820_{\pm0.024}$ | $0.925_{\pm0.078}$ | $0.244_{\pm0.196}$ |
| 1.5 | ✓ | 0.7 | Target Score | $83.100_{\pm19.354}$ | $-8.434_{\pm0.912}$ | $-9.420_{\pm1.229}$ | $-8.723_{\pm1.227}$ | $\mathbf{0.503_{\pm0.441}}$ | $0.799_{\pm0.030}$ | $0.888_{\pm0.094}$ | $0.162_{\pm0.205}$ |
| 1.5 | ✓ | 1.0 | Target Score | $75.798_{\pm32.984}$ | $-7.438_{\pm2.507}$ | $-8.582_{\pm3.200}$ | $\mathbf{-8.829_{\pm1.193}}$ | $0.446_{\pm0.454}$ | $0.651_{\pm0.102}$ | $0.475_{\pm0.169}$ | $0.100_{\pm0.157}$ |
| 2.0 | ✓ | 0.4 | Target Score | $79.734_{\pm17.631}$ | $-8.331_{\pm0.981}$ | $-9.283_{\pm0.968}$ | $-8.467_{\pm1.133}$ | $0.498_{\pm0.388}$ | $\mathbf{0.876_{\pm0.012}}$ | $\mathbf{0.988_{\pm0.015}}$ | $\mathbf{0.306_{\pm0.234}}$ |
| 2.0 | ✓ | 0.5 | Target Score | $84.949_{\pm19.056}$ | $-8.569_{\pm0.978}$ | $-9.529_{\pm1.018}$ | $-8.796_{\pm1.220}$ | $0.514_{\pm0.423}$ | $0.851_{\pm0.025}$ | $0.938_{\pm0.059}$ | $0.244_{\pm0.160}$ |
| 2.0 | ✓ | 0.6 | Target Score | $87.983_{\pm22.856}$ | $-8.790_{\pm1.231}$ | $-9.619_{\pm1.101}$ | $\mathbf{-8.988_{\pm1.442}}$ | $0.569_{\pm0.468}$ | $0.818_{\pm0.011}$ | $0.888_{\pm0.073}$ | $0.212_{\pm0.223}$ |
| 2.0 | ✓ | 0.7 | Target Score | $85.168_{\pm21.258}$ | $-8.574_{\pm1.097}$ | $-9.514_{\pm1.028}$ | $-8.827_{\pm1.452}$ | $0.593_{\pm0.484}$ | $0.786_{\pm0.034}$ | $0.806_{\pm0.096}$ | $0.231_{\pm0.224}$ |
| 2.0 | ✓ | 1.0 | Target Score | $\mathbf{91.845_{\pm28.421}}$ | $\mathbf{-8.977_{\pm1.433}}$ | $\mathbf{-9.984_{\pm1.533}}$ | $-8.978_{\pm1.434}$ | $\mathbf{0.671_{\pm0.419}}$ | $0.617_{\pm0.049}$ | $0.475_{\pm0.132}$ | $0.044_{\pm0.073}$ |
| 0.5 | ✗ | — | Target Score | $67.657_{\pm11.985}$ | $-7.667_{\pm0.687}$ | $-8.377_{\pm0.661}$ | $-7.986_{\pm0.948}$ | $0.251_{\pm0.199}$ | $0.886_{\pm0.006}$ | $\mathbf{0.969_{\pm0.062}}$ | $0.244_{\pm0.161}$ |
| 0.5 | ✗ | — | Tempered Noise | $\mathbf{71.606_{\pm15.139}}$ | $\mathbf{-7.838_{\pm0.872}}$ | $\mathbf{-8.727_{\pm0.878}}$ | $\mathbf{-8.085_{\pm1.088}}$ | $\mathbf{0.362_{\pm0.274}}$ | $\mathbf{0.887_{\pm0.007}}$ | $0.944_{\pm0.098}$ | $\mathbf{0.300_{\pm0.212}}$ |
| 0.5 | ✓ | 0.6 | Target Score | $77.100_{\pm16.533}$ | $-8.243_{\pm0.949}$ | $-9.112_{\pm0.898}$ | $-8.337_{\pm1.045}$ | $0.417_{\pm0.337}$ | $0.877_{\pm0.008}$ | $\mathbf{0.975_{\pm0.023}}$ | $0.212_{\pm0.155}$ |
| 0.5 | ✓ | 0.6 | Tempered Noise | $\mathbf{78.501_{\pm15.383}}$ | $\mathbf{-8.323_{\pm0.919}}$ | $\mathbf{-9.127_{\pm0.770}}$ | $\mathbf{-8.496_{\pm1.100}}$ | $\mathbf{0.496_{\pm0.308}}$ | $\mathbf{0.879_{\pm0.016}}$ | $0.931_{\pm0.064}$ | $\mathbf{0.250_{\pm0.163}}$ |
| 2.0 | ✗ | — | Target Score | $75.551_{\pm16.345}$ | $\mathbf{-8.089_{\pm0.899}}$ | $-8.966_{\pm0.884}$ | $-8.309_{\pm1.112}$ | $0.391_{\pm0.331}$ | $\mathbf{0.881_{\pm0.011}}$ | $\mathbf{0.994_{\pm0.012}}$ | $\mathbf{0.288_{\pm0.179}}$ |
| 2.0 | ✗ | — | Tempered Noise | $\mathbf{75.868_{\pm16.154}}$ | $-8.045_{\pm0.909}$ | $\mathbf{-8.977_{\pm0.978}}$ | $\mathbf{-8.352_{\pm1.095}}$ | $\mathbf{0.460_{\pm0.390}}$ | $0.874_{\pm0.008}$ | $\mathbf{0.994_{\pm0.012}}$ | $0.262_{\pm0.186}$ |
| 2.0 | ✓ | 0.6 | Target Score | $\mathbf{87.983_{\pm22.856}}$ | $\mathbf{-8.790_{\pm1.231}}$ | $\mathbf{-9.619_{\pm1.101}}$ | $\mathbf{-8.988_{\pm1.442}}$ | $\mathbf{0.569_{\pm0.468}}$ | $\mathbf{0.818_{\pm0.011}}$ | $\mathbf{0.888_{\pm0.073}}$ | $0.212_{\pm0.223}$ |
| 2.0 | ✓ | 0.6 | Tempered Noise | $79.696_{\pm18.087}$ | $-8.229_{\pm0.997}$ | $-9.104_{\pm0.921}$ | $-8.681_{\pm1.481}$ | $0.464_{\pm0.429}$ | $0.796_{\pm0.038}$ | $0.838_{\pm0.109}$ | $\mathbf{0.331_{\pm0.274}}$ |
| 2.0 | ✓ | 0.7 | Target Score | $\mathbf{85.168_{\pm21.258}}$ | $\mathbf{-8.574_{\pm1.097}}$ | $-9.514_{\pm1.028}$ | $\mathbf{-8.827_{\pm1.452}}$ | $\mathbf{0.593_{\pm0.484}}$ | $0.786_{\pm0.034}$ | $0.806_{\pm0.096}$ | $0.231_{\pm0.224}$ |
| 2.0 | ✓ | 0.7 | Tempered Noise | $84.969_{\pm18.906}$ | $-8.531_{\pm0.980}$ | $\mathbf{-9.809_{\pm1.170}}$ | $-8.545_{\pm0.991}$ | $0.589_{\pm0.427}$ | $\mathbf{0.796_{\pm0.035}}$ | $\mathbf{0.850_{\pm0.121}}$ | $\mathbf{0.262_{\pm0.154}}$ |

setting $t_{max}$ to a value in $[0.5, 0.7]$ generates molecules that are higher in quality compared to always resampling or no resampling for $\beta = 1.5$. For $\beta = 2.0$, the performance slightly decreases, but the diversity and uniqueness of the molecules is much higher at the end. Setting $t_{max}$ to 0.6 gives a good tradeoff in terms of generating molecules that perform well vs. maintaining diversity, and so we proceed with $\beta = 2.0$ and $t_{max} = 0.6$ for the final experiments.

**SDE Type** Finally, we try using different types of SDEs. We find that at lower $\beta$, the Tempered Noise SDE performs better with and without FKC. At higher $\beta$, however, using the Tempered Noise SDE does not significantly change performance or decreases performance. Thus, for the main experiments, we proceed with the Target Score SDE.

**Visualizing docked molecules** In Fig. A6, we visualize the molecules with the highest docking scores to the protein pair GRM5/RRM1 (UniProt IDs P41594 and P23921, respectively) at each molecule size.

**Correlation between FKC weights and target metric** In Fig. A7, we plot the Spearman rank correlation of FKC weights and docking scores of the final generated molecules for two protein pair examples to understand whether sampling according to these weights will select for better molecules. We observe that for a protein pair where molecules improve with FKC resampling, there tends to be a positive correlation between the FKC weight and docking score (avg. $\rho = 0.275$ for an example protein pair), while this trend is less clear for a case where FKC resampling does not improve the molecule (avg. $\rho = 0.087$ for an example protein pair).

### F.7. Molecule SMILES generation using latent diffusion models

We also investigate generating molecular SMILES strings conditioned on functional properties, which describe the desired function that the molecule should have. Molecules often need to possess multiple properties (e.g. bind to protein $X$ and be non-toxic) (Wang et al., 2025). By controlling for these properties during the molecular generation process (as opposed to post-hoc filtering), we aim to increase the probability of discovering molecules that exhibit all desired characteristics, thereby improving the efficiency of hit identification.

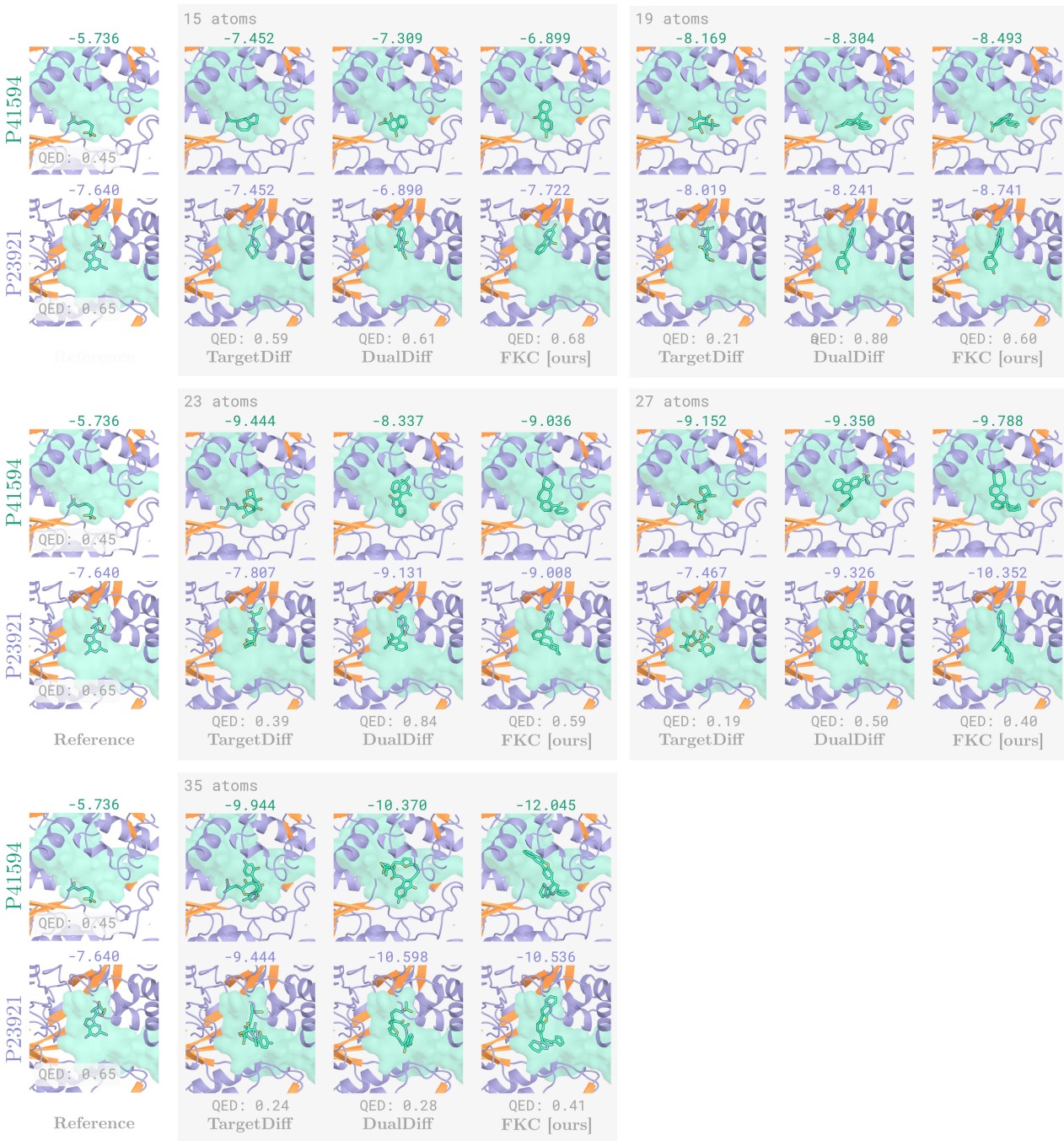

*Figure A6.* Molecules generated from our method (target score SDE with $\beta = 2.0$ and FKC resampling) and baselines in the binding pockets of two proteins: GRM5 (UniProt ID P41594) and RRM1 (UniProt ID P23921) for all 5 molecule sizes considered ($\{15, 19, 23, 27, 35\}$ atoms). Docking scores for each molecule and target are above each image; lower docking scores are better. The QED of the molecule is above each model name. The binding pocket is shaded in light green.

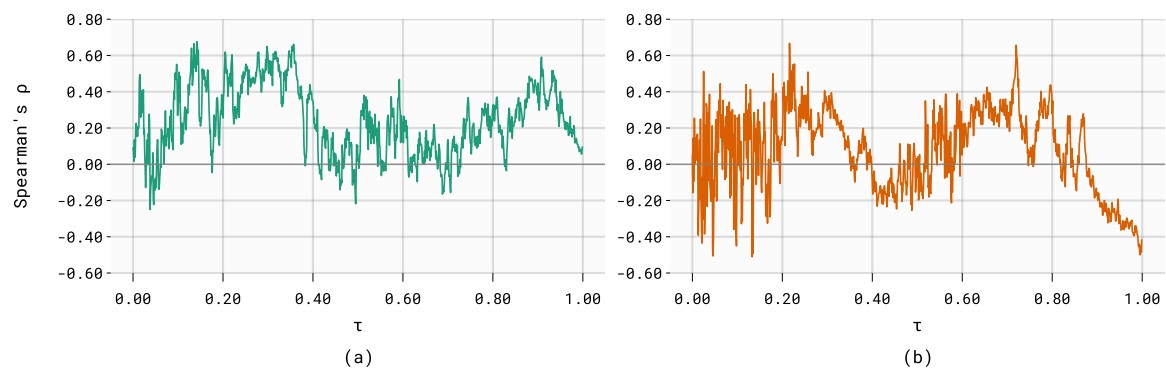

*Figure A7.* Spearman rank correlation coefficient ($\rho$) of FKC weights computed at each timestep $\tau$ and the docking score of the final molecule for (a) a protein pair where the final generated molecule improves with FKC and (b) a failure case where the molecule has a worse docking score than the baseline. Note that during inference, we only resample for $\tau \in [0.0, 0.6]$.

*Table A6.* Multi-property molecule generation results (PoE). For a set of two target properties ($P_1$ and $P_2$), we take the set of the top-10 best performing molecules from a batch-size of 512 as the molecules with the highest $P_1*P_2$ scores. We report averages of the top-10 molecules from 5 runs and the top-1 molecule overall. We also report the diversity, validity & uniqueness, and quality of all molecules.

| $P_1$ / $P_2$ | SDE Type | $\beta$ | FKC | $P_1$ top-10 ($\uparrow$) | $P_2$ top-10 ($\uparrow$) | ($P_1$, $P_2$) top-1 ($\uparrow$) | Div. ($\uparrow$) | Val. & Uniq. ($\uparrow$) | Qual. ($\uparrow$) |
|---|---|---|---|---|---|---|---|---|---|
| JNK3 GSK3$\beta$ | Target Score | 0.5 | ✗ | $0.212_{\pm 0.016}$ | $0.356_{\pm 0.046}$ | $(0.500, 0.580)$ | $\mathbf{0.910}_{\pm \mathbf{0.000}}$ | $0.713_{\pm 0.027}$ | $0.127_{\pm 0.015}$ |
| | Tempered Noise | 1.5 | ✗ | $0.341_{\pm 0.039}$ | $0.468_{\pm 0.041}$ | $(0.590, 0.560)$ | $0.881_{\pm 0.002}$ | $0.813_{\pm 0.025}$ | $0.352_{\pm 0.012}$ |
| | | | ✓ | $\mathbf{0.342}_{\pm \mathbf{0.012}}$ | $\mathbf{0.502}_{\pm \mathbf{0.034}}$ | $(\mathbf{0.500}, \mathbf{0.720})$ | $0.882_{\pm 0.002}$ | $\mathbf{0.832}_{\pm \mathbf{0.021}}$ | $\mathbf{0.360}_{\pm \mathbf{0.021}}$ |
| JNK3 DRD2 | Target Score | 0.5 | ✗ | $0.090_{\pm 0.018}$ | $0.434_{\pm 0.065}$ | $(0.150, 0.472)$ | $\mathbf{0.915}_{\pm \mathbf{0.001}}$ | $\mathbf{0.671}_{\pm \mathbf{0.022}}$ | $0.228_{\pm 0.011}$ |
| | Tempered Noise | 1.5 | ✗ | $0.132_{\pm 0.032}$ | $0.550_{\pm 0.036}$ | $(0.280, 0.469)$ | $0.884_{\pm 0.001}$ | $0.650_{\pm 0.021}$ | $\mathbf{0.258}_{\pm \mathbf{0.020}}$ |
| | | | ✓ | $\mathbf{0.141}_{\pm \mathbf{0.020}}$ | $\mathbf{0.617}_{\pm \mathbf{0.040}}$ | $(\mathbf{0.360}, \mathbf{0.655})$ | $0.884_{\pm 0.005}$ | $0.661_{\pm 0.018}$ | $0.252_{\pm 0.014}$ |
| GSK3$\beta$ DRD2 | Target Score | 0.5 | ✗ | $0.146_{\pm 0.034}$ | $0.528_{\pm 0.077}$ | $(0.051, 0.908)$ | $\mathbf{0.914}_{\pm \mathbf{0.001}}$ | $0.709_{\pm 0.021}$ | $0.203_{\pm 0.015}$ |
| | Tempered Noise | 1.5 | ✗ | $0.228_{\pm 0.016}$ | $\mathbf{0.649}_{\pm \mathbf{0.084}}$ | $(0.550, 0.655)$ | $0.884_{\pm 0.002}$ | $\mathbf{0.774}_{\pm \mathbf{0.015}}$ | $0.303_{\pm 0.012}$ |
| | | | ✓ | $\mathbf{0.266}_{\pm \mathbf{0.061}}$ | $0.638_{\pm 0.036}$ | $(\mathbf{0.520}, \mathbf{0.796})$ | $0.885_{\pm 0.002}$ | $0.774_{\pm 0.017}$ | $\mathbf{0.307}_{\pm \mathbf{0.012}}$ |

**Model**   We select LDMol (Chang & Ye, 2024) to generate molecules, which is a latent diffusion model conditioned on natural language descriptions of molecule properties; this gives flexibility of generating molecules with a wide range of properties.

**Experiment setup: TDC oracles**   We consider three proteins oracles from Therpeutic Data Commons (TDC) (Huang et al., 2021): JNK3, GSK3$\beta$, DRD2, which predicts whether or not a molecule binds to a protein. Note that while this task is similar in nature to the objective of SBDD, we are generating molecules conditioned on a functional text description instead of a 3D protein pocket. However, we could also consider other functional property descriptions, such as molecular solubility, toxicity, etc.

**Prompts**   To generate molecules that inhibit a specific protein, we prompt the model with "`This molecule inhibits {protein_name}`", following Wang et al. (2025).

**Metrics**   In addition to reporting the top-performing molecules, we report the percent of molecules that are valid *and* unique, as well as their diversity (evaluated using Tanimoto distance on Morgan fingerprints (Rogers & Hahn, 2010)) and quality, which is the set of unique and valid molecules that also have a quantitative estimate of drug-likeness (QED) $\geq 0.6$ and synthetic accessibility (SA) $\leq 4.0$. This metric was taken from Lee et al. (2025b).

**Results: TDC oracles**   We aim to generate molecules that satisfy the function of binding each protein when taking all combinations of the protein pairs. In Table A6, we show the best performance for each set of molecules and in Table A8 we ablate different SDE components. We find that the tempered noise SDE at higher $\beta$ generates molecules that have higher fitness for binding to each pair of proteins. When we incorporate FKC, the average performance of the molecules further increases. We also note that PoE+FKC tends to generate more molecules that are unique, valid and are higher drug-like quality, although their diversity decreases slightly, which is a common tradeoff. In practice, we find that the FKC weights with the latent diffusion model have a large variance during molecule generation. This is problematic, as a large number of

*Table A7.* Docking scores of generated molecules to $P_1$=ATP1A1 and $P_2$=CPT2. We used the Tempered Noise SDE with $\beta = 1.5$ and generated 32 molecules.

| FKC | $(P_1, P_2)$ top-10 ($\downarrow$) | $(P_1, P_2)$ top-1 ($\downarrow$) | Div. ($\uparrow$) |
|---|---|---|---|
| ❌ | $-6.65_{\pm1.05}, -7.36_{\pm0.854}$ | $(-8.87, -8.13)$ | **0.921** |
| ✅ | $(-\mathbf{7.49}_{\pm\mathbf{0.71}}, -\mathbf{8.31}_{\pm\mathbf{0.94}})$ | $(-\mathbf{8.41}, -\mathbf{9.73})$ | $0.895$ |

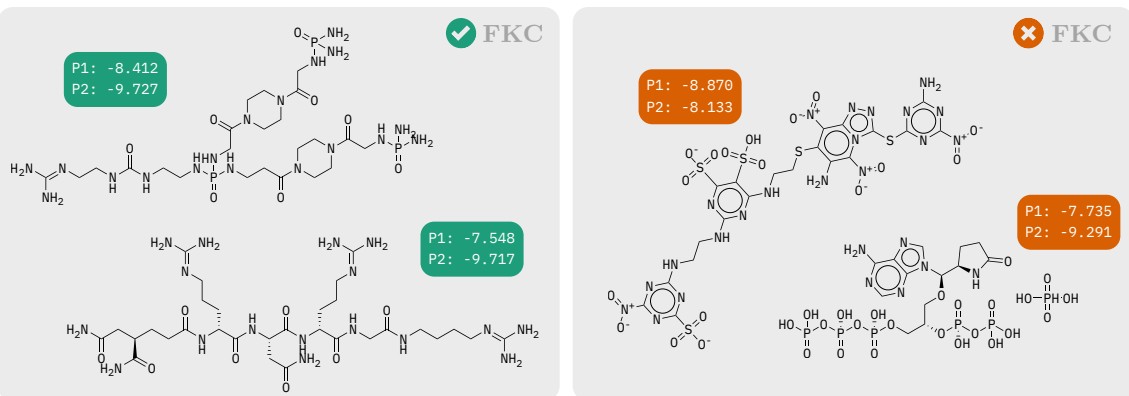

*Figure A8.* Molecules with best docking scores for binding to ATPA1 ($P_1$) and CPT2 ($P_2$) from PoE with FKC (left) and without (right).

samples are thrown away. Furthermore, we noted that the score was not always well-conditioned. To ameliorate this, for all experiments using LDMol, we divided the weights by a set temperature term ($T = 100$) to reduce their variance before resampling, clipped the top $20\%$ to account for any score instabilities, and did early-stopping (only resampled for $70\%$ of the timesteps).

**Experiment setup: protein docking**    Finally, we consider a more challenging setting of protein-ligand docking, where we generate molecules using LDMol based on text-based prompts of binding to the proteins ATP1A1 (UniProt ID P05023) and CPT2 (UniProt ID P23786), and then evaluate them using docking. The protein pockets were obtained from Zhou et al. (2024) and the final generated molecules were docked using AutoDock Vina (Eberhardt et al., 2021).

**Results: protein docking**    Table A7 shows the docking scores of molecules, and we find that incorporating FKC generates molecules with better scores. While ligands are typically generated using SBDD, we find it interesting that text-prompt generation is able to produce molecules that have reasonably good docking scores; known binders to ATP1A1 and CPT2 have docking scores of -8.168 and -9.174, respectively (Zhou et al., 2024). We visualize the top molecules in Fig. A8.

*Table A8.* Multi-property molecule generation results. For a set of two target properties ($P_1$ and $P_2$), we take the set of the top-10 best performing molecules as the molecules with the highest $P_1*P_2$ scores. We report the average properties of the top-10 molecules over five runs and the top-1 molecule overall. We also report the diversity, validity & uniqueness, and quality of all generated molecules, where quality is the percent of molecules that are valid, unique, have a QED $\geq 0.6$ and SA $< 0.4$. For $\beta = 1$, target score and tempering noise match (Prop. 3.3).

| $P_1$ $P_2$ | SDE Type | $\beta$ | FKC | $P_1$ top-10 ($\uparrow$) | $P_2$ top-10 ($\uparrow$) | ($P_1$, $P_2$) top-1 ($\uparrow$) | Div. ($\uparrow$) | Val. & Uniq. ($\uparrow$) | Qual. ($\uparrow$) |
|---|---|---|---|---|---|---|---|---|---|
| JNK3 GSK3$\beta$ | Target Score | 0.5 | ✗ | $0.212_{\pm0.016}$ | $0.356_{\pm0.046}$ | $(0.500, 0.580)$ | $\mathbf{0.910_{\pm0.000}}$ | $0.713_{\pm0.027}$ | $0.127_{\pm0.015}$ |
| | Tempered Noise | 0.5 | ✗ | $\mathbf{0.225_{\pm0.028}}$ | $\mathbf{0.385_{\pm0.042}}$ | $(\mathbf{0.440, 0.690})$ | $0.909_{\pm0.001}$ | $\mathbf{0.723_{\pm0.016}}$ | $\mathbf{0.134_{\pm0.006}}$ |
| | — | 1.0 | ✗ | $0.289_{\pm0.022}$ | $0.429_{\pm0.018}$ | $(0.470, 0.580)$ | $\mathbf{0.898_{\pm0.002}}$ | $0.811_{\pm0.008}$ | $0.205_{\pm0.011}$ |
| | — | 1.0 | ✓ | $\mathbf{0.342_{\pm0.029}}$ | $\mathbf{0.442_{\pm0.051}}$ | $(\mathbf{0.600, 0.650})$ | $0.897_{\pm0.002}$ | $0.804_{\pm0.015}$ | $\mathbf{0.205_{\pm0.015}}$ |
| | Target Score | 1.5 | ✗ | $0.336_{\pm0.031}$ | $0.484_{\pm0.052}$ | $(0.480, 0.780)$ | $\mathbf{0.886_{\pm0.003}}$ | $0.816_{\pm0.013}$ | $0.336_{\pm0.022}$ |
| | Target Score | 1.5 | ✓ | $\mathbf{0.351_{\pm0.0340}}$ | $0.447_{\pm0.026}$ | $(\mathbf{0.590, 0.780})$ | $\mathbf{0.886_{\pm0.003}}$ | $0.823_{\pm0.024}$ | $0.356_{\pm0.037}$ |
| | Tempered Noise | 1.5 | ✗ | $0.341_{\pm0.039}$ | $0.468_{\pm0.041}$ | $(0.590, 0.560)$ | $0.881_{\pm0.002}$ | $0.813_{\pm0.025}$ | $0.352_{\pm0.012}$ |
| | Tempered Noise | 1.5 | ✓ | $0.342_{\pm0.012}$ | $\mathbf{0.502_{\pm0.034}}$ | $(0.500, 0.720)$ | $0.882_{\pm0.002}$ | $\mathbf{0.832_{\pm0.021}}$ | $\mathbf{0.360_{\pm0.021}}$ |
| JNK3 DRD2 | Target Score | 0.5 | ✗ | $\mathbf{0.090_{\pm0.018}}$ | $0.434_{\pm0.065}$ | $(0.150, 0.472)$ | $\mathbf{0.915_{\pm0.001}}$ | $0.671_{\pm0.022}$ | $0.228_{\pm0.011}$ |
| | Tempered Score | 0.5 | ✗ | $0.066_{\pm0.015}$ | $\mathbf{0.571_{\pm0.187}}$ | $(\mathbf{0.110, 0.943})$ | $0.914_{\pm0.002}$ | $\mathbf{0.678_{\pm0.0187}}$ | $\mathbf{0.236_{\pm0.020}}$ |
| | — | 1.0 | ✗ | $0.087_{\pm0.028}$ | $0.624_{\pm0.094}$ | $(0.100, 0.978)$ | $\mathbf{0.903_{\pm0.001}}$ | $0.675_{\pm0.022}$ | $0.241_{\pm0.010}$ |
| | — | 1.0 | ✓ | $\mathbf{0.094_{\pm0.024}}$ | $\mathbf{0.635_{\pm0.067}}$ | $(\mathbf{0.413, 0.550})$ | $0.899_{\pm0.002}$ | $\mathbf{0.686_{\pm0.025}}$ | $\mathbf{0.263_{\pm0.023}}$ |
| | Target Score | 1.5 | ✗ | $0.136_{\pm0.046}$ | $0.582_{\pm0.067}$ | $(\mathbf{0.490, 0.640})$ | $\mathbf{0.886_{\pm0.003}}$ | $0.639_{\pm0.019}$ | $0.241_{\pm0.017}$ |
| | Target Score | 1.5 | ✓ | $0.102_{\pm0.031}$ | $\mathbf{0.620_{\pm0.148}}$ | $(0.320, 0.541)$ | $0.885_{\pm0.006}$ | $0.659_{\pm0.022}$ | $\mathbf{0.274_{\pm0.028}}$ |
| | Tempered Noise | 1.5 | ✗ | $0.132_{\pm0.032}$ | $0.550_{\pm0.036}$ | $(0.280, 0.469)$ | $0.884_{\pm0.001}$ | $0.650_{\pm0.021}$ | $0.258_{\pm0.020}$ |
| | Tempered Noise | 1.5 | ✓ | $\mathbf{0.141_{\pm0.020}}$ | $0.617_{\pm0.040}$ | $(0.360, 0.655)$ | $0.884_{\pm0.005}$ | $\mathbf{0.661_{\pm0.018}}$ | $0.252_{\pm0.014}$ |
| GSK3$\beta$ DRD2 | Target Score | 0.5 | ✗ | $0.146_{\pm0.034}$ | $0.528_{\pm0.077}$ | $(0.051, 0.908)$ | $\mathbf{0.914_{\pm0.001}}$ | $\mathbf{0.709_{\pm0.021}}$ | $\mathbf{0.203_{\pm0.015}}$ |
| | Tempered Score | 0.5 | ✗ | $\mathbf{0.162_{\pm0.025}}$ | $\mathbf{0.543_{\pm0.063}}$ | $(\mathbf{0.430, 0.965})$ | $\mathbf{0.914_{\pm0.001}}$ | $0.697_{\pm0.013}$ | $0.198_{\pm0.017}$ |
| | — | 1.0 | ✗ | $\mathbf{0.202_{\pm0.023}}$ | $0.620_{\pm0.057}$ | $(\mathbf{0.660, 0.726})$ | $\mathbf{0.908_{\pm0.002}}$ | $0.773_{\pm0.021}$ | $0.238_{\pm0.021}$ |
| | — | 1.0 | ✓ | $0.190_{\pm0.022}$ | $\mathbf{0.666_{\pm0.093}}$ | $(0.240, 0.986)$ | $0.907_{\pm0.002}$ | $\mathbf{0.784_{\pm0.010}}$ | $\mathbf{0.254_{\pm0.019}}$ |
| | Target Score | 1.5 | ✗ | $0.240_{\pm0.030}$ | $0.636_{\pm0.066}$ | $(0.350, 0.804)$ | $\mathbf{0.894_{\pm0.002}}$ | $0.759_{\pm0.015}$ | $0.290_{\pm0.016}$ |
| | Target Score | 1.5 | ✓ | $0.222_{\pm0.036}$ | $0.584_{\pm0.068}$ | $(0.630, 0.580)$ | $0.891_{\pm0.003}$ | $0.740_{\pm0.027}$ | $0.283_{\pm0.020}$ |
| | Tempered Score | 1.5 | ✗ | $0.228_{\pm0.016}$ | $\mathbf{0.649_{\pm0.084}}$ | $(0.550, 0.655)$ | $0.884_{\pm0.002}$ | $\mathbf{0.774_{\pm0.015}}$ | $0.303_{\pm0.012}$ |
| | Tempered Score | 1.5 | ✓ | $\mathbf{0.266_{\pm0.061}}$ | $0.638_{\pm0.036}$ | $(\mathbf{0.520, 0.796})$ | $0.885_{\pm0.002}$ | $0.774_{\pm0.017}$ | $\mathbf{0.307_{\pm0.012}}$ |

