# OpenReview forum: "Feynman-Kac Correctors in Diffusion: Annealing, Guidance, and Product of Experts"
_ICML.cc/2025/Conference — ICML 2025 spotlightposter_

### Official Review · Reviewer_65RJ · 2025-02-27

**Overall Recommendation:** 3

**Summary:**

This paper proposed that the Feynman-Kac Correctors enhance the sampling of several types of compositional distributions. The main idea is based on the Feynman-Kac formulation and the transformation of transport and diffusion to reweight operation.

## update after rebuttal
My view is not changed, so I maintain my original score.

**Claims And Evidence:**

The method of enhanced sampling from compositional distributions via reweighting is novel and reasonable. The derivation of the method is supported by valid MCMC/PDE theory.

**Essential References Not Discussed:**

no

**Experimental Designs Or Analyses:**

sec 5.1: sound.

sec 5.2: I am not familiar with the molecule generation task.

sec 5.3:
1. The result is not convincing enough. More experimental results on different CFG coefficients are needed. The Pareto curve between CLIP and FID across different beta can indicate the superiority of the proposed method.
2. Tests on large-scale prompts (at least 1k prompts) from standard datasets like MS-COCO are necessary. Tests on other models like PixArt, and SD2b are beneficial.
3. It is not clear why better adherence to the geometric average CFG distribution would contribute to enhancing the image quality.

**Methods And Evaluation Criteria:**

The evaluation on SD-XL is not that convincing, the metric should be tested on large-scale prompts (at least 1k prompts) from standard datasets like MS-COCO.

**Other Comments Or Suggestions:**

no

**Other Strengths And Weaknesses:**

no

**Questions For Authors:**

1. Considering the relation of MCMC and ParVI, it is possible to add particle interactive force to the proposed FKC method to mitigate the sampling covariance?

2. I noticed that the reweight update has a similar form to the Fisher-Rao gradient flow. There may be some underlying theoretical connection.

**Relation To Broader Scientific Literature:**

The proposed method contributes to the understanding of DMs and the potential enhanced performance of various downstream DM tasks.

**Theoretical Claims:**

I checked the theoretical claims. They are solid.

---

> ### Author Rebuttal · Authors · 2025-04-01
>
> We thank the reviewer for the detailed feedback and constructive suggestions. We are glad that they find the proposed idea 'novel' and its derivation 'supported by valid theory'. We are also happy to hear that our method 'contributes to the understanding of Diffusion Models.' We provide the suggested comparisons and answer the reviewer's questions below.
>
> > Testing the metrics on large-scale datasets (at least 1k prompts)
>
> We thank the reviewer for the constructive feedback which has led to some interesting findings and a deeper understanding of this work. We evaluate FKC on two large-scale tasks: on ImageNet-1K using EDM2 [4] and on the Geneval benchmark [1] using SDXL, which evaluates the algorithms on a predefined set of prompts and measures adherence to the prompts. These additional large-scale experiments have led to two new insights:
> 1. **FKC improves image quality and prompt adherence in ambient space but not latent space.**
> Point 1 is supported by the following two tables. We can see substantial improvement in the performance of FKC on an ambient diffusion model but not on a latent diffusion model.
> - **EDM2** We first conduct a new set of image experiments using the EDM2 model [4], which directly generates outputs in pixel-space. We compare generations from the baseline method, which uses CFG, with generations guided by FKC on ImageNet-1K and evaluate 10k samples using two metrics: CLIP Score and ImageReward [5], which reflects how closely they align with human preferences.
>
> ||steps|churn|Clip Score(↑)|ImageReward(↑)|
> |-|-|-|-|-|
> |CFG|32|40|28.75|-0.24|
> |FKC|32|40|29.00|0.04|
> |CFG|48|10|28.83|-0.18|
> |FKC|48|10|**29.02**|0.06|
> |CFG|64|80|28.67|-0.21|
> |FKC|64|80|28.88|**0.09**|
> |CFG|128|40|28.71|-0.19|
> |FKC|128|40|28.94|0.06|
>
> We find that incorporating FKC improves both scores and qualitatively generates better images; we include examples in [this PDF](https://anonymous.4open.science/r/FKC-D7F0/rebuttal.pdf). We use default settings of $\beta=1.4$ for both models.
> - **SDXL Geneval** The table below consists of three parts: Geneval benchmarks, performance of FKC on the prompts from Geneval (553 prompts), and on 1k prompts (553 + newly generated). We also rerun SDXL with the same hyperparameters as our algorithm for a fair comparison.
>
> |Model|$\beta$|Overall|Single object|Two object|Counting|Colors|Position|Color attribution|
> |-|-|-|-|-|-|-|-|-|
> |CLIP retrieval||**0.35**|0.89|0.22|0.37|0.62|0.03|0.00|
> |SD 1.5||**0.43**|0.97|0.38|0.35|0.76|0.04|0.06|
> |SDXL||**0.55**|0.98|0.74|0.39|0.85|0.15|0.23|
> |**Geneval Prompts**|
> |SDXL (Rerun)|7.5|**0.57**|0.99|0.80|0.46|0.86|0.11|0.22|
> |FKC|5.5|**0.58**|0.99|0.77|0.49|0.87|0.10|0.22|
> |FKC|7.5|**0.57**|0.99|0.78|0.46|0.83|0.13|0.23|
> |**1k Prompts**|
> |SDXL|7.5|**0.58**|0.99|0.79|0.45|0.88|0.11|0.21|
> |FKC|5.5|**0.57**|0.99|0.79|0.42|0.86|0.13|0.21|
> |FKC|7.5|**0.57**|0.99|0.80|0.45|0.83|0.13|0.22|
>
> 2. **FKC improves performance in the ambient space across tasks whereas improvements in latent space diffusion models are limited.** This is supported by Table 2 and Figure 2 of [the PDF](https://anonymous.4open.science/r/FKC-D7F0/rebuttal.pdf). Here, we present new and stronger results for the molecule generation task in the coordinate space instead of the latent space.
>
> > Is it possible to add particle interactive force to the proposed FKC method to mitigate the sampling covariance?
>
> Indeed, one could combine the proposed method with the Stein Variational Gradient Descent, targeting the intermediate marginals (annealed density or the product of densities) similar to [2]. This would require adding a term to the weights, which, theoretically, should reduce the variance of the weights.
>
> > Сonnection to the Fisher-Rao gradient flow
>
> The reweighting equation $\partial_t p_t^w(x) = p_t^w(x) \bar{g}_t(x)$ does correspond to the Fisher-Rao gradient flow of a linear functional $G[p_t] = \int {g}_t(x) p_t(x) dx$ (after constraining $p_t^w$ to remain normalized). The Wasserstein Fisher-Rao gradient flow has a form similar to the Feynman-Kac PDE with $\sigma_t=0$ (deterministic evolution). We have included a note about this in the Appendix of the updated revision (see e.g. Thm. 3.1 in [3]).
>
> ### Closing remarks
>
> We thank the reviewer for suggesting this comparison, which significantly improves our empirical study. We hope that our answers address all the important questions raised by the reviewer, and we are more than happy to consider any additional questions or further suggestions.
>
> [1] Ghosh, Dhruba et al. "Geneval: An object-focused framework for evaluating text-to-image alignment."
> [2] Corso, Gabriele et. al. "Particle guidance: non-iid diverse sampling with diffusion models."
> [3] Lu, Yulong, et al. "Accelerating Langevin sampling with birth-death."
> [4] Karras, Tero et al. "Analyzing and Improving the Training Dynamics of Diffusion Models."
> [5] Xu, Jiazheng et al. "ImageReward: Learning and Evaluating Human Preferences for Text-to-Image Generation"

---

> > ### Comment · Reviewer_65RJ · 2025-04-02
> >
> > Thanks for your reply.
> >
> > My score will remain positive, and this paper is worth acceptance. The rebuttal enhances my understanding of the FKC.
> >
> > However, current theory cannot suggest why FKC does better in the ambient and provides no help in the latent space.
> > I believe the latent/ambient should not matter with the reweighting mechanism, it may be due to other model-related reasons.
> > And its downstream application in CV may not be that promising.
> > Therefore, I cannot give a higher score.

---

> > > ### Author Response · Authors · 2025-04-02
> > >
> > > Thank you for your swift response! We are glad to hear that our rebuttal 'enhances your understanding' of the proposed methodology and that you believe the paper is worthy of acceptance.
> > >
> > > We would like to highlight that the main objective of the proposed method is to sample from the modified densities rather than to directly improve the quality of image generation. This discrepancy in motivation is likely the reason why the method does not improve the performance of Stable Diffusion XL. Indeed, such factors as the quality of the VAE model and the quality of the latent diffusion model define (a) the accuracy of the learned scores, which is crucial for our method (b) whether better sampling from the geometric average of densities in the latent space $q_t^{(1-\beta)}(z|\emptyset)q_t^{\beta}(z|c)$ result in better image generations after decoding.
> > >
> > > However, the results on the sampling task and other generative modeling tasks suggest that the practice agrees with our theoretical findings, i.e. introducing FKC allows for more accurate sampling from the modified densities.
> > >
> > > If you have any further suggestions for how we can improve our work and potentially your evaluation, please do let us know!

---

### Official Review · Reviewer_pXP5 · 2025-03-12

**Overall Recommendation:** 3

**Summary:**

This paper points out modifying score function of the pretrained generative modelings, such as classifier-free guidance might cause to generate samples that are not from the same distribution as the training data, and the corrector schemes used to address this problem either requires infinite many steps requiring more computation resources or is detrimental to the sampling quality. This paper proposes Feynman-Kac PDEs, aiming to generate samples from the same training data distribution and improve sampling efficiency.

**Claims And Evidence:**

For Section 3, this proposes composing a few diffusion models during inference time, especially the product and Geometric Avg examples,  but I am not sure if this is a proper conduct. As [1] proves and empirically shows that the reverse SDE induced by composed score function do not correspond to sampling from the composed model and the reverse diffusion sampling will generate incorrect samples from composed distributions. Instead, they propose to sample with Langevin dynamics.  Based on my understanding, Feynmann Kac PDEs have SDE component embedded, so I doubt simply performing sampling process according to what this paper introduces will give wrong samples too. I might make the wrong connection and correct me if I am wrong.

[1] Du, Y., Durkan, C., Strudel, R., Tenenbaum, J.B., Dieleman, S., Fergus, R., Sohl-Dickstein, J., Doucet, A. and Grathwohl, W.S., 2023, July. Reduce, reuse, recycle: Compositional generation with energy-based diffusion models and mcmc. In International conference on machine learning (pp. 8489-8510). PMLR.

**Essential References Not Discussed:**

I think it is complete.

**Experimental Designs Or Analyses:**

This paper proposes doing SMC and Jump Process for resampling to improve the performance, but I don't find the details specifically for each experiment. For example, this paper mentions SMC is only done when $t \in [t_{\mathrm{min}}, t_{\mathrm{max}}]$, but those $t$'s are not mentioned in the experiment. I also cannot find if both of resampling methods are really used during the inference time, as the tables or figures do not show for them. An ablation study is required if so.

With all those two methods plugged in for resampling, I am also wondering the latency and computational cost-.

In table 5, the experiment is missing when $\beta=7.5$ and FKC presents. I cannot see why this is not done.

**Methods And Evaluation Criteria:**

I think the experimental designs for performance evaluation are reasonable.

**Other Comments Or Suggestions:**

No.

**Other Strengths And Weaknesses:**

The paper is written well-structured and complete.

Again, the weaknesses would be I did not notice the computation cost is reported to support the true efficiency of the proposed approach. Otherwise, the resampling process should be very costly.

**Questions For Authors:**

1. For equation 6, I cannot really see how $$\frac{\partial p_t^w(x)}{\partial t} = \bar{g}_t(x) p_t^w(x)$$ is derived. I quickly skimmed through the Appendix but did not find it explained anywhere.

**Relation To Broader Scientific Literature:**

1. This paper derives PDEs to describe the time-evolution of sample density under the standard SDEs. This makes the sampling process more flexible.
 2. This papers propose a few composition approaches, such as annealed, product and geometric average distributions. Again, as I have mentioned, the correctness is doubted as the reverse SDEs wouldn't give the right distribution [1].

[1] Du, Y., Durkan, C., Strudel, R., Tenenbaum, J.B., Dieleman, S., Fergus, R., Sohl-Dickstein, J., Doucet, A. and Grathwohl, W.S., 2023, July. Reduce, reuse, recycle: Compositional generation with energy-based diffusion models and mcmc. In International conference on machine learning (pp. 8489-8510). PMLR.

**Theoretical Claims:**

I did not spot issues of the proofs or claims except the one I mentioned in the ``Claims and Evidence'' section.

---

> ### Author Rebuttal · Authors · 2025-04-01
>
> We thank the reviewer for the constructive feedback. We are glad that the reviewer finds our paper to be 'well-structured and complete' and the experimental design to be 'reasonable'. We also would like to thank the reviewer for bringing up reference [1] (Du et al.), as this is an excellent reference for clarifying our contributions.
>
> Before addressing the concerns raised, we would like to clarify a potential misunderstanding about the proposed method.
> 1. The main objective of the proposed method is to modify the target density at the inference time (e.g. by sampling from the annealed distribution or product of densities) rather than 'sampling from the training data and improving sampling efficiency'.
> 2. In complete agreement with [1], our work shows that simulating the reverse SDEs with modified scores **does not** sample from the target densities. This is exactly the goal of introducing the Feynman-Kac corrector scheme, which, as we prove, allows for consistent sampling from the target densities by re-weighting and re-sampling. We have added a self-contained proof of Eq. 9 in the updated Appendix to further emphasize this claim.
>
> Next, we would like to address the salient concerns raised by the reviewer individually:
>
> > the choice of $t_\max, t_\min$
>
> The choice of the interval $[t_\min, t_\max]$ depends on the hyperparameters and, especially, the noise schedule, but does not require significant tuning. In practice, we choose the interval based on the fraction of unique samples resampled at every iteration. We report the corresponding plot in Fig. 1 of [the PDF](https://anonymous.4open.science/r/FKC-D7F0/rebuttal.pdf) for the annealing experiments. Note that the scale of the weights is proportional to the noise $\sigma_t$ (see Prop 3.2), which results in a low number of unique samples close to $t=1.0$ due to the Variance Exploding schedule.
>
> For molecule generation experiments in Table 3, we selected $t_\max$ based on a validation task by going over the grid $[0.6,0.7,0.8,0.9,1.0]$; $t_\min$ was always set to $0$.
>
> We have also added a new set of molecule experiments for a harder set of tasks using a model that directly predicts the 3D coordinates of the atoms in a molecule from [2] (Zhou et al.) and docks molecules to a set of target protein pairs. Again, we use a validation task to sweep over $t_\max$ values, which we report in Table 2 of [the PDF](https://anonymous.4open.science/r/FKC-D7F0/rebuttal.pdf).
>
> > Ablation study for the resampling methods
>
> We perform the ablation study of the two considered resampling methods in Table 6 of the Appendix. We find that systematic resampling always performs better or comparably and opt to use it throughout the rest of the empirical study.
>
> > Computational cost of the resampling step
>
> The additional computational cost of the resampling step is negligible compared to the reverse SDE simulation. Indeed, all the weights depend only on scores, which are already evaluated at the forward pass. We purposefully avoid the computation of the divergence operators when deriving the integration scheme (see Lines 176-189 L). The cost of the systematic resampling step is equivalent to generating one uniform random variable, which is negligible.
>
> Our proposed method is notably more convenient than [1], which requires changing the diffusion model parameterization and performing additional MCMC steps or Metropolis-Hastings tests.
>
> > $\beta = 7.5$ for FKC on SDXL
>
> As requested by the reviewer, we provide results for both values of $\beta$ on the Geneval benchmark [3] (Dhruba Ghosh et al.). See the detailed image-generation study in response to Reviewer 65RJ.
> |Model|$\beta$|Overall|Single object|Two object|Counting|Colors|Position|Color attribution|
> |-|-|-|-|-|-|-|-|-|
> |SDXL (Rerun) |7.5|**0.57**|0.99|0.80|0.46|0.86|0.11|0.22|
> |FKC|5.5|**0.58**|0.99| 0.77|0.49|0.87|0.10|0.22|
> |FKC|7.5|**0.57**|0.99|0.78|0.46|0.83|0.13|0.23|
>
> > Derivation of equation (6)
>
> Equation (6) can be understood by the separation of variables. Indeed,
> $$\frac{\partial_t p_t^w(x)}{p_t^w(x)}=\partial_t\log p_t^w(x)=\bar{g}_t(x)\implies p_t^w(x)=p_0^w(x)\exp(\int_0^t ds \bar{g}_s (x)),$$
> where the exponent part corresponds to the update of the weights $dw_t=\bar{g}_t(x)dt$. Note that in eq. (9), $w$ appears as log-weights for Self-Normalized Importance Sampling.
>
> Note that Eq. (6) is meant to *introduce* the reweighting evolution. In later sections, we *derive* the weights which preserve the particular target distribution evolution under simulation or transport by an SDE with given drift/diffusion.
>
> ### Closing remarks
>
> Again, we thank the reviewer for their questions, which gave us the opportunity to clarify and improve our work. We hope that our answers fully address all the important questions raised by the reviewer, and we are happy to consider any additional questions or further suggestions.
>
> We kindly ask the reviewer to consider increasing their score if our responses address their concerns satisfactorily.

---

### Official Review · Reviewer_jpZY · 2025-03-14

**Overall Recommendation:** 4

**Summary:**

The paper derives a suite of new tools for modifying pretrained diffusion models at inference time, using particle resampling techniques. In particular, they use the Feynman-Kac formula to derive the evolution of weights for particles when simulating the diffusion reverse SDE (or different variants of it) such that resampling using the weights results in samples from 1) tempered versions of the original diffusion marginals at different diffusion times 2) products of the marginals of two diffusion models 3) geometric averages of the marginals of two diffusion models. These weighting terms are derived for a plethora of different SDEs, and the paper focuses experimentally on ones where the 1) score function is scaled with a scalar 2) both the score and SDE noise are scaled with specific scalars. The paper explores sequential Monte Carlo resampling methods, and jump process reweighting methods. The proposed methods are evaluated on
- different sampling problems, where the ability to change the temperature at runtime is validated
- multi-property molecule generation, where the authors use the method to generate molecules that simultaneously inhibit multiple proteins
- image generation, where the use of the method as a classifier-free guidance replacement is investigated

**Claims And Evidence:**

I think that the claims are supported by evidence.

**Essential References Not Discussed:**

I am not aware of essential references not discussed, although I am not also particularly familiar with the literature on using SMC to guide diffusion models.

**Experimental Designs Or Analyses:**

I think that the experimental design, and their analysis made sense, but I didn’t check the their validity in a lot of detail.

**Methods And Evaluation Criteria:**

I think that all of the datasets chosen make sense, each tests one clear ability of the collection of methods proposed. The evaluation criteria are sensible as well.

**Other Comments Or Suggestions:**

-

**Other Strengths And Weaknesses:**

I think that this is overall a very nice paper, and a useful contribution to the literature. The paper opens up new tools for diffusion guidance to the community, is well written and organised, and experiments are done on multiple examples, showing the

A weakness of the paper is that some practical details regarding the experiments are a bit more complicated than the theory would imply, and some hyperparameters exist that are not discussed at length in the main paper. I highlight two parts to kick of discussion:

From section 4:
“We find resampling only over an ‘active interval’ t ∈ [tmin, tmax] useful for improving sample quality and preserving diversity, and set weights to zero outside of this interval”

- What is this interval in the experiments used? If this requires significant tuning for different data sets, that seems to be a downside of the method. Regardless, this should be detailed in the paper.


From the Appendix, regarding the compositional molecule generation experiments:

“In practice, we find that the FKC weights have a large variance during molecule generation. This is problematic, as a large number of samples are thrown away. Furthermore, we noted that the score was not always well-conditioned. To ameliorate this, we divided the weights by a set temperature term (T = 100) to reduce their variance before resampling, clipped the top 20% to account for any score instabilities, and did early-stopping (only resampled for 70% of the timesteps).”

- Do the authors have intuition on why is it necessary to divide the weights by a set term 100 in this example? Is this done for the other experiments as well? 100 seems like a large number to use here, and a significant difference from the method implied by the mathematics seems, on the face of it, to raise into question the relevance of the mathematical method definition.
- That said, I suppose it makes sense that the particle resampling may be especially high variance in high dimensions?
- Is early stopping the right phrase to use here, as usually in machine learning it refers to a regularisation method for neural network training?

The image generation results seem quite okay, but there are not a lot of practical details. Weight stabilization techniques used? Active interval choices for resampling? How many particles? I understand that this is not necessarily the highlight of the paper, but since the experiments are there, it would be useful to include the details as well.

**Questions For Authors:**

See “Other strengths and weaknesses”

**Relation To Broader Scientific Literature:**

The paper presents methodological progress in the context of applying inference-time controls to pretrained diffusion models using particle resampling methods. The paper presents many new tools, and, e.g., the ability to temper the target distribution has not been considered in previous work, to the best of my knowledge. This could serve as a very useful building block for diffusion methods targeting at sampling from unnormalised densities. Perhaps the most new important contribution is the systematic mathematical exploration of different reverse SDEs and their impact on the resampling schemes, and the paper could serve as a helpful reference for future work to build on.

**Theoretical Claims:**

I did go through Proposition B.1 and B.2 and didn’t find issues. But there are many more propositions in the paper.

---

> ### Author Rebuttal · Authors · 2025-04-01
>
> We thank the reviewer for their detailed feedback! We are thrilled to hear that they find our paper to be a 'useful contribution to the literature', 'well written and organized' and a new tool for the diffusion sampling community. Below, we address the reviewer's questions.
>
> > What is the interval $[t_\min, t_\max]$ in the experiments used? Does it require significant tuning?
>
> The choice of the interval $[t_\min, t_\max]$ depends on the hyperparameters and, especially, the noise schedule, but does not require significant tuning. In practice, we chose the interval based on the fraction of unique samples resampled at every iteration during the resampling step. We report the corresponding plot in Fig. 2 of [the PDF](https://anonymous.4open.science/r/FKC-D7F0/rebuttal.pdf) for the annealing experiments. Note that the scale of the weights is proportional to the noise $\sigma_t$ (see Prop 3.2), which results in a low number of unique samples close to $t=1.0$ where the variance of the noise is the largest due to the Variance Exploding schedule. We have added corresponding plots and discussion to the manuscript.
>
> > Why is dividing the weights by some constant factor $T$ is necessary? I suppose it makes sense that the particle resampling may be especially high variance in high dimensions?
>
> Indeed, as the reviewer points out, the variance of the weights grows with the number of dimensions due to being proportional to the norm of the score. For the latent molecule generation (8000 dimensions), we had to divide the weights by $T=100$ and clip lowest and largest $20\\%$ of the weights in order to achieve reasonable variance of the weights. To verify that the need for such heuristics is caused by the dimensionality, we did a new empirical study for molecule generation in the coordinate space, where the dim. is 3 * number of atoms in the molecule  (<100). Here, we neither divided nor clipped the weights and only chose the interval $t_\max$ based on a validation set. We present new results below and have added them to the manuscript.
>
> > Is early stopping the right phrase to use here?
>
> Thank you for suggesting a better explanation! The closest analog to selecting $[t_\min, t_\max]$ is the reducing the time interval for the integration of diffusion models, i.e. instead of integrating $t\in[0,1]$, one usually integrates $t\in[ε,1]$, where ε is small constant. We have changed this explanation in the manuscript.
>
> > Practical details on image generation
>
> For image generation, we used Stable Diffusion XL with a variance-preserving SDE and an Euler–Maruyama solver, running for 100 steps. All the computations are done with float16 precision, which is crucial for stable generation via SDXL. All images are generated in 1024x1024 resolution. In the FKC portion of our experiments, we did not apply weight rescaling; moreover, we resampled all 64 particles after each time step throughout the entire time interval.
>
> ## New molecule experiments
> We did a new set of molecule experiments using a model that directly predicts the 3D coordinates of the atoms in a molecule from [1] on a harder set of tasks of generating molecules that dock to a pair of proteins simultaneously, expanding on the promising results from Table 4 of the original submission.  For 14 proteins pairs, we generated 32 molecules at 5 different molecule sizes (160 molecules per pair) using FKC product and found that it improved nicely over two SOTA methods:
>
> ---
> **Table: Docking scores of generated ligands for 14 protein target pairs (P₁, P₂).**  Lower docking scores are better.
> |||(P₁ * P₂) (↑)|max(P₁, P₂) (↓)|P₁ (↓)|P₂ (↓)|Div. (↑)|Val. & Uniq. (↑)|
> |-|-|-|-|-|-|-|-|
> |P₁ only [2] ||62.77±23.74|-7.30±1.90|-8.38±1.51|-7.44±1.93|**0.89±0.01**|**0.95±0.07**|
> |β|FKC|||||||
> |0.5|no [1]|64.35±21.54|-7.14±2.12|-7.90±1.99|-7.96±1.67|**0.89±0.01**|0.89±0.21|
> ||yes|64.05±31.21|-6.86±3.26|-7.89±2.90|-7.92±2.42|0.88±0.02|**0.95±0.11**|
> |1.0|no|69.03±21.61|-7.54±1.74|-8.24±1.71|-8.30±1.53|**0.89±0.01**|0.90±0.19|
> ||yes|69.83±32.70|-7.40±2.93|-8.51±1.82|-8.27±2.88|0.85±0.02|0.92±0.10|
> |2.0|no|68.12±18.56|-7.40±2.03|-8.21±1.66|-8.11±1.62|0.88±0.01|0.94±0.16|
> ||yes|**75.54±23.26**|**-7.91±1.62**|**-8.60±1.62**|**-8.66±1.55**|0.81±0.05|0.88±0.09|
>
> ---
> [1] Zhou et al. "Reprogramming Pretrained Target-Specific Diffusion Models for Dual-Target Drug Design." NeurIPS (2024).
> [2] Guan et al. ICLR (2023).
>
> > Implementation details
>
> For molecules, we chose the time interval based on a validation set of 1 protein pair at 5 molecule lengths (32x5 generated molecules). We kept $t_{\text{min}}$ at 0 and sweep over values of $t_{\text{max}}$ for $\beta=2$, SDE type="Target score", and FKC="on". We show this and additional ablations over $\beta$ and SDE type in [this PDF](https://anonymous.4open.science/r/FKC-D7F0/rebuttal.pdf). Setting $t_{\text{max}}$ to 0.6 gives a good tradeoff in terms of generating molecules that perform well vs. maintaining diversity, and so we proceed with $t_{\text{max}} = 0.6$.

---

### Official Review · Reviewer_3HLs · 2025-03-18

**Overall Recommendation:** 4

**Summary:**

Prior work has shown that composing multiple pre-trained diffusion models is not straightforward in the context of energy-based models. This paper investigates Feynman-Kac correctors based on the Feynman-Kac formula to sample from annealed, geometric-average, and product distributions. Compared to the Fokker-Planck equation, a reweighting function is introduced, allowing for the incorporation of Sequential Monte Carlo resampling schemes. Experimental results demonstrate the effectiveness of the proposed algorithm.

**Claims And Evidence:**

Yes.

**Essential References Not Discussed:**

No.

**Experimental Designs Or Analyses:**

Yes.

**Methods And Evaluation Criteria:**

Yes.

**Other Comments Or Suggestions:**

No.

**Other Strengths And Weaknesses:**

Strengths:

- The paper is generally well-structured and theoretically sound.

- The paper proposes Feynman-Kac correctors based on the FK formula and apply it to different scenarios, including anealed, geometric-average, and product distributions derived from pre-trained diffusion models.

Weaknesses:

- Please see the questions below for the authors.

**Questions For Authors:**

- When comparing Eq.(5) and Eq.(7), an additional term is added. I am curious whether this arises from the evolving normalizing constant $Z_{t}$ in the context of sampling scenario? In contrast, does the vanilla FP equation in Eq. (5) describe the evolution of distributions while keeping $Z_{t}$ unchanged?

- To obtain FK PDE in Eq.(7), is it simply the addition of the FP equation and the reweighting equation? I guess there may be something missing before the definition of the FK PDE.

**Relation To Broader Scientific Literature:**

This paper investigates Feynman-Kac correctors based on the Feynman-Kac formula to sample from annealed, geometric-average, and product distributions.

**Theoretical Claims:**

Yes, I checked some of proofs.

---

> ### Author Rebuttal · Authors · 2025-04-01
>
> We thank the reviewer for their time, feedback, and positive appraisal of our work. We are heartened to hear that the reveiwer feels that the paper is "theoretically sound" and found that the paper was "generally well-structured". We now address the questions and suggestions raised by the reviewer.
>
> > When comparing Eq.(5) and Eq.(7), an additional term is added. I am curious whether this arises from the evolving normalizing constant $Z_t$ in the context of sampling scenario? In contrast, does the vanilla FP equation in Eq. (5) describe the evolution of distributions while keeping $Z_t$ unchanged?
>
>   Both PDEs in eqs. (5) and (7) preserve the normalization of the density in the sense that $\int dx ~ p_t(x) = 1$ for all $t$, although the evolution of density changes with the addition of weighting terms.
>
>   Note that the normalization constants $Z_t$ defined in eq. (13) change in time to guarantee that the density on the left hand side is normalized, e.g. $\int dx ~ p_t^{\text{prod}}(x) = 1$. This is due to defining the density up to a constant, i.e. $p_t^{\text{prod}}(x) \propto q_t^1(x)q_t^2(x)$.
>    Note that, while the integral term $\int g_t(x) p_t(x) dx$ in eq. (6) ensures normalization for the density evolution PDE, our SMC resampling can proceed with access to only the unnormalized weights $g_t(x)$ (see eq. (9))
>
>
> > To obtain FK PDE in Eq.(7), is it simply the addition of the FP equation and the reweighting equation? I guess there may be something missing before the definition of the FK PDE.
>
>   FK PDE in eq. (7) is a more general type of PDEs than the FP equation in eq. (5). Indeed, the main difference between these PDEs is the reweighting term (the last term of FK PDE, i.e. eq. 6) that allows for simulation of processes not possible with FP equation. For instance, FK PDE can change the weights of disconnected modes by reweighting the samples without transporting them, while FP PDE would have to transport samples from one mode to another (via the vector field or noise) to adjust the relative weights. We will emphasize this point in the final version of the paper.
>
> We would like to thank the reviewer for their time and feedback. We hope our answers here allow the reviewer to continue to positively endorse our paper, and we would happily clarify any addition questions which may arise.

---

> > ### Comment · Reviewer_3HLs · 2025-04-04
> >
> > Thank you to the authors for their responses, which are clear to me.
> >
> > During the rebuttal period, I have another question:
> >
> > - In the appendix, Eqs. (74-75) and Eqs. (87-88) give the same marginal distributions. It is possible that, in some cases, multiple solutions exist. How should one choose between them? In this particular case, we should prefer Eqs. (74-75) over Eqs. (87-88), since Eq. (88) contains a Laplacian term?

---

> > > ### Author Response · Authors · 2025-04-04
> > >
> > > We are happy to hear that our initial response clearly answers the reviewer's questions!
> > >
> > > > In the appendix, Eqs. (74-75) and Eqs. (87-88) give the same marginal distributions. It is possible that, in some cases, multiple solutions exist. How should one choose between them? In this particular case, we should prefer Eqs. (74-75) over Eqs. (87-88), since Eq. (88) contains a Laplacian term?
> > >
> > > This is absolutely correct, as we discuss in Section 2.3., a given PDE can be simulated in multiple ways. This is achieved by moving the terms between the continuity equation, the diffusion equation and the reweighting equation and changing their interpretation.
> > >
> > > In the current paper, we consider two main motivations for the choice of the simulation scheme (see Lines 176-188 Left):
> > > 1. **Computational cost**: As the reviewer correctly points out, we should prefer (74-75) as they avoid expensive evaluation of the Laplacian and the divergence operators (87-88) during inference.
> > > 2. **Sampling efficiency**: In general, we should choose the scheme that maximizes sampling efficiency, i.e. minimizes the variance of the weights at minimal computational cost. This choice may depend on the specific setting and application. For example, in the annealing task (section 5.1 and Table 2), the choice of the scheme (tempered noise vs. target score) has differing performance at different annealing temperatures, with target score working better at lower target temperatures (more annealing).
> > >
> > > We thank the reviewer for their insightful question, we will be sure to include further discussion of these choices in the revised manuscript. We are happy to provide any further clarification and answer any questions that may arise.

---

### Decision · Program_Chairs · 2025-05-01

**Decision:**

Accept (spotlight poster)

**Comment:**

The paper introduces the concept of Feynman-Kac Correctors (FKCs), which are a powerful and versatile tool for inference time control in diffusion models, enabling combination of different models (product of experts), annealing and guidance (conditional sampling). The work is a refreshing new perspective on an important problem which has mostly seen incremental advances and almost no conceptual leaps the last years. Simultaneously the paper is practical and readily applies to practical problems as evidenced by the extensive experimental section, which was further expanded during the rebuttal-discussion phase. The reviewers unanimously agree that this paper should be accepted.